# MolEditRL: Structure-Preserving Molecular Editing via Discrete Diffusion and Reinforcement Learning

**Yuanxin Zhuang**[1]**, Dazhong Shen**[2]*,**Ying Sun**[1]*

[1] Artificial Intelligence Thrust, Hong Kong University of Science and Technology (Guangzhou)
[2] College of Computer Science and Technology, Nanjing University of Aeronautics and Astronautics
`yzhuang436@connect.hkust-gz.edu.cn`
`shendazhong@nuaa.edu.cn, sunyinggilly@gmail.com`

## Abstract

Molecular editing aims to modify a given molecule to optimize desired chemical properties while preserving structural similarity. However, current approaches typically rely on string-based or continuous representations, which fail to adequately capture the discrete, graph-structured nature of molecules, resulting in limited structural fidelity and poor controllability. In this paper, we propose MolEditRL, a molecular editing framework that explicitly integrates structural constraints with precise property optimization. Specifically, MolEditRL consists of two stages: (1) a discrete graph diffusion model pretrained to reconstruct target molecules conditioned on source structures and natural language instructions; (2) an editing-aware reinforcement learning fine-tuning stage that further enhances property alignment and structural preservation by explicitly optimizing editing decisions under graph constraints. For comprehensive evaluation, we construct MolEdit-Instruct, the largest and most property-rich molecular editing dataset, comprising 3 million diverse examples spanning single- and multi-property tasks across 10 chemical attributes. Experimental results demonstrate that MolEditRL significantly outperforms state-of-the-art methods in both property optimization accuracy and structural fidelity, achieving a 74% improvement in editing success rate while using 98% fewer parameters.

## 1 Introduction

Designing molecules with tailored properties is essential for drug discovery Ma et al. (2024); Huang et al. (2025). Unlike *de novo* molecular generation that creates molecules from scratch Wang et al. (2022), molecular editing Hui et al. (2022) focuses on precisely modifying existing molecules to optimize targeted properties while preserving known structure-activity relationships Hansch (1969).

Traditional molecular editing approaches fall into three main paradigms: (1) Rule-based methods apply predefined fragment transformations Chen et al. (2021); Fu et al. (2021), but their generalization is limited by manually designed rules. (2) Latent generative modelsJin et al. (2018); Shi et al. (2020) optimize molecular

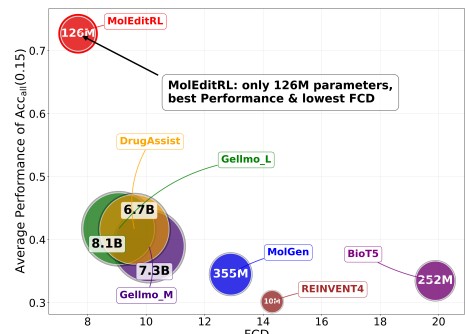

Figure 1: Performance, FCD, and parameter size comparison.

---

*Dazhong Shen and Ying Sun are co-corresponding authors.

properties in continuous spaces, yet often struggle with fine-grained control due to latent compression. (3) Sequence-to-sequence methodsHe et al. (2021); Loeffler et al. (2024); Wu et al. (2024) frame editing as SMILES translation, enabling scalable learning but lacking structural precision, as small token changes can produce unpredictable or invalid edits Kusner et al. (2017); Krenn et al. (2020).

Recently, language models have expanded the landscape of molecular editing by integrating natural language understanding with chemical representations, enabling models to leverage semantic instructions for molecular modifications: (1) Graph-embedding approach Liu et al. (2023) encodes both molecules and textual instructions into a shared latent space via contrastive learning. (2) SMILES-based models Ye et al. (2025); Le & Chawla (2024); Dey et al. (2025b) use retrieval-augmented generation or adopt instruction tuning to enhance editing relevance. (3) SELFIES-based method Fang et al. (2024) incorporates chemical feedback to reduce syntax errors and improve property alignment during generation. Despite their promise, these language-based methods often struggle to preserve scaffolds or perform precise, localized modifications, due to the non-uniqueness of textual molecular representations. Structurally similar molecules may appear textually distant, leading to inconsistent and unreliable edits Noutahi et al. (2024); Arús-Pous et al. (2019).

There are several critical challenges in molecular editing for ensuring structural preservation and editing precision: First, molecular editing must explicitly align with the discrete, graph-structured nature of molecules Gong & Sun; Shen et al. (2024). String-based representations fail to explicitly encode topological constraints, often limiting the models ability to preserve scaffolds and perform localized modifications. Second, existing methods trained solely on fixed datasets lack mechanisms for actively exploring novel editing strategies, restricting generalization and adaptability to complex or underexplored regions of chemical space. Third, performing discrete edits directly on molecular graphs while preserving structural fidelity and aligning with natural language instructions is technically challenging due to the non-differentiable, high-dimensional nature of graph representations.

In this paper, we propose MolEditRL, a structure-aware molecular editing framework that combines discrete graph diffusion with reinforcement learning. MolEditRL first employs discrete diffusion to reconstruct target molecules conditioned simultaneously on source molecular structures and natural language instructions, effectively capturing both structural and semantic relationships. To further enhance the precision of property optimization and alignment with instructions, we introduce editing-aware reinforcement learning guided by explicit property rewards, while incorporating constraints to preserve structural integrity. To enable comprehensive evaluation, we introduce MolEdit-Instruct, a large-scale molecular editing dataset containing 3 million editing examples spanning 10 diverse chemical properties, including biological activities, physicochemical attributes, and synthetic accessibility. Compared to existing datasets Ye et al. (2025); Dey et al. (2025b), MolEdit-Instruct provides broader property coverage and more realistic single- and multi-property editing scenarios. We release MolEdit-Instruct publicly on Hugging Face to facilitate future research.

Experimental results demonstrate that MolEditRL significantly outperforms state-of-the-art methods in both editing accuracy and distributional fidelity (measured by Fréchet ChemNet Distance, FCD). Remarkably, MolEditRL achieves a 74% improvement in editing success rate over leading baselines while requiring 98% fewer parameters (Figure 1). Our contributions are summarized as follows: (1) We propose MolEditRL, a molecular editing framework explicitly designed to maintain structural integrity during editing. (2) We introduce a two-stage training strategy that combines discrete diffusion pretraining with reinforcement learning fine-tuning, achieving precise property optimization with structural constraints. (3) MolEditRL achieves SOTA editing performance with substantially fewer parameters and the lowest distributional distance (FCD) compared to existing methods.

## 2 RELATED WORKS

**Molecular Editing.** Molecular editing aims to modify a given molecule to enhance specific chemical properties while preserving its structural similarity. Formally, given a source molecule $G_{src}$ and a textual instruction $S$ describing desired modifications, the goal is to generate an edited molecule $G_{tgt}$ that satisfies both the semantic intent of $S$ and structural similarity constraints. This formulation enables flexible, user-centric molecular design where optimization objectives can be expressed intuitively. Existing molecular editing approaches typically fall into three main paradigms: (1) Rule-based Graph Editing. These methods directly manipulate molecular graphs using predefined

or data-driven transformation rules, such as fragment replacements or bond editing templates, inspired by Matched Molecular Pairs (MMP) Dalke et al. (2018); Chen et al. (2021); Fu et al. (2021). While offering high chemical interpretability and precise local modifications, their generalizability is limited by the coverage and flexibility of manually or heuristically derived rules. (2) Latent Generative Graph Editing. Approaches such as JT-VAE Jin et al. (2018) and GraphAF Shi et al. (2020) encode molecules into continuous latent spaces and decode edited structures by sampling. Hierarchical decoding techniques like HierG2G Jin et al. (2020); Zhuang et al. (2025) enhance structural preservation by generating molecules in a coarse-to-fine manner. However, these methods frequently face issues such as information loss due to latent compression, resulting in limited accuracy and insufficient control over edits. (3) Sequence-based Generation. These approaches treat molecular editing as a sequence translation task, converting source SMILES strings into target SMILES using Transformer-based architectures He et al. (2021); Loeffler et al. (2024); Wu et al. (2024). These models suffer from syntactic instability and representation ambiguity: structurally similar molecules can have significantly different SMILES representations, and small token-level edits may lead to unpredictable or chemically invalid outputs, limiting their precision and structural controllability. (4) Language-based models. MOLGEN Fang et al. (2024) addresses SMILES fragility by adopting the SELFIES representation. Methods such as ChatDrug Liu et al. (2024a), DrugAssist Ye et al. (2025), and Re2DF Le & Chawla (2024) utilize retrieval-augmented generation or instruction tuning to enhance editing relevance. Additionally, embedding-based methods leveraging diffusion Xiong et al. (2024) or contrastive learning Liu et al. (2023) have been proposed. Nonetheless, these approaches continue to rely on textual or continuous representations that lack explicit alignment with discrete molecular graph structures, compromising structural fidelity and editing accuracy.

**Reinforcement Learning in Molecular Generation.** Reinforcement learning (RL) provides a flexible framework for molecular optimization by formulating molecule generation as a Markov Decision Process (MDP), in which agents sequentially modify molecular structures to maximize rewards associated with desired chemical properties Sridharan et al. (2024); Ji et al. (2025); Sun et al. (2024); Ji et al. (2026). SMILES-based RL methods such as ReLeaSE Popova et al. (2018) and REINVENT Olivecrona et al. (2017) guide generative models using property predictors and prior policies. Graph-based RL methods, including MolGAN De Cao & Kipf (2018), GCPN You et al. (2018), and MolDQN Zhou et al. (2019), facilitate goal-directed graph construction through adversarial training, policy gradients, or Q-learning. Although effective in exploring chemical space, these RL-based frameworks typically focus on *de novo* molecule generation and lack explicit mechanisms to enforce structural constraints derived from source molecules, limiting their applicability to structurally constrained molecular editing tasks.

## 3 METHOD

We present MolEditRL, a structure-preserving molecular editing framework trained in two stages. First, molecules and instructions are encoded into unified graph-text representations. Then, a structure-aware editing network is trained via (1) discrete diffusion pretraining to reconstruct target molecules from noisy graphs and instructions, and (2) reinforcement learning fine-tuning to optimize property alignment while preserving structural fidelity.

### 3.1 MOLECULAR TOKENIZING

As illustrated in Figure 2 (a), a molecule is represented as an attributed graph $G = (V, E)$, where $V$ denotes atom nodes with associated features, and $E$ denotes bond edges with bond-type attributes. The editing instruction is a sequence of tokens $S = [s_1, \ldots, s_n]$. Given a source molecule graph $G_{src} = (V_{src}, E_{src})$, the model aims to predict the target graph $G_{tgt} = (V_{tgt}, E_{tgt})$ that reflects the required edits. These components are embedded and concatenated into a unified input sequence:

$$h^0 = [\, h_1, \ldots, h_n, \; h_{n+1}^{src}, \ldots, h_{n+k}^{src}, \; h_{n+k+1}^{tgt}, \ldots, h_{n+k+m}^{tgt} \,] \in \mathbb{R}^{(n+k+m) \times d_h}, \qquad (1)$$

where $h_1, \ldots, h_n \in \mathbb{R}^{d_h}$ are embeddings for the instruction tokens, $h_{n+1}^{src}, \ldots, h_{n+k}^{src} \in \mathbb{R}^{d_h}$ encode source atoms, and $h_{n+k+1}^{tgt}, \ldots, h_{n+k+m}^{tgt} \in \mathbb{R}^{d_h}$ encode target atoms. $d_h$ is the models hidden size, and variable-size graphs are handled via sequence serialization with dynamic padding.

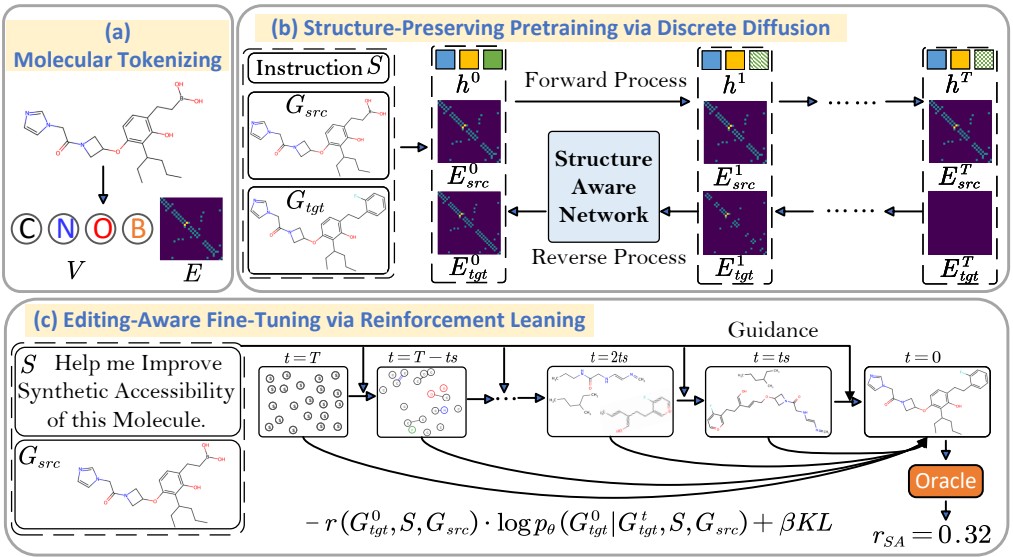

Figure 2: Overview of MolEditRL.

## 3.2 STRUCTURE-PRESERVING EDITING NETWORK

Recent work has demonstrated the effectiveness of unified architectures for multi-modal learning Xiang et al. (2024); Zhao et al. (2025). Such unified frameworks enable better capture of cross-modal dependencies and shared representations while reducing architectural complexity. To enable precise and structure-aware molecular editing, we propose the Structure-Preserving Editing Network, which jointly encodes the semantic intent of natural language instructions and the topological features of source and target graphs. We initialize our transformer encoder with a pretrained RoBERTa model Liu et al. (2019), but enhance it with a structure-aware attention mechanism. This mechanism injects bond-level connectivity priors into attention scores via learnable bias terms that encode graph connectivity, guiding attention flows to preserve structural integrity while selectively updating target representations. For tokens $i$ and $j$ at layer $l$, the raw attention score $\hat{A}_{i,j}^l$ is computed as:

$$\hat{A}_{i,j}^l = \frac{1}{\sqrt{d_k}} \left( h_i^l W_Q \right) \left( h_j^l W_K \right)^\top + b_{i,j}^l, \tag{2}$$

where $W_Q, W_K \in \mathbb{R}^{d_h \times d_k}$ are learnable projection matrices. To incorporate graph-level dependencies and better preserve structure during editing, we introduce $b_{i,j}^l$ to encode structural priors:

$$b_{i,j}^l = \begin{cases} E_{src}[i-n, j-n, :], & \text{if } n < i, j \leq n+k, \\ E_{tgt}[i-(n+k), j-(n+k), :], & \text{if } l = 0 \text{ and } n+k < i, j, \\ A_{i,j}^{l-1}, & \text{if } l > 0 \text{ and } n+k < i, j, \\ 0, & \text{otherwise.} \end{cases} \tag{3}$$

The bias term $b_{i,j}^l$ preserves structure by injecting source adjacency at all layers, using target adjacency at the first layer, and propagating attention from previous layers to maintain topology-aware attention flow. The attention weights are then normalized, and the updated hidden states are calculated as:

$$A_{i,j}^l = \frac{\exp(\hat{A}_{i,j}^l)}{\sum_k \exp(\hat{A}_{i,k}^l)}, \quad h_i^{l+1} = \sum_j A_{i,j}^l \left( h_j^l W_V \right) W_O, \tag{4}$$

where $W_V \in \mathbb{R}^{d_h \times d_k}$ and $W_O \in \mathbb{R}^{d_k \times d_h}$ are learnable parameters. After $L$ transformer layers, the model outputs $\hat{p}(V_{\text{tgt}}) \in \mathbb{R}^{m \times a}$ for atom types and $\hat{p}(E_{\text{tgt}}) \in \mathbb{R}^{m \times m \times b}$ for bond types, where $m$ is the maximum number of atoms per molecule, $a$ the atom vocabulary size, and $b$ the number of bond types. Edge predictions are symmetrized $(e_{i,j} + e_{j,i})/2$ to respect bond-direction constraints.

### 3.3 STRUCTURE-PRESERVING PRETRAINING VIA DISCRETE DIFFUSION

As Figure 2 (b), we pretrain the editing network via a discrete denoising diffusion process conditioned on the source molecule and instruction, enabling topology-aware generation.

**1. Forward Process.** We define a discrete forward process that gradually corrupts the target molecular graph $G_{\text{tgt}}^0$ over $T$ timesteps. At each step $t$, atom and bond features are independently masked with probability $\beta(t) = (T - t + 1)^{-1}$:

$$q(G_{\text{tgt}}^{1:T} \mid G_{\text{tgt}}^0) \;=\; \prod_{t=1}^{T} q(G_{\text{tgt}}^t \mid G_{\text{tgt}}^{t-1}), \quad q(G_{\text{tgt}}^t \mid G_{\text{tgt}}^{t-1}) = \big(V_{\text{tgt}}^{t-1} Q_t^V, \; E_{\text{tgt}}^{t-1} Q_t^E\big), \quad (5)$$

where $Q_t$ is a transition matrix that gradually increases the masking rate and $q()$ denotes the transition distribution of the forward diffusion process. At each step, each element remains the same with probability $1 - \beta(t)$ or transitions to [MASK] with probability $\beta(t)$. This process gradually converts $G_{\text{tgt}}^0$ into a fully masked graph $G_{\text{tgt}}^T$.

**2. Reverse Process.** To recover $G_{\text{tgt}}^0$ from the fully corrupted graph $G_{\text{tgt}}^T$, we train a denoising model $\phi_\theta$ to iteratively refine $G_{\text{tgt}}^t$ conditioned on the source molecule $G_{\text{src}}$ and instruction $S$:

$$p_\theta\big(G_{0:T-1} \mid G_{\text{tgt}}^T, \, G_{\text{src}}, \, S\big) \;=\; \prod_{t=1}^{T} p_\theta\big(G_{\text{tgt}}^{t-1} \mid G_{\text{tgt}}^t, \, G_{\text{src}}, \, S\big). \quad (6)$$

At each timestep, $\phi_\theta$ predicts the denoised graph $G_{\text{tgt}}^{t-1}$ using the editing network described earlier.

**3. Training Objective.** Although instruction $S$ is not corrupted during diffusion, we include a cross-entropy loss on instruction tokens to enforce semantic alignment with the predicted molecule.

$$\mathcal{L}_{pre} = \sum_{i=n+1}^{n+k+m} \text{CE}\big(v_i, \, \hat{p}_i(v_i)\big) \;+\; \sum_{i,j=n+k+1}^{n+k+m} \text{CE}\big(e_{i,j}, \, \hat{p}_{i,j}(e_{i,j}^t)\big) \;+\; \sum_{i=1}^{n} \text{CE}\big(s_i, \, \hat{p}_i(s_i)\big). \quad (7)$$

**4. Sampling.** At inference time we start from a fully masked graph $G_{\text{tgt}}^T$ and iteratively apply the reverse process. Each step: (1) *Graph-Text Encoding*: The transformer $\phi_\theta$ encodes $G_{\text{tgt}}^t$, $G_{\text{src}}$, and $S$, producing logits $\hat{p}(V_{\text{tgt}})$ and $\hat{p}(E_{\text{tgt}})$. (2) *Prediction*: Following $x_0$-parameterization Austin et al. (2021), the model predicts the denoised graph as $\hat{G}_{\text{tgt}}^0 = \arg\max \hat{p}(V_{\text{tgt}}, E_{\text{tgt}})$. (3) *Sampling*: The next graph $G_{\text{tgt}}^{t-1}$ is sampled from the posterior $q(G_{\text{tgt}}^{t-1} \mid G_{\text{tgt}}^t, \hat{G}_{\text{tgt}}^0)$ by independently sampling atoms and bonds: $G_{\text{tgt}}^{t-1} \sim \prod_i p_\theta(v_i^{t-1}) \prod_{i,j} p_\theta(e_{i,j}^{t-1})$.

### 3.4 EDITING-AWARE FINE-TUNING VIA REINFORCEMENT LEARNING

While the pretrained diffusion model captures molecular structure and ensures validity, it lacks explicit optimization for property-specific editing. We address this by introducing Editing-Aware Reinforcement Learning, which fine-tunes the model using rewards computed from well-established chemical toolkits (RDKit Bento et al. (2020) and TDC Huang et al. (2021)). A KL-regularized objective guides optimization toward desired properties while preserving structural consistency.

**1. MDP Formulation for Molecular Editing.** We recast discrete graph denoising as a Markov Decision Process (MDP) tailored specifically for molecular editing Uehara et al. (2024): (1) *State:* $s_t = \big(S, G_{\text{src}}, G_{\text{tgt}}^{T-t}\big)$ includes the instruction, source molecule, and current noisy target graph. (2) *Action:* $a_t = G_{\text{tgt}}^{T-t-1}$ is sampled from the models predicted distribution over denoised graphs at the next step. (3) *Initial State:* $P_0(s_0) = p(S)\, p(G_{\text{src}})\, q(G_{\text{tgt}}^T)$ combines an instruction, a source molecule, and a fully masked target graph. (4) *Transition:* Given a sampled action $a_t$, the next state becomes $s_{t+1} = \big(S, G_{\text{src}}, a_t\big)$. (5) *Policy:* The stochastic policy $\pi_\theta(a_t \mid s_t)$ outputs a categorical distribution over atom and bond types, from which $a_t$ is sampled. (6) *Reward:* A scalar reward is assigned only at the final step ($t = T-1$) to evaluate editing success:

$$R(s_t, a_t) = \begin{cases} r(G_{\text{tgt}}^0, S, G_{\text{src}}), & \text{if } t = T-1, \\ 0, & \text{otherwise,} \end{cases} \quad (8)$$

where $r(\cdot)$ equals 1 if the generated molecule successfully performs the required edit, 0.2 if it is chemically valid but does not fully satisfy the instruction, and 0 if it is invalid.

**2. KL-Regularized RL Objective.** To optimize molecular editing while preserving structural fidelity, we adopt a KL-regularized reinforcement learning objective. Formally, the objective is:

$$\mathcal{L}(\theta) = -\mathbb{E}_{p(S)\,p(G_{\mathrm{src}})}\,\mathbb{E}_{p_\theta(G_{\mathrm{tgt}}^0)}\big[r(G_{\mathrm{tgt}}^0, S, G_{\mathrm{src}})\big]$$
$$+ \beta \sum_{t=1}^{T} \mathbb{E}_{p_\theta(G_{\mathrm{tgt}}^t)}\Big[D_{\mathrm{KL}}\big(p_\theta(G_{\mathrm{tgt}}^0 \mid G_{\mathrm{tgt}}^t, S, G_{\mathrm{src}}) \,\|\, p_{\mathrm{pre}}(G_{\mathrm{tgt}}^0 \mid G_{\mathrm{tgt}}^t, S, G_{\mathrm{src}})\big)\Big], \quad (9)$$

where $p_\theta$ is the current policys distribution over denoised target molecules at timestep $t$, and $p_{\mathrm{pre}}$ is the pretrained diffusion model, acting as a structure-aware prior. The coefficient $\beta$ balances reward maximization and structural consistency and is set to 0.1 in our experiments. To stabilize training, we normalize the final reward of each trajectory within each batch: $\hat{A} = \frac{r - \mathrm{mean}(r)}{\mathrm{std}(r) + 10^{-6}}$. To reduce computation and improve efficiency, we apply policy updates at a fixed stride $t_s$, rather than every timestep. Specifically, we define the update set as: $\mathcal{T} = \{t \in [1, T] \mid t \bmod t_s = 0\}$, and compute gradients only at $t \in \mathcal{T}$. The resulting policy gradient becomes:

$$\nabla_\theta J(\theta) = \mathbb{E}_{G_{\mathrm{tgt}}^{0:T} \sim p_\theta}\Big[\hat{A} \cdot \sum_{t \in \mathcal{T}} \nabla_\theta \log p_\theta(\hat{G}_{\mathrm{tgt}}^0 \mid G_{\mathrm{tgt}}^t, S, G_{\mathrm{src}}) - \beta \sum_{t \in \mathcal{T}} \nabla_\theta D_{\mathrm{KL}}\left(p_\theta \,\|\, p_{\mathrm{pre}}\right)\Big]. \quad (10)$$

**3. Gradient Estimation via $x_0$-Parameterization.** We estimate the gradient of the reward term (i.e., the first term of Eq. 9). By the policy gradient theorem, the gradient is:

$$\nabla_\theta \mathbb{E}_{G_{\mathrm{tgt}}^{0:T}}[r] = \mathbb{E}_{G_{\mathrm{tgt}}^{0:T}}\Big[r(G_{\mathrm{tgt}}^0, S, G_{\mathrm{src}}) \cdot \sum_{t \in \mathcal{T}} \nabla_\theta \log p_\theta(G_{\mathrm{tgt}}^{t-1} \mid G_{\mathrm{tgt}}^t, S, G_{\mathrm{src}})\Big]. \quad (11)$$

Since rewards are only available at t=0, directly estimating the gradient suffers from high variance Liu et al. (2024b). To reduce this, we adopt $x_0$-parameterization Austin et al. (2021), rewriting the reverse transition as:

$$p_\theta(G_{\mathrm{tgt}}^{t-1} \mid G_{\mathrm{tgt}}^t, S, G_{\mathrm{src}}) = \sum_{G_{\mathrm{tgt}}^0} q(G_{\mathrm{tgt}}^{t-1} \mid G_{\mathrm{tgt}}^t, G_{\mathrm{tgt}}^0)\, p_\theta(G_{\mathrm{tgt}}^0 \mid G_{\mathrm{tgt}}^t, S, G_{\mathrm{src}}), \quad (12)$$

where $q(\cdot)$ denotes the corruption distribution from the forward process. This approximation yields:

$$\nabla_\theta \log p_\theta(G_{\mathrm{tgt}}^{t-1} \mid G_{\mathrm{tgt}}^t, S, G_{\mathrm{src}}) \approx \nabla_\theta \log p_\theta(\hat{G}_{\mathrm{tgt}}^0 \mid G_{\mathrm{tgt}}^t, S, G_{\mathrm{src}}). \quad (13)$$

A detailed derivation of this approximation is provided in Appendix Q. Under this formulation, the gradient of the reward term can be approximated via a reward-weighted cross-entropy loss:

$$\sum_{t \in \mathcal{T}} r(G_{\mathrm{tgt}}^0, S, G_{\mathrm{src}}) \cdot \Big(\sum_i \mathrm{CE}\big(v_i^0,\ p_\theta(\cdot \mid G_{\mathrm{tgt}}^t, S, G_{\mathrm{src}})\big) + \sum_{i,j} \mathrm{CE}\big(e_{i,j}^0,\ p_\theta(\cdot \mid G_{\mathrm{tgt}}^t, S, G_{\mathrm{src}})\big)\Big), \quad (14)$$

where $v_i^0$ and $e_{i,j}^0$ are atoms and bonds in the final predicted molecule $G_{\mathrm{tgt}}^0$, reused as supervision targets at each selected step $t \in \mathcal{T}$.

## 4 EXPERIMENTS

### 4.1 DATA CONSTRUCTION

We construct MolEdit, a large-scale and property-rich dataset specifically tailored for molecular editing with natural language instructions. Existing datasets, such as MolOpt-Instructions Ye et al. (2025), MuMOInstruct Dey et al. (2025b) and C-MuMOInstruct Dey et al. (2025a), are limited in either property coverage, task diversity, or data scale. MolEdit addresses these gaps by extending the property set to 10 diverse chemical attributesspanning biological activity, physicochemical characteristics, and synthetic accessibility Zhang et al. (2025)and defining 20 representative editing tasks (10 increases and 10 decreases). It contains 3 million high-quality molecular pairs (967K unique), each exhibiting substantial property shifts while maintaining high structural similarity (Tanimoto scores from 0.650 to 0.982). This provides a more realistic and comprehensive testbed for training and evaluating editing models. Further dataset construction details are provided in Appendix R. The model architecture is described in Appendix B, and the training setup is detailed in Appendix C. Additionally, to further validate our approach against existing methods, we also conducted pretraining and evaluation experiments on the C-MuMOInstruct dataset, with comprehensive results presented in Section 4.5.

Table 1: Comparison of molecular editing models across tasks. Bold indicates best performance. Arrows ($\uparrow$, $\downarrow$) denote desired property increase or decrease.

| Model | Task | Validity | TS≥0.65 Acc$_{all}$ | Acc$_{valid}$ | MCS≥0.6 Acc$_{all}$ | Acc$_{valid}$ | GED≤4 Acc$_{all}$ | Acc$_{valid}$ | FCD | Task | Validity | TS≥0.65 Acc$_{all}$ | Acc$_{valid}$ | MCS≥0.6 Acc$_{all}$ | Acc$_{valid}$ | GED≤4 Acc$_{all}$ | Acc$_{valid}$ | FCD |
|---|---|---|---|---|---|---|---|---|---|---|---|---|---|---|---|---|---|---|
| BioT5 | | 1 | 0 | 0 | 0.0 | 0.0 | 0.066 | 0.066 | 15.00 | | 1 | 0 | 0 | 0.004 | 0.004 | 0 | 0 | 13.40 |
| MolGen | | 1 | 0.024 | 0.024 | 0.032 | 0.032 | 0.018 | 0.018 | 11.61 | | 1 | 0.016 | 0.016 | 0.020 | 0.020 | 0.015 | 0.015 | 14.19 |
| MoleculeSTM | | 0.794 | 0.096 | 0.097 | 0.116 | 0.120 | 0.094 | 0.112 | 12.81 | | 0.728 | 0.074 | 0.086 | 0.098 | 0.115 | 0.044 | 0.063 | 13.59 |
| Reinvent4 | | 0.722 | 0.130 | 0.152 | 0.106 | 0.117 | 0.101 | 0.115 | 21.31 | | 0.582 | 0.010 | 0.017 | 0.048 | 0.063 | 0.056 | 0.061 | 10.98 |
| DrugAssist | GSK3 $\beta\uparrow$ | 0.976 | 0.236 | 0.242 | 0.258 | 0.264 | 0.212 | 0.222 | 9.42 | SA$\downarrow$ | 0.988 | 0.537 | 0.544 | 0.551 | 0.558 | 0.202 | 0.205 | 9.05 |
| Gellm$^3$o_M | | 0.924 | 0.164 | 0.178 | 0.284 | 0.307 | 0.122 | 0.132 | 10.32 | | 0.916 | 0.350 | 0.382 | 0.352 | 0.363 | 0.232 | 0.246 | 8.85 |
| Gellm$^3$o_L | | 0.902 | 0.114 | 0.126 | 0.256 | 0.260 | 0.170 | 0.198 | 9.74 | | 0.888 | 0.238 | 0.268 | 0.262 | 0.276 | 0.194 | 0.208 | 9.10 |
| Gellm$^4$o-C_M | | 0.908 | 0.144 | 0.152 | 0.224 | 0.235 | 0.214 | 0.232 | 9.84 | | 0.864 | 0.326 | 0.377 | 0.401 | 0.409 | 0.217 | 0.229 | 9.85 |
| Gellm$^4$o-C_L | | 0.922 | 0.138 | 0.145 | 0.196 | 0.199 | 0.162 | 0.190 | 10.82 | | 0.849 | 0.218 | 0.226 | 0.224 | 0.248 | 0.104 | 0.143 | 10.67 |
| MolEditRL | | 0.952 | **0.342** | **0.359** | **0.364** | **0.382** | **0.242** | **0.254** | **7.99** | | **0.988** | **0.628** | **0.636** | **0.694** | **0.702** | **0.248** | **0.251** | **7.10** |
| BioT5 | | 1 | 0 | 0 | 0.048 | 0.048 | 0 | 0 | 17.21 | | 1 | 0 | 0 | 0.101 | 0.101 | 0 | 0 | 16.95 |
| MolGen | | 1 | 0.017 | 0.017 | 0.062 | 0.062 | 0.037 | 0.037 | 16.19 | | 1 | 0.072 | 0.072 | 0.094 | 0.094 | 0.063 | 0.063 | 12.57 |
| MoleculeSTM | | 0.741 | 0.044 | 0.049 | 0.066 | 0.080 | 0.048 | 0.052 | 11.93 | | 0.672 | 0.098 | 0.105 | 0.129 | 0.148 | 0.091 | 0.122 | 11.67 |
| Reinvent4 | | 0.701 | 0.143 | 0.241 | 0.113 | 0.164 | 0.106 | 0.109 | 11.70 | | 0.581 | 0.173 | 0.197 | 0.189 | 0.197 | 0.105 | 0.109 | 12.15 |
| DrugAssist | QED$\uparrow$ SA$\downarrow$ | 0.980 | 0.532 | 0.543 | 0.449 | 0.456 | 0.216 | 0.255 | 9.68 | Haccept$\downarrow$ LogP$\uparrow$ | 0.984 | 0.372 | 0.378 | 0.328 | 0.335 | 0.126 | 0.152 | 12.33 |
| Gellm$^3$o_M | | 0.882 | 0.012 | 0.014 | 0.138 | 0.143 | 0.107 | 0.113 | 13.20 | | 0.906 | 0.224 | 0.247 | 0.254 | 0.268 | 0.207 | 0.212 | 10.68 |
| Gellm$^3$o_L | | 0.904 | 0.206 | 0.228 | 0.213 | 0.226 | 0.114 | 0.127 | 9.76 | | 0.904 | 0.130 | 0.144 | 0.168 | 0.176 | 0.138 | 0.149 | 11.24 |
| Gellm$^4$o-C_M | | 0.924 | 0.188 | 0.197 | 0.192 | 0.198 | 0.135 | 0.149 | 10.47 | | 0.905 | 0.237 | 0.243 | 0.254 | 0.259 | 0.169 | 0.176 | 10.91 |
| Gellm$^4$o-C_L | | 0.894 | 0.209 | 0.216 | 0.248 | 0.280 | 0.108 | 0.123 | 11.09 | | 0.911 | 0.220 | 0.223 | 0.248 | 0.251 | 0.184 | 0.192 | 10.83 |
| MolEditRL | | 0.974 | **0.632** | **0.649** | **0.678** | **0.715** | **0.268** | **0.271** | **7.54** | | 0.946 | **0.316** | **0.334** | **0.344** | **0.356** | **0.224** | **0.232** | **10.11** |
| BioT5 | | 1 | 0 | 0 | 0 | 0 | 0 | 0 | 24.32 | | 1 | 0 | 0 | 0.064 | 0.064 | 0 | 0 | 26.24 |
| MolGen | | 1 | 0.039 | 0.039 | 0.075 | 0.075 | 0.012 | 0.012 | 11.95 | | 1 | 0.033 | 0.033 | 0.061 | 0.061 | 0.031 | 0.031 | 13.75 |
| MoleculeSTM | | 0.693 | 0.038 | 0.041 | 0.090 | 0.109 | 0.076 | 0.081 | 11.43 | | 0.638 | 0.014 | 0.016 | 0.032 | 0.035 | 0.040 | 0.046 | 14.87 |
| Reinvent4 | DRD2$\downarrow$ | 0.522 | 0.093 | 0.230 | 0.153 | 0.163 | 0.124 | 0.135 | 11.49 | Haccept$\uparrow$ | 0.638 | 0.017 | 0.163 | 0.103 | 0.112 | 0.091 | 0.107 | 12.08 |
| DrugAssist | MW$\downarrow$ | 0.980 | 0.422 | 0.431 | 0.388 | 0.472 | 0.236 | 0.242 | 9.89 | MW$\uparrow$ | 0.956 | 0.230 | 0.241 | 0.248 | 0.251 | 0.126 | 0.129 | 11.72 |
| Gellm$^3$o_M | SA$\downarrow$ | 0.900 | 0.080 | 0.089 | 0.150 | 0.181 | 0.112 | 0.114 | 10.35 | QED$\downarrow$ | 0.906 | 0.010 | 0.016 | 0.023 | 0.029 | 0.014 | 0.015 | 16.22 |
| Gellm$^3$o_L | | 0.918 | 0.108 | 0.118 | 0.130 | 0.152 | 0.104 | 0.109 | 10.19 | | 0.886 | 0.042 | 0.047 | 0.051 | 0.060 | 0.032 | 0.039 | 15.70 |
| Gellm$^4$o-C_M | | 0.916 | 0.072 | 0.080 | 0.127 | 0.144 | 0.116 | 0.118 | 11.21 | | 0.897 | 0.128 | 0.131 | 0.119 | 0.125 | 0.094 | 0.097 | 11.76 |
| Gellm$^4$o-C_L | | 0.909 | 0.155 | 0.164 | 0.198 | 0.218 | 0.164 | 0.198 | 10.07 | | 0.853 | 0.188 | 0.196 | 0.198 | 0.205 | 0.144 | 0.161 | 10.38 |
| MolEditRL | | 0.986 | **0.518** | **0.525** | **0.548** | **0.566** | **0.252** | **0.261** | **7.28** | | 0.958 | **0.430** | **0.449** | **0.432** | **0.436** | **0.228** | **0.232** | **9.79** |

## 4.2 EVALUATION METRICS

To comprehensively assess molecular editing performance, we use the following metrics to evaluate chemical validity, editing accuracy under structural constraints, and overall molecular quality. Chemical validity and property values are computed using RDKit Bento et al. (2020) and Therapeutics Data Commons (TDC) Huang et al. (2021), two widely used and trusted toolkits for molecular analysis: (1) **Validity** is the proportion of generated molecules that are chemically valid, reflecting the model's ability to produce syntactically correct molecular structures. (2) **Overall Accuracy (Acc$_{all}(\tau)$) and Valid Accuracy (Acc$_{valid}(\tau)$)** jointly measure editing success under structural similarity constraints. We employ three complementary structural similarity metrics with corresponding thresholds: Tanimoto similarity (TS $\geq 0.65$) Bajusz et al. (2015), Maximum Common Substructure similarity (MCS $\geq 0.6$) Cao et al. (2008), and Graph Edit Distance (GED $\leq 4$) Gao et al. (2010). For each threshold $\tau$, Acc$_{all}(\tau)$ is the percentage of all outputs that satisfy both the desired property changes and structural similarity constraints; Acc$_{valid}(\tau)$ restricts this to valid molecules only. This multi-metric approach provides comprehensive evaluation of structure preservation: TS captures fingerprint-based similarity, MCS quantifies shared molecular scaffolds, and GED measures the minimum structural editing operations required. (3) **Fréchet ChemNet Distance (FCD)** Preuer et al. (2018) quantifies the distributional distance between generated and reference molecules. Lower FCD values indicate better alignment in chemical space, capturing both diversity and realism.

## 4.3 EXPERIMENTAL RESULTS

We compare MolEditRL against publicly released, large-scaletrained molecule-editing models, evaluated in their released form; baseline details are provided in Appendix A. Table 1 compares the performance of various molecular editing models on single-property and multi-property tasks using our comprehensive multi-metric evaluation framework. MolEditRL consistently achieves the highest editing accuracy across all tasks and structural similarity metrics (TS$\geq$0.65, MCS$\geq$0.6, GED$\leq$4). Although SELFIES-based models (BioT5, MolGen) guarantee perfect chemical validity, they fail to maintain structural similarity, achieving zero accuracy on most metrics, which reflects a lack of structural alignment. DrugAssist, based on SMILES and LLM fine-tuning, maintains high validity but performs significantly worse than MolEditRL on both Accall and Accvalid. This indicates that chemical correctness alone is insufficient for precise, property-aligned editing. Although DrugAssist generates valid molecules, it struggles to retain scaffold similarity while optimizing properties. All baseline models yield substantially higher FCD scores than MolEditRL, suggesting greater divergence from real molecule distributions. In contrast, MolEditRL generates molecules that are both valid and distributionally faithful, benefiting from structure-aware graph editing. For multi-property tasks, we evaluate scenarios aligned with real-world drug discovery objectives, such as improving

Table 2: Generalization to unseen properties. Results on editing three held-out molecular properties (BBBP, HIA, hERG) that are excluded from the pretraining dataset.

| Model | Task | Validity | Acc_all (0.65) | Acc_valid (0.65) | Acc_all (0.15) | Acc_valid (0.15) | FCD | Task | Validity | Acc_all (0.65) | Acc_valid (0.65) | Acc_all (0.15) | Acc_valid (0.15) | FCD |
|---|---|---|---|---|---|---|---|---|---|---|---|---|---|---|
| BioT5 | | 1.0 | 0.0 | 0.0 | 0.276 | 0.276 | 25.7031 | | 1.0 | 0.0 | 0.0 | 0.452 | 0.452 | 21.4869 |
| DrugAssist | | 0.9719 | 0.3066 | 0.3155 | 0.3727 | 0.3835 | 7.4848 | | 0.9879 | 0.3952 | 0.4 | 0.5423 | 0.549 | 7.9316 |
| GeLLM³O_M | BBBP↓ | 0.9 | 0.17 | 0.1889 | 0.33 | 0.3667 | 7.8632 | BBBP↑ | 0.908 | 0.06 | 0.0661 | 0.356 | 0.3921 | 9.4042 |
| GeLLM³O_L | | 0.92 | 0.116 | 0.1261 | 0.292 | 0.3174 | 7.8038 | | 0.91 | 0.254 | 0.2791 | 0.716 | 0.7868 | 7.2142 |
| MolEditRL | | 0.944 | **0.326** | **0.337** | **0.516** | **0.5347** | **7.0901** | | 0.954 | **0.409** | **0.418** | **0.782** | **0.8075** | **6.7043** |
| BioT5 | | 1.0 | 0.0 | 0.0 | 0.426 | 0.426 | 15.246 | | 1.0 | 0.0 | 0.0 | 0.356 | 0.356 | 15.7829 |
| DrugAssist | | 0.982 | 0.344 | 0.3503 | 0.408 | 0.4155 | 7.0462 | | 0.976 | 0.2725 | 0.2793 | 0.4068 | 0.4168 | 9.7314 |
| GeLLM³O_M | HIA↓ | 0.904 | 0.124 | 0.1372 | 0.286 | 0.3164 | 8.2303 | HIA↑ | 0.904 | 0.348 | 0.385 | 0.662 | 0.7323 | 7.2866 |
| GeLLM³O_L | | 0.894 | 0.134 | 0.1499 | 0.382 | 0.4273 | 8.1844 | | 0.922 | 0.222 | 0.2408 | 0.554 | 0.6009 | 7.7484 |
| MolEditRL | | 0.98 | **0.374** | **0.3816** | **0.628** | **0.6396** | **6.8726** | | 0.986 | **0.446** | **0.4523** | **0.738** | **0.7459** | **6.7928** |
| BioT5 | | 1.0 | 0.0 | 0.0 | 0.396 | 0.396 | 16.3127 | | 1.0 | 0.0 | 0.0 | 0.368 | 0.368 | 15.0537 |
| DrugAssist | | 0.9659 | 0.2992 | 0.3098 | 0.4438 | 0.4595 | 7.815 | | 0.9839 | 0.2294 | 0.2331 | 0.4064 | 0.4131 | 9.9241 |
| GeLLM³O_M | hERG↓ | 0.91 | 0.162 | 0.178 | 0.308 | 0.3385 | 7.9945 | hERG↑ | 0.89 | 0.194 | 0.218 | 0.554 | 0.6225 | 7.5913 |
| GeLLM³O_L | | 0.914 | 0.144 | 0.1575 | 0.424 | 0.4639 | 8.0866 | | 0.922 | 0.062 | 0.0672 | 0.364 | 0.3948 | 11.7363 |
| MolEditRL | | 0.986 | **0.474** | **0.4807** | **0.694** | **0.7039** | **6.0764** | | 0.972 | **0.31** | **0.3189** | **0.59** | **0.6147** | **6.8304** |

Table 3: Performance comparison on C-MuMOInstruct dataset. Bold values indicate the best performance for each metric.

| Properties | Validity | | | Total_Accuracy | | | Valid_Accuracy | | |
|---|---|---|---|---|---|---|---|---|---|
| | GeLLM⁴O-C_M | GeLLM⁴O-C_L | MolEditRL | GeLLM⁴O-C_M | GeLLM⁴O-C_L | MolEditRL | GeLLM⁴O-C_M | GeLLM⁴O-C_L | MolEditRL |
| bbbp+plogp+qed | 0.9118 | 0.9076 | **0.952** | 0.2064 | 0.2605 | **0.302** | 0.2264 | 0.287 | **0.3172** |
| erg+liver+qed | 0.934 | 0.925 | **0.952** | 0.174 | 0.25 | **0.278** | 0.1863 | 0.2703 | **0.289** |
| ampa+carc+erg+plogp | 0.934 | 0.944 | **0.954** | 0.226 | 0.208 | **0.229** | **0.242** | 0.2203 | 0.231 |
| bbbp+drd2+plogp+qed | 0.9359 | 0.9518 | **0.956** | 0.2285 | 0.3373 | **0.338** | 0.2441 | **0.3544** | 0.341 |
| drd2+hia+mutagenicity+qed | 0.924 | 0.9326 | **0.958** | 0.18 | 0.191 | **0.286** | 0.1948 | 0.2048 | **0.294** |
| carc+drd2+erg | 0.894 | 0.8043 | **0.948** | 0.138 | 0.2174 | **0.266** | 0.1544 | 0.2703 | **0.275** |
| ampa+bbbp+mutagenicity+plogp | **0.9499** | 0.9024 | 0.94 | **0.2545** | 0.2358 | 0.202 | **0.2679** | 0.2613 | 0.223 |
| bbbp+carc+mutagenicity+qed | 0.9519 | 0.8889 | **0.96** | 0.2104 | 0.1818 | **0.247** | 0.2211 | 0.2045 | **0.288** |
| bbbp+drd2+erg+qed | 0.908 | 0.8302 | **0.944** | 0.186 | 0.1887 | **0.236** | 0.2048 | 0.2273 | **0.275** |
| hia+liver+mutagenicity+plogp+qed | 0.942 | 0.9423 | **0.946** | 0.212 | **0.2212** | 0.202 | 0.2251 | **0.2347** | 0.2135 |

stability and synthesis (Haccept↓, LogP↑), balancing drug-likeness and accessibility (QED↑, SA↓), and managing conflicting constraints (Haccept↑, MW↑, QED↓). MolEditRL consistently outperforms all baselines across all similarity metrics, with the convergent high performance across TS, MCS, and GED providing comprehensive evidence of effective multi-objective optimization while preserving molecular scaffolds. Extended results on single-property and multi-property tasks are available in Appendix U and Appendix V, respectively.

## 4.4 GENERALIZATION TO UNSEEN PROPERTIES

We evaluate MolEditRL on three propertiesBBBP, HIA, and hERG inhibitionthat are entirely absent from the MolEdit-Instruct pretraining corpus. These pharmacokinetic and safety-related attributes allow us to assess the models ability to adapt to new optimization objectives not seen during pretraining. In this setup, the pretrained model is fine-tuned via reinforcement learning using property-specific oracles, without requiring any prior data or moleculeproperty pairs for these tasks. MolEditRL only receives natural-language descriptions of the unseen objectives and relies on RL fine-tuning over arbitrary molecules, simulating realistic deployment scenarios where new properties emerge after pretraining. Unlike baseline approaches, no additional pretraining or dataset construction is needed. As shown in Table 2, MolEditRL achieves the best performance across all unseen-property tasks, with the highest editing accuracy under both strict ($\tau = 0.65$) and relaxed ($\tau = 0.15$) structural similarity constraints, while maintaining high validity and the lowest FCD. These results demonstrate that MolEditRL can efficiently generalize to entirely new property objectives through task-specific reward oracleswithout retraining the model from scratch.

## 4.5 EVALUATION ON C-MuMOINSTRUCT DATASET

To further assess the generality of MolEditRL, we evaluate it on the publicly available C-MuMOInstruct dataset Dey et al. (2025a), a large instruction-tuning benchmark for controllable multi-property molecular optimization. Each task specifies which properties must increase or decrease to target thresholds while keeping others unchanged, requiring models to satisfy complex multi-objective constraints across 35 attributes simultaneously. We generate one edited molecule per instruction and measure chemical validity, total accuracy, and valid accuracy. As shown in Table 3, MolEditRL achieves competitive or superior performance compared to the much larger

GeLLM$^4$O-C models. It obtains the highest validity in 8 of 10 tasks and outperforms both base-lines in 7 tasks in terms of total accuracy, with clear gains on challenging combinations such as drd2+hia+mutagenicity+qed and carc+drd2+erg. Although larger models show slight advantages on a few tasks, the differences remain small. These results demonstrate that MolEditRL can match or exceed the performance of models with over 50Œ more parameters (7B+ vs. 125M), highlighting the effectiveness of our structure-aware diffusion and reinforcement learning framework for multi-property molecular optimization.

## 4.6 MULTI-PROPERTY EDITING PERFORMANCE

To assess model robustness under increasing task complexity, we evaluate performance on molecular editing tasks involving 1, 2, or 3 simultaneous property changes. For each setting, 10 combinations of editing objectives are randomly sampled, and the results are averaged. Figure 3(a) shows mean chemical validity, while (b) presents mean editing accuracy using TS similarity threshold of 0.15. As expected, accuracy drops for all models as the number of target properties increases, reflecting the challenge of jointly satisfying multiple constraints while preserving molecular structure. Some mod-els maintain high validity but suffer from very low accuracy, indicating that generating chemically plausible molecules alone is insufficient for precise, property-aligned edits. MolEditRL consistently outperforms all baselines across all settings. In the most difficult 3-property scenario, it achieves an average accuracy of 0.363, more than double the second-best baseline (DrugAssist, 0.165). These results demonstrate the effectiveness of our structure-aware diffusion framework and reinforcement learning fine-tuning in enabling scalable and precise instruction-based molecular editing.

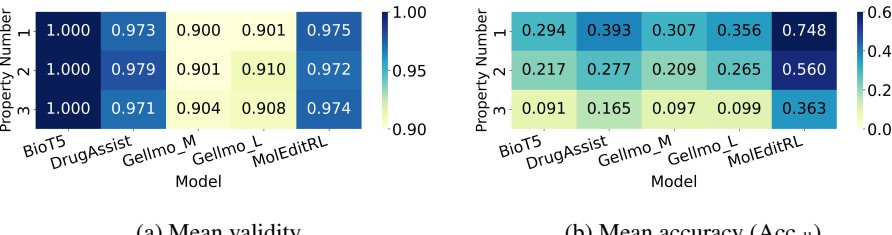

(a) Mean validity.  (b) Mean accuracy ($Acc_{all}$).

Figure 3: Performance by number of edited properties.

## 4.7 EFFECT OF FINE-TUNING STRATEGIES AND KL REGULARIZATION

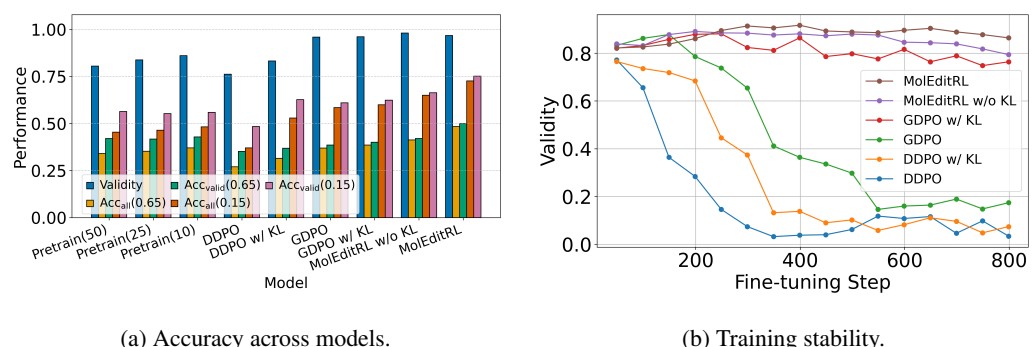

(a) Accuracy across models.  (b) Training stability.

Figure 4: Impact of step size, fine-tuning strategy, and KL regularization.

We conduct ablation studies on denoising step size, RL fine-tuning strategies, and KL regularization (Figure 4). Results are averaged over 20 single-property editing tasks. Subfigure (a) shows accuracy and validity; (b) shows training stability. "Pretrain($x$)" denotes models without RL fine-tuning, where $x \in \{50, 25, 10\}$ is the denoising step size. Smaller $x$ improves accuracy but increases computational cost. We use $t_s = 50$ as the policy update stride for efficiency. We compare two RL strategies: DDPO applies REINFORCE independently at each denoising step, while GDPO leverages $x_0$-parameterization to optimize only the final output. DDPO performs joint fine-tuning across all tasks but suffers from instability since intermediate molecules are chemically meaningless. GDPO improves stability but requires separate models for each task, limiting scalability. Neither method enforces structural constraints during fine-tuning. MolEditRL introduces KL-regularized

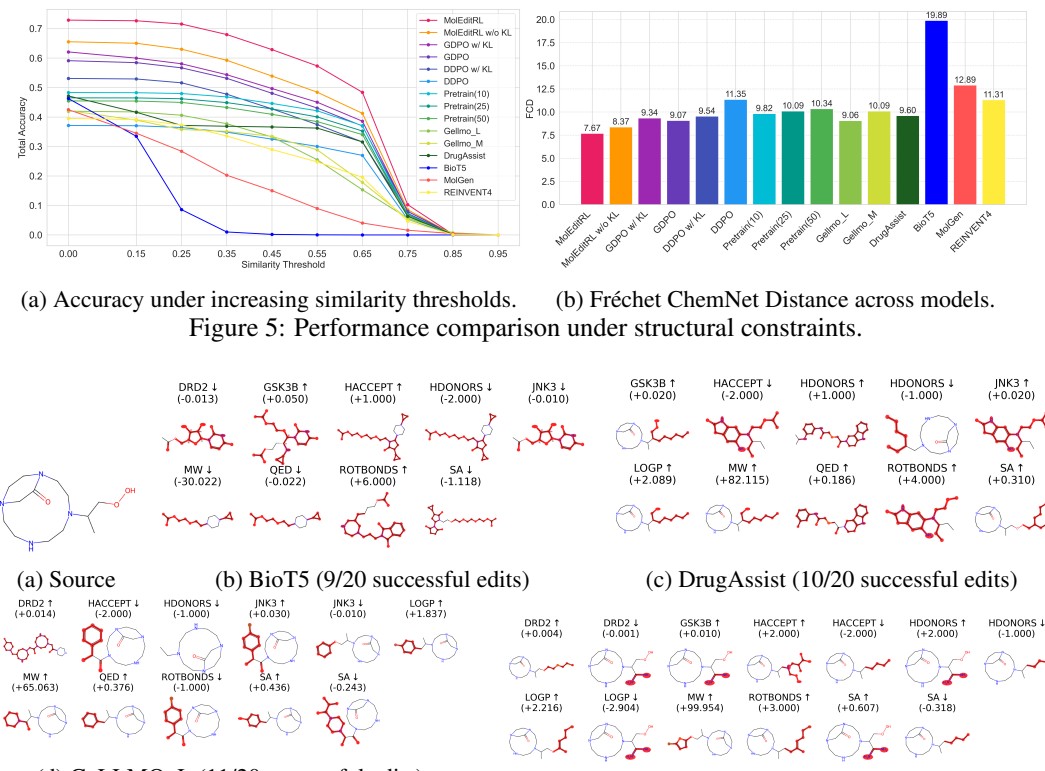

(a) Accuracy under increasing similarity thresholds.    (b) Fréchet ChemNet Distance across models.

Figure 5: Performance comparison under structural constraints.

(a) Source    (b) BioT5 (9/20 successful edits)    (c) DrugAssist (10/20 successful edits)

(d) GeLLMO_L (11/20 successful edits)    (e) MolEditRL (13/20 successful edits)

Figure 6: Visualization of edits. Red highlights indicate structural changes from the source.

optimization over the entire diffusion process, enabling stable, structure-aware fine-tuning across diverse tasks. It consistently achieves higher accuracy and validity than both alternatives. Figure 4(b) shows DDPO exhibits rapid validity degradation regardless of KL regularization, indicating inherent instability in step-wise optimization. GDPO shows improved stability with KL regularization but plateaus below MolEditRL's performance due to task-specific limitations.

### 4.8    STRUCTURE FIDELITY AND DISTRIBUTIONAL QUALITY

Figure 5(a) shows $Acc_{all}$ across different Tanimoto similarity thresholds. MolEditRL consistently achieves the highest accuracy at all thresholds, demonstrating its ability to generate molecules that satisfy desired property changes while preserving structural similarity. In contrast, LLM-based baselines such as BioT5 and MolGen perform substantially worse, especially under stricter similarity constraints. Figure 5(b) reports Fréchet ChemNet Distance (FCD) at a fixed threshold of 0.15. Lower FCD indicates better alignment between the distributions of generated and real molecules.

### 4.9    QUALITATIVE ANALYSIS OF MOLECULAR EDITING

We visualize successful molecular modifications from four representative models across 20 single-property tasks using the same source molecule. As shown in Figure 6, MolEditRL achieves the highest task success rate and is the only model that consistently preserves the core scaffold across all edits, demonstrating strong structural controllability.

## 5    CONCLUSION

We introduce MolEditRL, a novel framework that integrates discrete graph diffusion with reinforcement learning to enable precise, structure-preserving molecular edits. It achieves state-of-the-art performance on the MolEdit-Instruct benchmark while using significantly fewer parameters.

## ACKNOWLEDGMENTS

This work was supported by the National Natural Science Foundation of China (Grant Nos. 62306255, 92370204, and 62406141), the National Key Research and Development Program of China (Grant No. 2023YFF0725000), the China Postdoctoral Science Foundation (Grant Nos. GZC20252740 and 2025M784335), the Guangdong Basic and Applied Basic Research Foundation (Grant No. 2024A1515011839), and the Education Bureau of Guangzhou Municipality.

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

APPENDIX

This appendix provides extended technical and experimental details that support the findings in the main paper. It is organized as follows:

(1) Appendix A: **Baseline methods** (BioT5, DrugAssist, GeLLMO, MolGen, REINVENT 4, MoleculeSTM).

(2) Appendix B: **Model architecture** (RoBERTa, embeddings, diffusion modules).

(3) Appendix C: **Training setup** (hyperparameters, optimization, hardware).

(4) Appendix D: **KL regularization weight** ablation ($\beta$ values).

(5) Appendix E: **Policy-update stride** ablation (update frequency).

(6) Appendix F: **Top-k sampling** ablation (sampling diversity).

(7) Appendix G: **Partial-success reward** ablation (reward values).

(8) Appendix H: **Structure-aware attention** ablation (graph-level constraints).

(9) Appendix I: **Pretrain vs. RL fine-tuning** comparison.

(10) Appendix J: **Chemical realism metrics** vs. RL baselines (synthesizability, drug-likeness).

(11) Appendix K: **Prompt sensitivity** analysis (instruction robustness).

(12) Appendix L: **Complex localized editing** (fine-grained operations).

(13) Appendix M: **Noisy oracle robustness** (estimation errors).

(14) Appendix N: **Oracle-query efficiency** (limited oracle budgets).

(15) Appendix O: **Inference efficiency** (denoising steps vs. runtime).

(16) Appendix P: **Computational efficiency** (accuracy vs. cost).

(17) Appendix Q: **Gradient derivation** ($x_0$-parameterization).

(18) Appendix R: **Dataset statistics** (property ranges, prompts).

(19) Appendix S: **Limitations and future work**.

(20) Appendix T: **LLM usage statement**.

(21) Appendix U: **Extended single-property results** (10 tasks).

(22) Appendix V: **Extended multi-property results** (2-4 constraints).

(23) Appendix Y: **Qualitative visualizations** (structural modifications).

## A  BASELINES

(1) BioT5 Pei et al. (2023) leverages SELFIES and a T5-style architecture for cross-modal learning between molecules and text. (2) DrugAssist Ye et al. (2025) is a Llama2-7B-based dialogue model for interactive molecule optimization. (3) GeLLM$^3$O Dey et al. (2025b) uses instruction tuning on Mistral and Llama3 models for multi-property optimization; we evaluate both GeLLM$^3$O_M and GeLLM$^3$O_L. (4) MolGen Fang et al. (2024) is a domain-agnostic language model trained with chemical feedback to reduce invalid generations. (5) REINVENT 4 Loeffler et al. (2024) integrates reinforcement learning, transfer learning, and curriculum learning for molecular design using RNN and Transformer backbones. (6) MoleculeSTM Liu et al. (2023) is a multi-modal molecule structuretext model trained on large structuretext pairs to enable zero-shot text-guided retrieval and editing of molecules. (7) GeLLM$^4$O-C Dey et al. (2025a) is an instruction-tuned LLM on the C-MuMOInstruct dataset, built on Mistral-7B and Llama3 models.

## B  MODEL ARCHITECTURE

Table 4 details the structure-aware diffusion model used in MolEditRL. The architecture includes token and edge embeddings, a RoBERTa-based transformer with graph-aware attention, a discrete diffusion module for masked denoising, and a reinforcement learning component guided by property-based rewards. During inference, we employ top-$k$ sampling and apply policy updates at fixed stride intervals to improve efficiency.

Table 4: Technical specifications of the structure-aware diffusion model in MolEditRL.

| Module Types | Dimensions | Structures |
|---|---|---|
| Input Layer | – | Source Tokens [batch, seq_len] $\rightarrow$ Concat |
| Embedding | Vocab Size = 51,933 | TokenEmbedding [seq_len, 768] + PositionEmbedding [seq_len, 768] |
| Edge Embedding | Edge Types = 6 | EdgeEmbedding [nodes, nodes, 768] |
| RoBERTa | 12 Layers | Input [batch, seq_len, 768] $\downarrow$ Self-Attention (12 Œ 64) $\rightarrow$ LayerNorm + Residual $\downarrow$ FFN (7683072768) $\rightarrow$ LayerNorm + Residual |
| Diffusion | 2000 steps | Forward: Input $\rightarrow$ Masked Tokens $\downarrow$ Reverse (stride = 50) $\downarrow$ Denoising Network |
| Prediction | seq_len Œ 51,933 nodes Œ nodes Œ 6 | AtomLogits [batch, seq_len, 51933] EdgeLogits [batch, nodes, nodes, 6] |
| Sampling | Top-k = 15 | Atom Categorical Sampling Edge Structure Sampling |
| Property-Guided | – | Reward Calculation (0, 0.2, 1.0) Advantage Function $\rightarrow$ Loss Weighting |

## C  TRAINING AND HYPERPARAMETER SETUP

We train MolEditRL on a multi-GPU cluster using PyTorch with Distributed Data Parallel (DDP). The model is initialized from a RoBERTa-base encoder with 12 layers, 12 heads, and hidden size 768. The tokenizer is extended to 51,933 tokens to accommodate molecular and instruction-specific vocabulary, and the embedding layer is resized accordingly. For optimization, we use the AdamW optimizer with a learning rate of 5e-5, weight decay of 0.01, and a linear warm-up scheduler over 10,000 steps. Mixed precision (FP16) is enabled to reduce memory usage and accelerate training. During pretraining, the model is trained for 100 epochs with a per-GPU batch size of 16. A discrete diffusion schedule with 2,000 denoising steps is used, following a mutual noise schedule $\beta_t = 1/(T-t)$ where $T = 2000$. We apply word- and edge-level frequency weighting with sinusoidal modulation ($\lambda = 0.3$) to guide denoising dynamics. The edge vocabulary includes 6 bond types. During reinforcement learning fine-tuning, rewards are computed using property oracles (e.g., RDKit, TDC). A key advantage of MolEditRL is its remarkable oracle efficiency. The pretrained model achieves strong property optimization performance even without any oracle calls during inference, already outperforming most state-of-the-art baselines as demonstrated in our ablation studies. This efficiency stems from our editing-based formulation, which performs structure-constrained, localized modifications starting from known molecules. This approach drastically reduces the chemical search space and required oracle queries by orders of magnitude compared to de novo generation methods. Our empirical results confirm this efficiency under strict oracle budgets, where MolEditRL's performance quickly saturates with most improvements achieved within just 6,400 oracle queries during the standard 400-step fine-tuning protocol. We use top-$k$ sampling with $k = 15$ and a temperature of 1.0 during evaluation. To improve efficiency, the policy is updated every $t_s = 50$ steps, resulting in 40 updates over the 2000-step diffusion process. For consistency, inference also runs for 40 denoising steps, starting from a fully masked graph and progressively reconstructing the final molecule. All

experiments were conducted on a single NVIDIA A6000 GPU using PyTorch and DGL. Pretraining on the MolEdit-Instruct dataset (3M examples) took approximately 100 hours. RL fine-tuning for each task required 12 hours.

# D    ABLATION STUDY ON KL REGULARIZATION WEIGHT

Table 5 reports an ablation study on the KL-regularization weight $\beta$, evaluating models trained with $\beta \in \{0, 0.1, 0.2, 0.3, 0.4, 0.5\}$ under a fixed fine-tuning budget of 500 steps. The results highlight the importance of balancing structural preservation and reward-driven optimization. When $\beta = 0$, the policy is no longer anchored to the pretrained diffusion prior, resulting in unstable behavior and overly aggressive edits that harm structural similarity. Conversely, larger $\beta$ values impose excessive regularization, restricting the policys ability to improve the target property and increasing FCD. Across both LogP and SA tasks, moderate regularization consistently yields the best trade-off between validity, similarity-constrained accuracy, and distributional quality. Based on these trends, we adopt $\beta = 0.1$ as the default setting in all experiments, as it provides stable structure-preserving updates while enabling effective property optimization.

Table 5: Ablation study on KL regularization weight $\beta$

| Task | $\beta$ | Validity | $\text{Acc}_{all}$ (TS$\geq$0.65) | $\text{Acc}_{all}$ (MCS$\geq$0.6) | $\text{Acc}_{all}$ (GED$\leq$4) | FCD$\downarrow$ |
|---|---|---|---|---|---|---|
| LogP $\uparrow$ | 0.0 | **0.986** | 0.278 | 0.376 | 0.186 | 9.142 |
| | 0.1 | 0.976 | **0.462** | **0.498** | **0.214** | **7.812** |
| | 0.2 | 0.920 | 0.416 | 0.458 | 0.196 | 9.762 |
| | 0.3 | 0.884 | 0.386 | 0.414 | 0.206 | 9.649 |
| | 0.4 | 0.840 | 0.350 | 0.402 | 0.210 | 10.812 |
| | 0.5 | 0.842 | 0.356 | 0.380 | 0.194 | 10.700 |
| SA $\downarrow$ | 0.0 | 0.982 | 0.602 | **0.700** | 0.208 | 7.295 |
| | 0.1 | **0.988** | **0.608** | 0.680 | **0.258** | **6.735** |
| | 0.2 | 0.942 | 0.576 | 0.582 | 0.186 | 8.503 |
| | 0.3 | 0.956 | 0.574 | 0.576 | 0.192 | 8.679 |
| | 0.4 | 0.928 | 0.562 | 0.558 | 0.186 | 8.900 |
| | 0.5 | 0.918 | 0.544 | 0.548 | 0.194 | 9.100 |

# E    ABLATION STUDY ON POLICY-UPDATE STRIDE

Table 6 presents a sensitivity analysis of the policy-update stride, comparing stride values in $\{1, 2, 3, 4, 5\}$ under an identical fine-tuning budget of 500 steps. Each table entry reports Validity / $\text{Acc}_{all}$(TS $\geq 0.65$) / FCD, enabling joint assessment of chemical correctness, structural-similarityconstrained accuracy, and distributional fidelity. The results show that stride = 1 consistently achieves the best performance across both LogP$\uparrow$ and SA$\downarrow$ tasks, providing the highest accuracy and lowest FCD throughout the training trajectory. Increasing the stride reduces the frequency of policy updates, which slows optimization progress and leads to noticeable degradation in both property alignment and structural quality. These findings demonstrate that frequent policy updates are essential for stable and effective reinforcement learning in the discrete diffusion setting, and we therefore adopt stride = 1 as the default configuration for all experiments.

# F    ABLATION STUDY ON TOP-K SAMPLING

Table 7 analyzes the effect of the top-k sampling parameter by evaluating settings from k = 5 to k = 25 under otherwise identical inference conditions. Across both the LogP and SA tasks, model performance remains largely stable, with only mild fluctuations in validity, structural-similarity accuracy, and FCD. Smaller top-k values can restrict sampling diversity and slightly reduce structural flexi-

Table 6: Sensitivity analysis of policy-update stride

| Task | Step | Stride=1 | Stride=2 | Stride=3 | Stride=4 | Stride=5 |
|------|------|----------|----------|----------|----------|----------|
| LogP ↑ | 99 | 0.910/0.436/8.425 | 0.816/0.326/11.164 | 0.802/0.334/11.037 | 0.776/0.332/10.998 | 0.780/0.290/11.597 |
| | 199 | 0.933/0.442/8.076 | 0.842/0.318/11.159 | 0.834/0.358/10.430 | 0.796/0.346/11.031 | 0.824/0.344/10.494 |
| | 299 | 0.958/0.438/8.294 | 0.860/0.370/10.569 | 0.866/0.368/9.431 | 0.838/0.372/9.901 | 0.828/0.348/10.499 |
| | 399 | 0.956/0.456/7.965 | 0.914/0.364/9.844 | 0.862/0.356/10.451 | 0.866/0.370/9.950 | 0.860/0.358/10.736 |
| | 499 | **0.976/0.462/7.812** | 0.938/0.408/9.095 | 0.898/0.382/9.884 | 0.866/0.372/9.803 | 0.888/0.366/10.348 |
| SA ↓ | 99 | 0.984/0.587/7.466 | 0.848/0.526/9.482 | 0.844/0.516/9.466 | 0.780/0.430/11.379 | 0.788/0.436/10.665 |
| | 199 | 0.982/0.580/7.510 | 0.872/0.536/9.378 | 0.876/0.522/9.474 | 0.818/0.456/10.692 | 0.834/0.486/10.459 |
| | 299 | 0.978/0.591/7.322 | 0.916/0.556/8.818 | 0.904/0.550/9.191 | 0.862/0.480/10.197 | 0.896/0.548/8.881 |
| | 399 | 0.971/0.599/7.251 | 0.946/0.598/8.333 | 0.910/0.558/8.757 | 0.904/0.504/9.925 | 0.904/0.558/8.636 |
| | 499 | **0.988/0.608/6.735** | 0.956/0.592/8.202 | 0.932/0.512/9.737 | 0.924/0.530/9.523 | 0.920/0.574/8.904 |

bility, while excessively large values introduce unnecessary stochasticity that may weaken property alignment. Overall, moderate top-k values achieve the best balance between diversity and reliability. Based on the observed trends, we adopt top-k = 15 as the default sampling configuration in all experiments.

Table 7: Sensitivity analysis of top-k sampling parameter

| Task | Top-k | Validity | $\text{Acc}_{\text{all}}$ (TS≥0.65) | $\text{Acc}_{\text{all}}$ (MCS≥0.6) | $\text{Acc}_{\text{all}}$ (GED≤4) | FCD↓ |
|------|-------|----------|------------------------|------------------------|-----------------------|------|
| LogP ↑ | 5 | 0.956 | 0.460 | 0.462 | 0.208 | 7.924 |
| | 10 | 0.950 | 0.458 | 0.438 | 0.202 | 7.977 |
| | 15 | **0.976** | **0.462** | **0.498** | 0.214 | **7.812** |
| | 20 | 0.946 | 0.448 | 0.490 | 0.216 | 8.110 |
| | 25 | 0.940 | 0.456 | 0.492 | **0.224** | 8.082 |
| SA ↓ | 5 | 0.982 | 0.600 | 0.642 | 0.232 | 7.043 |
| | 10 | **0.990** | 0.578 | 0.564 | 0.202 | 7.192 |
| | 15 | 0.988 | 0.608 | **0.690** | **0.258** | 6.735 |
| | 20 | 0.980 | **0.638** | 0.684 | 0.252 | **6.657** |
| | 25 | 0.990 | 0.616 | 0.671 | 0.242 | 6.921 |

## G   ABLATION STUDY ON PARTIAL-SUCCESS REWARD

Table 8 presents an ablation study on the partial-success reward, evaluated using values in {0, 0.2, 0.4, 0.6, 0.8, 1.0}. This reward is assigned to molecules that are chemically valid but fail to satisfy the editing objective, allowing the model to differentiate between invalid outputs and structurally plausible but suboptimal edits. The results show that removing this reward entirely (0.0) leads to unstable optimization and decreased validity, as the model receives no guidance for valid-but-incorrect molecules. Conversely, overly large partial-success rewards (≥ 0.6) diminish the incentive to complete the desired edit, resulting in lower similarity-constrained accuracy and higher FCD. Moderate values in the range 0.20.4 provide the most effective balance between stability, structural fidelity, and property optimization. Based on these observations, we adopt 0.2 as the default partial-success reward in all experiments.

## H   ABLATION STUDY ON STRUCTURE-AWARE ATTENTION MECHANISM

Table 9 presents an ablation study evaluating the structure-aware attention bias during both the pretraining and fine-tuning stages. Including the bias during pretraining yields modest gains in structural-similarity accuracy and FCD, reflecting its role as a helpful but not dominant inductive prior when the model is learning general molecular distributions. However, the benefit becomes significantly more pronounced after reinforcement learning fine-tuning. Models equipped with the structure-aware bias during fine-tuning achieve substantially higher accuracy across all similarity-constrained metrics and markedly lower FCD for both LogP and SA tasks. These results indicate

Table 8: Sensitivity analysis of partial-success reward values

| Task | Partial Reward | Validity | Acc$_{all}$ (TS$\geq$0.65) | Acc$_{all}$ (MCS$\geq$0.6) | Acc$_{all}$ (GED$\leq$4) | FCD$\downarrow$ |
|------|----------------|----------|------|------|------|------|
| LogP $\uparrow$ | 0.0 | 0.952 | **0.484** | 0.508 | 0.202 | 7.906 |
|      | 0.2 | **0.976** | 0.462 | 0.498 | 0.214 | **7.812** |
|      | 0.4 | 0.962 | 0.464 | **0.528** | **0.218** | 7.859 |
|      | 0.6 | 0.958 | 0.480 | 0.504 | 0.202 | 8.234 |
|      | 0.8 | 0.964 | 0.434 | 0.468 | 0.208 | 8.890 |
|      | 1.0 | 0.964 | 0.402 | 0.438 | 0.216 | 9.458 |
| SA $\downarrow$ | 0.0 | 0.976 | 0.600 | 0.634 | 0.234 | 7.110 |
|      | 0.2 | **0.988** | 0.608 | **0.680** | **0.258** | **6.735** |
|      | 0.4 | 0.984 | **0.626** | 0.656 | 0.220 | 7.142 |
|      | 0.6 | 0.980 | 0.542 | 0.582 | 0.208 | 7.773 |
|      | 0.8 | 0.984 | 0.590 | 0.594 | 0.212 | 7.998 |
|      | 1.0 | 0.984 | 0.530 | 0.556 | 0.196 | 8.530 |

that while structural bias provides useful guidance during pretraining, its primary impact emerges during RL optimization, where explicit graph-level constraints help the model perform chemically valid, topology-preserving edits and avoid drifting away from realistic molecular structures.

Table 9: Ablation Study on Structure-Aware Attention Mechanism Across Pretraining and Fine-tuning Stages

| Task | Setting | Validity | Acc$_{all}$ (TS$\geq$0.65) | Acc$_{all}$ (MCS$\geq$0.6) | Acc$_{all}$ (GED$\leq$4) | FCD$\downarrow$ |
|------|---------|----------|------|------|------|------|
| LogP $\uparrow$ | Pretrain w/o Structure Bias | 0.744 | 0.176 | 0.208 | 0.182 | 13.714 |
|      | Pretrain w/ Structure Bias | 0.758 | 0.316 | 0.232 | 0.196 | 11.896 |
|      | Finetune w/o Structure Bias | 0.890 | 0.212 | 0.266 | 0.213 | 12.486 |
|      | Finetune w/ Structure Bias | **0.976** | **0.462** | **0.498** | **0.218** | **7.812** |
| SA $\downarrow$ | Pretrain w/o Structure Bias | 0.836 | 0.196 | 0.224 | 0.128 | 12.364 |
|      | Pretrain w/ Structure Bias | 0.842 | 0.213 | 0.256 | 0.195 | 10.522 |
|      | Finetune w/o Structure Bias | 0.904 | 0.412 | 0.468 | 0.176 | 8.452 |
|      | Finetune w/ Structure Bias | **0.988** | **0.608** | **0.680** | **0.258** | **6.735** |

# I COMPARISON OF PRETRAINED DIFFUSION AND RL FINE-TUNED MODELS

Table 10 presents a detailed comparison between the pretrained diffusion model and the RL fine-tuned version of MolEditRL across five representative editing tasks. The results show that the diffusion model alone already achieves strong structural fidelity, as reflected by the consistently high MACCS_FTS, RDK_FTS, and Morgan_FTS scores, indicating that pretraining successfully learns a stable and realistic structural prior. After RL fine-tuning, these structural similarity metrics remain largely unchanged, demonstrating that the KL-regularized optimization preserves the learned molecular topology instead of distorting it. In contrast, RL fine-tuning brings substantial improvements in validity and property-aligned accuracy, and consistently reduces FCD across all tasks, confirming that the edited molecular distribution becomes closer to real molecules while more effectively satisfying target properties. Overall, Table 10 highlights the complementary nature of the two stages: diffusion pretraining establishes a reliable structure-aware foundation, and RL fine-tuning delivers targeted property optimization without compromising structural integrity.

# J COMPARISON WITH RL-BASED METHODS ON CHEMICAL REALISM METRICS

Figure 7 provides a comprehensive evaluation of the distributional shifts in key physicochemical properties between the source molecules (Input) and the molecules optimized by MolEditRL (Output). The green shaded regions delineate the ideal ranges for drug-like compounds according to medicinal chemistry standards (e.g., Lipinski's Rule of Five). The results demonstrate that MolEd-

Table 10: Performance comparison between pretrained diffusion model and RL fine-tuned model

| Task | | Validity | $\text{Acc}_{all}$(TS$\geq$0.65) | $\text{Acc}_{valid}$(TS$\geq$0.65) | MACCS_FTS | RDK_FTS | Morgan_FTS | FCD$\downarrow$ |
|---|---|---|---|---|---|---|---|---|
| QED$\uparrow$ | Pretrain | 0.812 | 0.460 | 0.567 | 0.832 | **0.736** | **0.684** | 10.518 |
| | Finetune | **0.974** | **0.604** | **0.620** | **0.834** | 0.735 | 0.667 | **7.678** |
| Haccept$\uparrow$ | Pretrain | 0.750 | 0.266 | 0.355 | 0.789 | 0.688 | 0.609 | 9.489 |
| | Finetune | **0.968** | **0.484** | **0.500** | **0.798** | **0.691** | **0.623** | **7.316** |
| LogP$\uparrow$ | Pretrain | 0.758 | 0.316 | 0.417 | 0.776 | 0.672 | **0.643** | 11.896 |
| | Finetune | **0.964** | **0.578** | **0.599** | **0.795** | **0.682** | 0.620 | **7.012** |
| DRD2$\uparrow$ | Pretrain | 0.850 | 0.220 | 0.259 | **0.796** | **0.698** | **0.637** | 11.194 |
| | Finetune | **0.966** | **0.308** | **0.319** | 0.791 | 0.677 | 0.629 | **9.389** |
| MW$\uparrow$ | Pretrain | 0.774 | 0.142 | 0.184 | 0.783 | 0.651 | 0.562 | 10.890 |
| | Finetune | **0.960** | **0.404** | **0.421** | **0.805** | **0.673** | **0.576** | **6.588** |

itRL does not merely preserve the validity of the source molecules but actively optimizes their pharmacological quality. Specifically, the Quantitative Estimate of Drug-likeness (QED) shows a substantial improvement, with the mean value increasing from 0.44 to 0.59, shifting the distribution significantly into the highly desirable range ($> 0.5$). Similarly, the Synthetic Accessibility (SA) score decreases from 3.48 to 3.16, indicating that the generated molecules are chemically easier to synthesize. Furthermore, fundamental properties such as Molecular Weight (MW) and LogP shift towards more favorable, central values within the ideal windows, avoiding the property drift often observed in generative models. Most notably, the compliance ratio with Lipinski's Rule of 5 improves dramatically from 52.7% in the source molecules to 84.7% in the output, underscoring MolEditRL's ability to generate structures that are not only target-optimized but also highly realistic, stable, and developable.

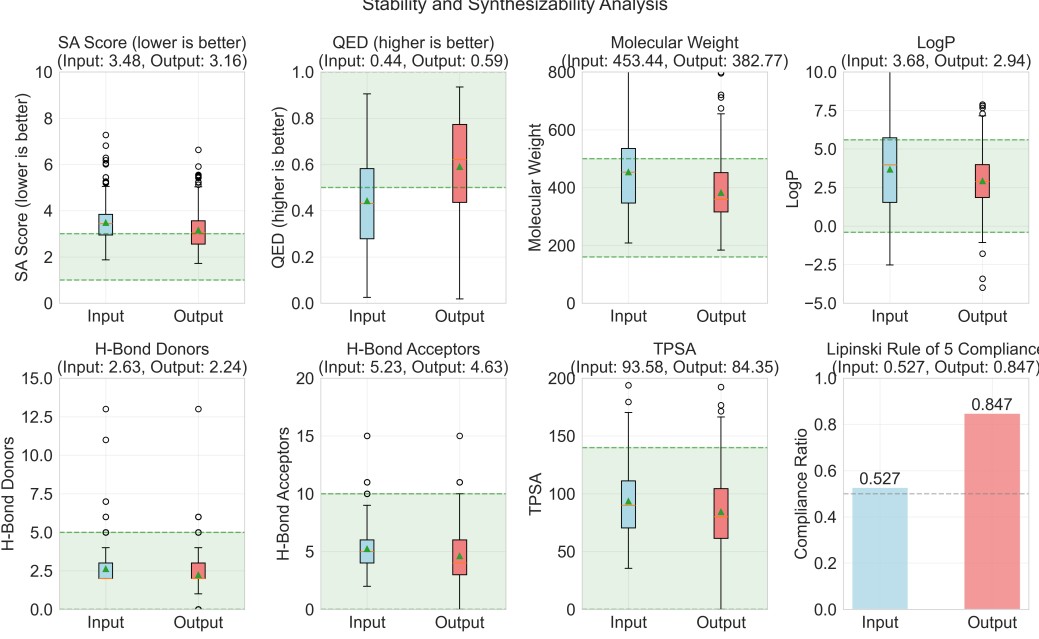

Figure 7: **Stability and Synthesizability Analysis.** Comparison of property distributions between source molecules (Input) and MolEditRL-generated outputs (Output). The green shaded areas represent ideal value ranges for drug-like candidates.

Table 11 compares MolEditRL with three representative RL-based molecular optimization frameworksGCPN You et al. (2018), MolDQN Zhou et al. (2019), and REINVENT 4 Loeffler et al. (2024)under a strict oracle budget of 5,000 queries on the HIA$\uparrow$ task, which is fully held out from pretraining. Across all chemical realism metrics, including synthesizability, drug-likeness, and Lipinski compliance, MolEditRL significantly outperforms existing RL methods and even improves beyond the source molecules themselves. While traditional RL approaches tend to suffer from distributional drift and generate chemically implausible structures when optimizing unseen properties, MolEditRL maintains high validity, low FCD, and superior realism due to its KL-regularized objective. By an-

choring policy updates to the pretrained diffusion prior, the model avoids degenerate exploration and consistently produces realistic, synthesizable, and pharmacologically relevant molecules.

Table 11: Comparison with RL-based methods on chemical realism metrics

| Method | Validity | $Acc_{all}$ (TS$\geq$0.65) | FCD$\downarrow$ | Is_Synthesizable | Is_Druglike | Lipinski_RO5 |
|---|---|---|---|---|---|---|
| Source Molecule | 1.000 | - | - | 0.278 | 0.376 | 0.527 |
| GCPN | 0.858 | 0.000 | 20.260 | 0.047 | 0.153 | 0.205 |
| MolDQN | **1.000** | 0.003 | 13.152 | 0.092 | 0.178 | 0.246 |
| Reinvent4 | 0.835 | 0.124 | 13.786 | 0.278 | 0.370 | 0.428 |
| MolEditRL | 0.958 | **0.466** | **7.964** | **0.460** | **0.681** | **0.847** |

## K  PROMPT SENSITIVITY ANALYSIS

Table 12 evaluates the prompt robustness of MolEditRL on the SA task by testing five distinct paraphrased natural-language instructions. Although Table 19 presents only one representative template, the MolEdit-Instruct pretraining corpus contains many alternative linguistic formulations for each editing objective, exposing the model to broad variability in syntax, vocabulary, and semantic emphasis. To explicitly assess the effect of such variation, we evaluate the following five prompts: (1) P1: Reduce the synthetic accessibility of molecule SMILE. (2) P2: Make this molecule SMILE easier to synthesize. (3) P3: Adjust the structure of SMILE to lower its synthetic complexity. (4) P4: Modify SMILE so that its overall synthetic accessibility score decreases. (5) P5: Transform the molecule SMILE into a form that is simpler to assemble synthetically. The results show that MolEditRL sustains consistently high validity, structural similarity accuracy, and competitive FCD scores across all prompt variants, with only minor performance fluctuations. This indicates that the model does not rely on any specific phrasing pattern; instead, it benefits from the diverse paraphrasing present during pretraining, enabling a prompt-invariant and semantically robust understanding of user instructions.

Table 12: Prompt sensitivity analysis on SA$\downarrow$ task with diverse instruction formulations

| Prompt | Validity | $Acc_{all}$(TS$\geq$0.65) | $Acc_{all}$(MCS$\geq$0.6) | $Acc_{all}$(GED$\leq$4) | FCD$\downarrow$ |
|---|---|---|---|---|---|
| P1 | **0.986** | 0.636 | **0.696** | 0.244 | 7.107 |
| P2 | 0.974 | 0.604 | 0.618 | 0.253 | 7.307 |
| P3 | 0.979 | **0.652** | 0.636 | 0.224 | 7.157 |
| P4 | 0.984 | 0.629 | 0.614 | 0.244 | **7.052** |
| P5 | 0.980 | 0.646 | 0.642 | **0.256** | 7.339 |

## L  COMPLEX LOCALIZED EDITING INSTRUCTIONS

To assess the flexibility of MolEditRL in interpreting complex, localized natural-language editing instructions, we fine-tuned the pretrained model for 500 steps using five structurally explicit prompts. Unlike simple property-based commands, these instructions specify concrete chemical operations such as functional group removal, fragment addition, and scaffold simplification. The five prompts used are: (1) P1: Remove a $CO_2H$ group from SMILE and decrease its H-bond donor characteristics. (2) P2: Add an additional amide fragment to SMILE to increase its molecular weight. (3) P3: Remove an aromatic ring from SMILE to lower its structural complexity and improve synthetic accessibility. (4) P4: Reduce the synthetic accessibility of molecule SMILE. (5) P5: Eliminate a CONH unit from SMILE to make the scaffold easier to assemble. Table 13 reports the results using two key metrics: FG Editing Success, which evaluates whether the required functional group operation is correctly executed, and $Acc_{all}$(TS $\geq$ 0.65), which additionally requires structural similarity and successful property alignment. Across all prompts, MolEditRL consistently achieves higher validity, substantially better functional-group editing accuracy, and more realistic molecular outputs than Reinvent4, while also maintaining strong synthesizability, drug-likeness, and Lipinski compliance. These findings demonstrate that MolEditRL can reliably interpret fine-grained chemical instructions and execute highly localized editscapabilities that cannot be captured using scalar

property targets alone. This highlights natural language as a powerful and expressive interface for precise, interpretable molecular manipulation.

Table 13: Performance on complex localized editing instructions demonstrating natural language flexibility

| Prompt | Model | Validity | FG Editing Success | Acc$_{all}$(TS≥0.65) | FCD | MACCS_FTS | Is_Synthesizable | Is_Druglike | Lipinski_RO5 |
|---|---|---|---|---|---|---|---|---|---|
| - | Source Molecule | 1.000 | - | - | - | - | 0.278 | 0.376 | 0.520 |
| P1 | Reinvent4 | 0.656 | 0.252 | 0.078 | 15.77 | 0.428 | 0.157 | 0.267 | 0.297 |
|  | MolEditRL | **0.970** | **0.810** | **0.572** | **7.01** | **0.789** | **0.396** | **0.655** | **0.787** |
| P2 | Reinvent4 | 0.512 | 0.328 | 0.102 | 14.58 | 0.387 | 0.185 | 0.252 | 0.389 |
|  | MolEditRL | **0.968** | **0.798** | **0.360** | **8.98** | **0.785** | **0.332** | **0.574** | **0.884** |
| P3 | Reinvent4 | 0.606 | 0.320 | 0.112 | 13.28 | 0.385 | 0.157 | 0.188 | 0.246 |
|  | MolEditRL | **0.994** | **0.744** | **0.358** | **7.98** | **0.784** | **0.328** | **0.505** | **0.874** |
| P4 | Reinvent4 | 0.632 | 0.340 | 0.126 | 12.86 | 0.379 | 0.104 | 0.185 | 0.283 |
|  | MolEditRL | **0.984** | **0.872** | **0.370** | **7.93** | **0.763** | **0.392** | **0.591** | **0.839** |
| P5 | Reinvent4 | 0.642 | 0.234 | 0.128 | 12.32 | 0.365 | 0.160 | 0.179 | 0.204 |
|  | MolEditRL | **0.978** | **0.752** | **0.310** | **7.60** | **0.778** | **0.327** | **0.503** | **0.712** |

## M  ROBUSTNESS TO NOISY ORACLE

Table 14 evaluates the robustness of MolEditRL when the optimization oracle is imperfect or noisya realistic scenario in molecular design where property predictors often contain estimation errors. To simulate such conditions, we inject controlled label noise by randomly flipping oracle outputs at varying rates from 0 to 0.2. Across both the LogP and SA tasks, MolEditRL maintains high validity and stable editing accuracy, with only marginal fluctuations even under the highest noise level. Importantly, structural similarity metrics and FCD remain largely unaffected, indicating that the model continues to operate within a realistic chemical distribution despite corrupted reward signals. These results show that MolEditRL does not rely on a perfect or deterministic oracle; instead, its KL-regularized optimization anchors policy updates to the pretrained diffusion prior, preventing overreaction to noisy rewards and ensuring consistent, reliable editing behavior.

Table 14: Robustness to Noisy Oracle Across Different Noise Levels

| Task | Noise Level | Validity | Acc$_{all}$ (TS≥0.65) | Acc$_{all}$ (MCS≥0.6) | Acc$_{all}$ (GED≤4) | FCD↓ |
|---|---|---|---|---|---|---|
| LogP ↑ | 0.0 | **0.976** | **0.462** | 0.498 | 0.214 | 7.812 |
|  | 0.05 | 0.920 | 0.457 | 0.508 | **0.220** | **7.529** |
|  | 0.1 | 0.944 | 0.460 | **0.514** | 0.219 | 7.582 |
|  | 0.15 | 0.924 | 0.454 | 0.488 | 0.216 | 7.925 |
|  | 0.2 | 0.914 | 0.448 | 0.472 | 0.212 | 8.215 |
| SA ↓ | 0.0 | **0.988** | 0.608 | **0.680** | **0.258** | **6.735** |
|  | 0.05 | 0.980 | **0.618** | 0.658 | 0.236 | 7.174 |
|  | 0.1 | 0.986 | 0.612 | 0.668 | 0.236 | 7.146 |
|  | 0.15 | 0.984 | 0.618 | 0.672 | 0.238 | 7.055 |
|  | 0.2 | 0.980 | 0.592 | 0.666 | 0.242 | 7.172 |

## N  ORACLE-QUERY EFFICIENCY COMPARISON

Table 15 presents a detailed comparison of MolEditRL with three classical RL-based molecular optimization frameworksGCPN, MolDQN, and REINVENT4under progressively increasing oracle-query budgets from 1,000 to 5,000. We evaluate the HIA property, which is entirely absent from pretraining, making it an ideal benchmark for understanding real-world oracle efficiency when optimizing previously unseen molecular attributes. Across all query budgets, MolEditRL consistently achieves higher editing accuracy, lower FCD, and competitive validity compared to baseline RL methods. This superior efficiency stems from the strong structure-aware prior established during discrete diffusion pretraining, allowing MolEditRL to begin reinforcement learning from an already realistic and structurally faithful distribution. In contrast, conventional RL approaches must rely on uninformed trial-and-error exploration, requiring a large number of oracle interactions to discover viable editing strategies. The consistent gains observed across all budgets highlight MolEditRLs ability to perform effective, low-cost molecular editing even when oracle access is limited.

Table 15: Oracle-query efficiency comparison on HIA↑ task across different oracle budgets

| Oracle Queries | Method | Validity | Acc$_{all}$ (TS≥0.65) | Acc$_{all}$ (MCS≥0.6) | Acc$_{all}$ (GED≤4) | FCD↓ |
|---|---|---|---|---|---|---|
| 1000 | GCPN | 0.842 | 0.000 | 0.000 | 0.026 | 22.140 |
| | MolDQN | **1.000** | 0.002 | 0.138 | 0.086 | 15.598 |
| | Reinvent4 | 0.598 | 0.092 | 0.148 | 0.044 | 14.682 |
| | MolEditRL | 0.864 | **0.388** | **0.408** | **0.202** | **9.686** |
| 2000 | GCPN | 0.882 | 0.000 | 0.000 | 0.022 | 21.175 |
| | MolDQN | **1.000** | 0.000 | 0.122 | 0.026 | 14.857 |
| | Reinvent4 | 0.608 | 0.104 | 0.167 | 0.052 | 14.236 |
| | MolEditRL | 0.898 | **0.396** | **0.448** | **0.218** | **9.489** |
| 3000 | GCPN | 0.878 | 0.000 | 0.000 | 0.016 | 21.324 |
| | MolDQN | **1.000** | 0.002 | 0.156 | 0.036 | 14.828 |
| | Reinvent4 | 0.711 | 0.114 | 0.187 | 0.080 | 13.997 |
| | MolEditRL | 0.902 | **0.430** | **0.472** | **0.216** | **8.609** |
| 4000 | GCPN | 0.850 | 0.000 | 0.000 | 0.022 | 20.672 |
| | MolDQN | **1.000** | 0.002 | 0.156 | 0.032 | 13.362 |
| | Reinvent4 | 0.769 | 0.118 | 0.192 | 0.105 | 13.868 |
| | MolEditRL | 0.928 | **0.448** | **0.488** | **0.214** | **8.387** |
| 5000 | GCPN | 0.858 | 0.000 | 0.000 | 0.029 | 20.260 |
| | MolDQN | **1.000** | 0.003 | 0.142 | 0.045 | 13.152 |
| | Reinvent4 | 0.835 | 0.124 | 0.196 | 0.139 | 13.786 |
| | MolEditRL | 0.958 | **0.466** | **0.490** | **0.226** | **7.964** |

## O    INFERENCE EFFICIENCY COMPARISON

Table 16 compares inference efficiency across baseline models and MolEditRL under varying skip-step configurations during the reverse diffusion process. Here, step refers to the stride of the denoising trajectory: with a total of 2000 diffusion steps, larger stride values (e.g., step = 500) correspond to fewer denoising updates, while smaller stride values (e.g., step = 50) yield finer-grained refinement with more update iterations. The results show that MolEditRL maintains strong accuracy and favorable distributional fidelity even under coarse schedules, achieving lower FCD than large LLM-based baselines while preserving fast per-sample inference time on an A6000 GPU. As the stride decreases, performance gradually improves with moderate increases in computation. Overall, MolEditRL demonstrates robust behavior under step skipping, delivering high-quality generations across a wide range of schedules. Based on the balance between accuracy and runtime, we adopt step = 50 as the default configuration in the main experiments.

Table 16: Inference efficiency comparison across baselines and different denoising step settings

| Method | Validity | Acc$_{all}$(TS≥0.65) | Acc$_{all}$(MCS≥0.6) | Acc$_{all}$(GED≤4) | FCD↓ | Time/Sample (s) |
|---|---|---|---|---|---|---|
| BioT5 | 1.000 | 0.000 | 0.004 | 0.000 | 13.4000 | 0.7550 |
| DrugAssist | 0.988 | 0.537 | 0.551 | 0.202 | 9.0500 | 1.5221 |
| Gellm$^4$o-C_L | 0.849 | 0.218 | 0.224 | 0.104 | 10.6700 | 4.5600 |
| MolEditRL (step=500) | 0.900 | 0.536 | 0.590 | 0.220 | 7.9273 | 1.0608 |
| (step=400) | 0.924 | 0.568 | 0.628 | 0.240 | 7.5318 | 1.0684 |
| (step=250) | 0.952 | 0.588 | 0.644 | 0.234 | 7.3480 | 1.0954 |
| (step=200) | 0.956 | 0.576 | 0.638 | 0.234 | 7.4078 | 1.1135 |
| (step=100) | 0.972 | 0.590 | 0.668 | 0.232 | 7.1624 | 1.2039 |
| **(step=50)** | **0.988** | **0.608** | **0.680** | **0.258** | **6.7350** | **1.3857** |
| (step=40) | 0.979 | 0.624 | 0.682 | 0.234 | 6.5894 | 1.4778 |
| (step=20) | 0.988 | 0.628 | 0.686 | 0.246 | 6.4828 | 1.9286 |
| (step=10) | 0.984 | 0.636 | 0.690 | 0.238 | 6.2055 | 2.8393 |

## P    COMPUTATIONAL EFFICIENCY COMPARISON

Table 17 compares MolEditRL with a range of RL-based molecular editing approaches in terms of both editing performance and computational efficiency on the HIA task, which is fully ex-

cluded from pretraining. Under a fixed 5,000-oracle budget, traditional RL methods such as GCPN, MolDQN, and REINVENT4 fail to achieve meaningful accuracy due to their reliance on costly trial-and-error exploration. In contrast, MolEditRL benefits from its pretrained structure-aware diffusion prior, enabling effective fine-tuning with far fewer oracle queries and delivering substantially higher similarity-constrained accuracy and lower FCD. Although MolEditRL incurs a one-time fine-tuning cost, this investment yields performance that surpasses both RL-from-scratch and large LLM-based baselines, which either suffer from poor editing precision or require heavy inference computation. At inference time, MolEditRL requires no oracle calls and achieves 1.39 s/sample, supporting real-time interactive molecular editing. These results demonstrate that MolEditRL achieves an advantageous balance of accuracy, robustness, and computational efficiency compared to existing baselines.

Table 17: Computational Efficiency and Editing Performance Comparison Across Baselines

| Method | Validity | $\text{Acc}_{\text{all}}$ (TS≥0.65) | $\text{Acc}_{\text{all}}$ (MCS≥0.6) | $\text{Acc}_{\text{all}}$ (GED≤4) | FCD↓ | Finetune Time (min) | Inference Time (s/sample) |
|---|---|---|---|---|---|---|---|
| GCPN | 0.858 | 0.000 | 0.000 | 0.029 | 20.260 | 32.92 | 0.78 |
| MolDQN | 1.000 | 0.003 | 0.142 | 0.045 | 13.152 | 35.76 | 4.73 |
| Reinvent4 | 0.835 | 0.124 | 0.196 | 0.139 | 13.786 | 40.51 | 0.85 |
| MolEditRL | 0.958 | **0.466** | **0.490** | **0.226** | **7.964** | 58.14 | 1.39 |

## Q GRADIENT DERIVATION UNDER $x_0$-PARAMETERIZATION

This appendix provides a complete derivation for the step in the main text going from the policy-gradient term $\nabla_\theta \log p_\theta(G_{\text{tgt}}^{t-1} \mid G_{\text{tgt}}^t, S, G_{\text{src}})$ to its $x_0$-parameterized form and the resulting reward-weighted cross-entropy objective.

### Q.1 SETUP AND ASSUMPTIONS

We consider a discrete diffusion setting on molecular graphs. The forward process defines a known corruption distribution with an analytic posterior $q(G_{\text{tgt}}^{t-1} \mid G_{\text{tgt}}^t, G_{\text{tgt}}^0)$. Let the conditioning variables be $C = (S, G_{\text{src}})$. Our modeling assumptions are standard in conditional diffusion: (i) the reverse-step factorization uses the *forward* posterior kernel, which depends only on $(G_{\text{tgt}}^t, G_{\text{tgt}}^0)$ and is independent of $\theta$ and $C$; and (ii) the model predicts $p_\theta(G_{\text{tgt}}^0 \mid G_{\text{tgt}}^t, C)$.

### Q.2 THE PRECISE VERSION OF THE MAIN-TEXT EQ. (12)

By the law of total probability and the above conditional independence,

$$p_\theta\big(G_{\text{tgt}}^{t-1} \mid G_{\text{tgt}}^t, C\big) = \sum_{G_{\text{tgt}}^0} q\big(G_{\text{tgt}}^{t-1} \mid G_{\text{tgt}}^t, G_{\text{tgt}}^0\big) \; p_\theta\big(G_{\text{tgt}}^0 \mid G_{\text{tgt}}^t, C\big). \tag{15}$$

Equation equation 15 is an *equality* in the discrete diffusion setting. The common single-sample/MAP form that replaces the sum by $\hat{G}_{\text{tgt}}^0$ then becomes an approximation.

### Q.3 LOG-GRADIENT OF A MIXTURE: A WEIGHTED EXPECTATION

Taking the logarithm of equation 15 and differentiating w.r.t. $\theta$, using that $q$ does not depend on $\theta$, yields the exact identity

$$\nabla_\theta \log p_\theta\big(G_{\text{tgt}}^{t-1} \mid G_{\text{tgt}}^t, C\big) = \sum_{G_{\text{tgt}}^0} w_\theta\big(G_{\text{tgt}}^0; G_{\text{tgt}}^{t-1}, G_{\text{tgt}}^t, C\big) \; \nabla_\theta \log p_\theta\big(G_{\text{tgt}}^0 \mid G_{\text{tgt}}^t, C\big), \tag{16}$$

with normalized weights

$$w_\theta\big(G_{\text{tgt}}^0; \cdot\big) = \frac{q\big(G_{\text{tgt}}^{t-1} \mid G_{\text{tgt}}^t, G_{\text{tgt}}^0\big) \; p_\theta\big(G_{\text{tgt}}^0 \mid G_{\text{tgt}}^t, C\big)}{\sum_{G'} q\big(G_{\text{tgt}}^{t-1} \mid G_{\text{tgt}}^t, G'\big) \; p_\theta\big(G' \mid G_{\text{tgt}}^t, C\big)}. \tag{17}$$

Thus, the exact gradient is a posterior-weighted expectation of $\nabla_\theta \log p_\theta(G_{\text{tgt}}^0 \mid G_{\text{tgt}}^t, C)$.

Q.4 PRACTICAL APPROXIMATIONS

Exact evaluation of equation 16 is intractable due to the exponential number of graphs. According to the Single-sample / straight-through (MAP) approximation, with $K=1$, or taking $\hat{G}^0_{\text{tgt}} = \arg\max p_\theta(G^0_{\text{tgt}} \mid G^t_{\text{tgt}}, C)$ and ignoring (or stop-gradient on) normalization, we obtain the widely used estimator

$$\nabla_\theta \log p_\theta(G^{t-1}_{\text{tgt}} \mid G^t_{\text{tgt}}, C) \ \approx \ \nabla_\theta \log p_\theta\left(\hat{G}^0_{\text{tgt}} \mid G^t_{\text{tgt}}, C\right), \tag{18}$$

which is the approximation used to move from the equation 12 to equation 13. Direct REINFORCE on $\log p_\theta(G^{t-1}_{\text{tgt}} \mid \cdot)$ has high variance due to (a) sparse rewards (only at $t{=}0$), (b) weak correlation between intermediate noisy states and the terminal reward, and (c) accumulated stochasticity across transitions. DDPO Black et al. (2023) treats the denoising steps as an MDP and applies step-wise policy-gradient surrogates; GDPO Liu et al. (2024b) adapts these ideas to discrete graph diffusion and proposes eager/low-variance estimators. The single-sample $x_0$ surrogate in equation 18 is a practical variance/computation trade-off also adopted in these works.

# R  DATASET STATISTICS

Our dataset is constructed following a procedure similar to DrugAssist Ye et al. (2025), involving three main steps: (1) drug-like molecules are filtered from public databases such as ZINC and ChEMBL based on Lipinskis Rule of Five; (2) Matched Molecular Pairs (MMP) are extracted using BRICS fragmentation to identify structurally similar molecule pairs with local edits; and (3) pairs showing significant property shifts are retained, and corresponding natural language instructions are generated to describe the desired property modifications. To prevent data leakage, we construct the splits at the molecule level using canonical SMILES: duplicate or equivalent structures are removed, and no molecule in the test set appears in the training set or in any RL fine-tuning inputs. This ensures that all test-time molecules are entirely unseen by the model. Table 18 reports descriptive statistics drawn directly from the MolEdit dataset. These ranges reflect the empirical distributions of property values in our collected molecule pairs. The physicochemical properties in our dataset are carefully selected in accordance with Lipinski's Rule of Five, a key set of guidelines for drug-like molecules that includes constraints on molecular weight ($\leq$ 500 Da), LogP ($\leq$ 5), hydrogen bond donors ($\leq$ 5), and hydrogen bond acceptors ($\leq$ 10). These constraints are reflected in the value ranges of our dataset properties. The dataset covers both biological activity properties and physicochemical properties, each playing crucial roles in drug discovery:

## R.1  BIOLOGICAL ACTIVITY PROPERTIES

- **DRD2** (Dopamine D2 receptor): A key target in antipsychotic drug development, with values ranging from 0 to 1 indicating binding probability. Our dataset captures substantial changes in DRD2 activity, from minor adjustments ($\pm 0.050$) to major shifts ($\pm 0.951$), where positive values indicate decreased binding and negative values indicate increased binding.

- **GSK3$\beta$** (Glycogen synthase kinase-3 beta): An important target in treating neurological disorders, with values from 0 to 1 representing inhibition probability. The dataset includes modifications ranging from $\pm 0.050$ to $\pm 0.750$.

- **JNK3** (c-Jun N-terminal kinase 3): A target for neurodegenerative diseases, with values from 0 to 1 indicating inhibition probability. Property changes range from subtle ($\pm 0.030$) to significant ($\pm 0.690$).

## R.2  PHYSICOCHEMICAL PROPERTIES

- **QED** (Quantitative Estimate of Drug-likeness): Ranges from 0 to 1, where higher values indicate better drug-likeness. Our dataset covers modifications from $\pm 0.380$ to $\pm 0.794$.

- **SA** (Synthetic Accessibility): Ranges from 1 to 10, where lower values indicate easier synthesis. The dataset includes substantial changes from $\pm 0.700$ to $\pm 6.563$.

Table 18: Descriptive statistics of property changes in the MolEdit dataset.

| Property | Direction | Pairs | Δ Range | Source Range | Target Range |
|---|---|---|---|---|---|
| DRD2 | ↑ | 80,627 | [-0.951, -0.050] | [0.000, 0.944] | [0.050, 1.000] |
| | ↓ | 80,627 | [0.050, 0.951] | [0.050, 1.000] | [0.000, 0.944] |
| GSK3β | ↑ | 98,310 | [-0.750, -0.050] | [0.000, 0.940] | [0.052, 0.990] |
| | ↓ | 98,310 | [0.050, 0.750] | [0.052, 0.990] | [0.000, 0.940] |
| JNK3 | ↑ | 94,131 | [-0.690, -0.030] | [0.000, 0.880] | [0.040, 0.990] |
| | ↓ | 94,131 | [0.030, 0.690] | [0.040, 0.990] | [0.000, 0.880] |
| QED | ↑ | 97,750 | [-0.794, -0.380] | [0.041, 0.564] | [0.438, 0.948] |
| | ↓ | 98,249 | [0.380, 0.794] | [0.438, 0.948] | [0.050, 0.565] |
| SA | ↑ | 90,192 | [-6.563, -0.700] | [1.059, 7.268] | [2.189, 7.999] |
| | ↓ | 87,453 | [0.700, 6.104] | [2.277, 7.996] | [1.397, 7.268] |
| LogP | ↑ | 89,088 | [-6.132, -2.625] | [-17.073, 2.369] | [-13.745, 5.000] |
| | ↓ | 90,489 | [2.625, 6.132] | [-13.745, 5.000] | [-17.073, 2.372] |
| MW | ↑ | 80,647 | [-195.744, -99.031] | [218.106, 399.216] | [336.084, 499.999] |
| | ↓ | 79,712 | [99.031, 195.744] | [336.073, 499.994] | [218.094, 400.241] |
| HAccept | ↑ | 98,562 | [-7.000, -2.000] | [0.000, 8.000] | [2.000, 10.000] |
| | ↓ | 98,562 | [2.000, 7.000] | [2.000, 10.000] | [0.000, 8.000] |
| HDonors | ↑ | 104,468 | [-5.000, -2.000] | [0.000, 3.000] | [2.000, 5.000] |
| | ↓ | 104,468 | [2.000, 5.000] | [2.000, 5.000] | [0.000, 3.000] |
| RotBonds | ↑ | 66,369 | [-9.000, -3.000] | [0.000, 7.000] | [3.000, 10.000] |
| | ↓ | 65,806 | [3.000, 9.000] | [3.000, 10.000] | [0.000, 7.000] |

- **MW** (Molecular Weight): A fundamental property ranging from 218 to 500 Da in our dataset, with modifications spanning $\pm 99.031$ to $\pm 195.744$ Da.

- **LogP** (Octanol-water partition coefficient): Measures lipophilicity, ranging from $-17$ to $5$ in our dataset, with changes from $\pm 2.625$ to $\pm 6.132$.

- **HDONORS** (Hydrogen Bond Donors): Ranges from 0 to 5, with modifications of $\pm 2$ to $\pm 5$ donors.

- **HACCEPT** (Hydrogen Bond Acceptors): Ranges from 0 to 10, with changes of $\pm 2$ to $\pm 7$ acceptors.

- **ROTBONDS** (Rotatable Bonds): Ranges from 0 to 10, with modifications of $\pm 3$ to $\pm 9$ bonds, affecting molecular flexibility.

For each property, table 18 shows the number of molecular pairs, the range of property changes (Δ Range), and the value distributions in both source and target molecules. The $\pm$ notation indicates that changes occur in both directions  positive values for property reduction and negative values for property increase, representing the observed range of property modifications across all molecule pairs in the dataset.

R.3  NATURAL LANGUAGE PROMPTS

Table 19 presents the natural language prompts designed for our single property editing tasks. For each of the ten molecular properties, we crafted two complementary prompts corresponding to property value increase and decrease. The prompts are purposefully designed to be clear and concise while maintaining chemical accuracy and relevance. For biological activity properties (DRD2, GSK3β, JNK3), the prompts emphasize binding affinity and inhibitory activity. For physicochemical properties, the prompts use specific chemical terminology (e.g., "hydrogen bond acceptors," "rotatable bonds") while remaining accessible. Some prompts, such as those for LogP, include additional context about the property's practical implications (e.g., "enhance its fat solubility" or "improve its water solubility"). Each prompt contains a [SMILE] placeholder that is replaced with the actual SMILES string of the molecule to be modified during the editing process.

Table 19: Natural language prompts for single property editing tasks.

| Property | Direction | Prompt |
|---|---|---|
| DRD2 | ↑ ↓ | Optimize this molecule [SMILE] to increase its DRD2 binding affinity. Help me reduce the DRD2 binding activity of molecule [SMILE]. |
| GSK3$\beta$ | ↑ ↓ | Help me optimize this molecule [SMILE] to improve its GSK3$\beta$ inhibitory activity. Reduce the GSK3$\beta$ inhibition potential of this molecule [SMILE]. |
| JNK3 | ↑ ↓ | Enhance the JNK3 binding properties of molecule [SMILE]. Make changes to lower the JNK3 binding affinity of molecule [SMILE]. |
| QED | ↑ ↓ | Optimize the QED score of molecule [SMILE] to make it more drug-like. Decrease the QED value of this molecule [SMILE]. |
| SA | ↑ ↓ | Make this molecule [SMILE] harder to synthesize. Make this molecule [SMILE] easier to synthesize. |
| LogP | ↑ ↓ | Help me increase the LogP value of molecule [SMILE] to enhance its fat solubility. Help me decrease the LogP value of molecule [SMILE] to improve its water solubility. |
| MW | ↑ ↓ | Help me increase the molecular weight of this molecule [SMILE]. Help me reduce the molecular weight of this molecule [SMILE]. |
| HAccept | ↑ ↓ | Add more hydrogen bond acceptors to this molecule [SMILE]. Reduce the number of hydrogen bond acceptors in molecule [SMILE]. |
| HDonors | ↑ ↓ | Help me increase the number of H-bond donors in [SMILE]. Help me decrease the H-bond donor count in this molecule [SMILE]. |
| RotBonds | ↑ ↓ | Add more rotatable bonds to this molecule [SMILE]. Reduce the number of rotatable bonds in molecule [SMILE]. |

## S    LIMITATIONS AND FUTURE WORK

MolEditRL demonstrates strong and consistent performance in structure-preserving editing across a wide range of chemical properties, particularly on small to medium-sized molecules. While our current experiments focus on this regime, the underlying framework is designed to generalize and is expected to extend effectively to larger biomolecules, such as proteins or complex natural products, with minor adaptations. The reinforcement learning component leverages property oracles (e.g., from RDKit and TDC) to guide optimization. These oracles validate MolEditRLs effectiveness on widely studied molecular properties. For less-characterized or emerging attributes, task-specific predictors can be trained and integrated, enabling flexible extension of the framework to new property domains.

However, in particular, the framework may struggle when a target property is extremely rare, lacks a reliable predictive model, or requires prohibitively expensive evaluations. In addition, when user instructions contain logical contradictionssuch as requesting simultaneous improvement of mutually exclusive propertiesthe model is unable to produce feasible edits. Looking ahead, we plan to explore interactive, dialogue-based molecular editing, enabling users to iteratively refine molecules via multi-turn natural language instructions. This direction could support more intuitive and human-centric workflows for molecular design and lead optimization.

## T    LLM USAGE STATEMENT

Large language models were employed solely as general-purpose assistance tools during the writing process, specifically for improving clarity and checking grammar. All technical contributions, experimental results, and scientific insights are entirely the authors own work. No LLMs were used to generate core research ideas, experimental data, or technical implementations. The authors take full responsibility for all content and claims presented in this paper.

Table 20: Extended results on single-property molecular editing tasks. Bold indicates best performance. Arrows (↑, ↓) denote desired property increase or decrease.

| Model | Task | Validity | Acc$_{all}$ (0.65) | Acc$_{valid}$ (0.65) | Acc$_{all}$ (0.15) | Acc$_{valid}$ (0.15) | FCD | Task | Validity | Acc$_{all}$ (0.65) | Acc$_{valid}$ (0.65) | Acc$_{all}$ (0.15) | Acc$_{valid}$ (0.15) | FCD |
|---|---|---|---|---|---|---|---|---|---|---|---|---|---|---|
| REINVENT4 | | 0.524 | 0.19 | 0.3626 | 0.4 | 0.7634 | 11.0566 | | 0.704 | 0.19 | 0.2699 | 0.442 | 0.6278 | 11.7456 |
| MolGen | | 1.0 | 0.022 | 0.022 | 0.256 | 0.256 | 14.6826 | | 1.0 | 0.004 | 0.004 | 0.404 | 0.404 | 14.943 |
| BioT5 | | 1.0 | 0.0 | 0.0 | 0.148 | 0.148 | 30.3159 | | 1.0 | 0.0 | 0.0 | 0.472 | 0.472 | 15.1916 |
| DrugAssist | HACCEPT↑ | 0.9439 | 0.3467 | 0.3673 | 0.4429 | 0.4692 | 8.7609 | HACCEPT↓ | 0.9819 | 0.161 | 0.1639 | 0.3421 | 0.3484 | 11.8052 |
| Gellmo_M | | 0.904 | 0.064 | 0.0708 | 0.15 | 0.1659 | 14.259 | | 0.89 | 0.298 | 0.3348 | 0.524 | 0.5888 | 9.2035 |
| Gellmo_L | | 0.89 | 0.07 | 0.0787 | 0.162 | 0.182 | 12.7839 | | 0.914 | 0.178 | 0.1947 | 0.508 | 0.5558 | 8.8893 |
| MolEditRL | | 0.968 | **0.484** | **0.5** | **0.826** | **0.8533** | **7.3163** | | 0.974 | **0.388** | **0.3984** | **0.712** | **0.731** | **9.0711** |
| REINVENT4 | | 0.568 | 0.268 | 0.4718 | 0.548 | 0.7648 | 9.966 | | 0.7581 | 0.3841 | 0.5067 | 0.6585 | 0.8686 | 7.7976 |
| MolGen | | 1.0 | 0.038 | 0.038 | 0.418 | 0.418 | 9.8619 | | 1.0 | 0.016 | 0.016 | 0.432 | 0.432 | 14.4007 |
| BioT5 | | 1.0 | 0.0 | 0.0 | 0.36 | 0.36 | 16.5037 | | 1.0 | 0.0 | 0.0 | 0.348 | 0.348 | 17.2348 |
| DrugAssist | SA↑ | 0.988 | 0.216 | 0.2186 | 0.294 | 0.2976 | 10.14 | MW↓ | 0.98 | 0.5391 | 0.5501 | 0.5872 | 0.5992 | 9.2859 |
| Gellmo_M | | 0.91 | 0.12 | 0.1319 | 0.288 | 0.3165 | 8.9319 | | 0.898 | 0.29 | 0.3229 | 0.564 | 0.6281 | 6.6778 |
| Gellmo_L | | 0.912 | 0.104 | 0.114 | 0.28 | 0.307 | 8.9652 | | 0.906 | 0.26 | 0.287 | 0.684 | 0.755 | 6.6965 |
| MolEditRL | | 0.95 | **0.49** | **0.5158** | **0.776** | **0.8168** | **7.5281** | | 0.984 | **0.632** | **0.6423** | **0.952** | **0.9675** | **6.3935** |
| REINVENT4 | | 0.678 | 0.268 | 0.3953 | 0.458 | 0.6755 | 11.739 | | 0.7 | 0.286 | 0.4086 | 0.418 | 0.5971 | 9.2662 |
| MolGen | | 1.0 | 0.022 | 0.022 | 0.243 | 0.243 | 13.4729 | | 1.0 | 0.042 | 0.042 | 0.418 | 0.418 | 11.0476 |
| BioT5 | | 1.0 | 0.0 | 0.0 | 0.144 | 0.144 | 23.7964 | | 1.0 | 0.0 | 0.0 | 0.272 | 0.272 | 16.231 |
| DrugAssist | HDONORS↑ | 0.9319 | 0.3267 | 0.3505 | 0.4449 | 0.4774 | 8.4057 | DRD2↓ | 0.984 | 0.524 | 0.5325 | 0.57 | 0.5793 | 7.5935 |
| Gellmo_M | | 0.898 | 0.044 | 0.049 | 0.092 | 0.1024 | 12.5726 | | 0.922 | 0.136 | 0.1475 | 0.274 | 0.2972 | 9.3582 |
| Gellmo_L | | 0.896 | 0.04 | 0.0446 | 0.1 | 0.1116 | 14.7698 | | 0.916 | 0.132 | 0.1441 | 0.336 | 0.3668 | 9.8566 |
| MolEditRL | | 0.942 | **0.582** | **0.6178** | **0.842** | **0.8938** | 8.0114 | | 0.986 | **0.656** | **0.6639** | **0.72** | **0.7302** | **6.549** |
| REINVENT4 | | 0.61 | 0.114 | 0.1869 | 0.36 | 0.5902 | 13.997 | | 0.508 | 0.268 | 0.5276 | 0.424 | 0.8346 | 7.5582 |
| MolGen | | 1.0 | 0.094 | 0.094 | 0.474 | 0.474 | 10.5549 | | 1.0 | 0.11 | 0.11 | 0.474 | 0.474 | 13.7158 |
| BioT5 | | 1.0 | 0.0 | 0.0 | 0.492 | 0.492 | 27.5634 | | 1.0 | 0.0 | 0.0 | 0.202 | 0.202 | 34.7724 |
| DrugAssist | LOGP↑ | 0.964 | 0.382 | 0.3963 | 0.442 | 0.4585 | 11.3282 | LOGP↓ | 0.966 | 0.548 | 0.5673 | 0.604 | 0.6253 | 6.3703 |
| Gellmo_M | | 0.89 | 0.374 | 0.4202 | 0.724 | 0.8135 | 6.878 | | 0.906 | 0.004 | 0.0044 | 0.224 | 0.2472 | 12.9582 |
| Gellmo_L | | 0.91 | 0.268 | 0.2945 | 0.59 | 0.6484 | 6.7566 | | 0.918 | 0.15 | 0.1634 | 0.444 | 0.4837 | 7.7802 |
| MolEditRL | | 0.964 | **0.578** | **0.5996** | **0.91** | **0.944** | **6.0118** | | 0.972 | **0.71** | **0.7305** | **0.94** | **0.9671** | **5.1015** |
| REINVENT4 | | 0.61 | 0.112 | 0.1836 | 0.384 | 0.6295 | 11.671 | | 0.652 | 0.15 | 0.2301 | 0.312 | 0.4785 | 11.1066 |
| MolGen | | 1.0 | 0.084 | 0.084 | 0.356 | 0.356 | 11.3428 | | 1.0 | 0.024 | 0.024 | 0.421 | 0.421 | 10.9996 |
| BioT5 | | 1.0 | 0.0 | 0.0 | 0.306 | 0.306 | 16.85 | | 1.0 | 0.0 | 0.0 | 0.374 | 0.374 | 15.7723 |
| DrugAssist | ROTBONDS↑ | 0.9537 | 0.1469 | 0.154 | 0.2716 | 0.2848 | 10.7588 | QED↓ | 0.9859 | 0.1044 | 0.1059 | 0.247 | 0.2505 | 10.8724 |
| Gellmo_M | | 0.888 | 0.072 | 0.0811 | 0.16 | 0.1802 | 12.1059 | | 0.924 | 0.012 | 0.013 | 0.15 | 0.1623 | 15.4165 |
| Gellmo_L | | 0.888 | 0.098 | 0.1104 | 0.218 | 0.2455 | 10.0684 | | 0.904 | 0.088 | 0.0973 | 0.218 | 0.2412 | 10.3736 |
| MolEditRL | | 0.934 | **0.392** | **0.4197** | **0.764** | **0.818** | **7.2532** | | 0.948 | **0.612** | **0.6456** | **0.894** | **0.943** | **6.9314** |

## U    EXTENDED SINGLE-PROPERTY RESULTS

Table 20 reports extended quantitative results for 10 representative single-property molecular editing tasks from the MolEdit-Instruct benchmark. MolEditRL consistently achieves the highest accuracy across both similarity thresholds, while maintaining high chemical validity and the lowest FCD scores across most tasks. This indicates strong structural fidelity and superior alignment with target property distributions. In contrast, baselines such as BioT5 and MolGen often generate valid molecules but fail to satisfy property and similarity constraints. REINVENT4 and DrugAssist perform moderately well but fall short in structural preservation and distributional realism. These detailed results further confirm the robustness and effectiveness of MolEditRL in single-property editing scenarios.

## V    EXTENDED MULTI-PROPERTY RESULTS

Table 21 presents detailed evaluation results on multi-property molecular editing tasks from the MolEdit-Instruct benchmark. Each task involves optimizing 2 to 4 chemical properties simultaneously, reflecting practical constraints encountered in real-world molecular design. MolEditRL consistently achieves strong performance across all multi-property tasks, demonstrating its ability to balance complex property requirements while preserving molecular validity and structural similarity. The results confirm its robustness under increasingly constrained and realistic editing scenarios. The property combinations in these tasks are carefully selected to reflect common design goals in medicinal chemistry. For example, tasks like (HACCEPT↓, HDONORS↓) aim to reduce molecular polarity, which is essential for improving membrane permeability and bioavailability. (LOGP↓, ROTBONDS↓) targets molecules with lower lipophilicity and rigidity, which improves metabolic stability and reduces off-target binding. On the other hand, combinations such as (MW↑, QED↓) simulate early-stage exploration of larger, less drug-like molecules, often relevant in hit expansion or macrocycle design. Biologically motivated combinations like (DRD2↓, GSK3$\beta$↑) reflect efforts to reduce off-target dopamine receptor activity while enhancing GSK3$\beta$ inhibition, a common challenge in polypharmacology. Furthermore, high-complexity tasks such as (GSK3$\beta$↑, HDONORS↑,

Table 21: Extended results on multi-property molecular editing tasks. Bold indicates best performance. Arrows (↑, ↓) denote desired property increase or decrease.

| Model | Task | Validity | Acc$_{all}$ (0.65) | Acc$_{valid}$ (0.65) | Acc$_{all}$ (0.15) | Acc$_{valid}$ (0.15) | FCD | Task | Validity | Acc$_{all}$ (0.65) | Acc$_{valid}$ (0.65) | Acc$_{all}$ (0.15) | Acc$_{valid}$ (0.15) | FCD |
|---|---|---|---|---|---|---|---|---|---|---|---|---|---|---|
| BioT5 | | 1.0 | 0.0 | 0.0 | 0.352 | 0.352 | 17.731 | | 1.0 | 0.0 | 0.0 | 0.19 | 0.19 | 19.8292 |
| DrugAssist | HACCEPT↓ | 0.9819 | 0.2711 | 0.2761 | 0.3574 | 0.364 | 12.987 | JNK3↓ | 0.98 | 0.292 | 0.298 | 0.336 | 0.3429 | 11.1755 |
| GeLLM$^3$O_M | HDONORS↓ | 0.89 | 0.108 | 0.1213 | 0.264 | 0.2966 | 14.7575 | QED↑ | 0.914 | 0.148 | 0.1619 | 0.326 | 0.3567 | 10.123 |
| GeLLM$^3$O_L | | 0.9 | 0.146 | 0.1622 | 0.36 | 0.4 | 12.0622 | | 0.9 | 0.098 | 0.1089 | 0.352 | 0.3911 | 10.866 |
| MolEditRL | | 0.972 | **0.358** | **0.3739** | **0.612** | **0.6497** | **11.7393** | | 0.976 | **0.33** | **0.3381** | **0.416** | **0.4262** | **9.6139** |
| BioT5 | | 1.0 | 0.0 | 0.0 | 0.098 | 0.098 | 24.7313 | | 1.0 | 0.0 | 0.0 | 0.088 | 0.088 | 24.19 |
| DrugAssist | HACCEPT↑ | 0.954 | 0.226 | 0.2369 | 0.284 | 0.2977 | 11.5424 | DRD2↓ | 0.992 | 0.104 | 0.1048 | 0.126 | 0.127 | 12.3998 |
| GeLLM$^3$O_M | SA↑ | 0.918 | 0.012 | 0.0131 | 0.07 | 0.0763 | 23.0712 | GSK3B↑ | 0.942 | 0.036 | 0.0382 | 0.064 | 0.0679 | 15.6116 |
| GeLLM$^3$O_L | | 0.904 | 0.026 | 0.0288 | 0.048 | 0.0531 | 14.8785 | | 0.9 | 0.05 | 0.0556 | 0.098 | 0.1089 | 12.7831 |
| MolEditRL | | 0.962 | **0.316** | **0.3583** | **0.58** | **0.6576** | **11.2492** | | 0.97 | **0.186** | **0.1918** | **0.228** | **0.2351** | **11.4433** |
| BioT5 | | 1.0 | 0.0 | 0.0 | 0.104 | 0.104 | 27.651 | | 1.0 | 0.0 | 0.0 | 0.25 | 0.25 | 28.7563 |
| DrugAssist | LOGP↓ | 0.98 | 0.346 | 0.3531 | 0.384 | 0.3918 | 8.4341 | DRD2↑ | 0.976 | 0.212 | 0.2172 | 0.254 | 0.2602 | 11.4352 |
| GeLLM$^3$O_M | ROTBONDS↓ | 0.892 | 0.032 | 0.0359 | 0.128 | 0.1435 | 16.9039 | SA↑ | 0.898 | 0.102 | 0.1136 | 0.232 | 0.2584 | 11.5436 |
| GeLLM$^3$O_L | | 0.908 | 0.09 | 0.0991 | 0.3 | 0.3304 | 10.9156 | | 0.924 | 0.07 | 0.0758 | 0.222 | 0.2403 | 12.0196 |
| MolEditRL | | 0.97 | **0.454** | **0.468** | **0.686** | **0.7072** | **6.2095** | | 0.912 | **0.23** | **0.2522** | **0.398** | **0.4364** | **10.8934** |
| BioT5 | | 1.0 | 0.0 | 0.0 | 0.072 | 0.072 | 31.688 | | 1.0 | 0.0 | 0.0 | 0.216 | 0.216 | 19.1627 |
| DrugAssist | LOGP↑ | 0.96 | 0.09 | 0.0938 | 0.1 | 0.1042 | 19.404 | QED↓ | 0.984 | 0.24 | 0.2439 | 0.276 | 0.2805 | 11.0221 |
| GeLLM$^3$O_M | ROTBONDS↑ | 0.86 | 0.014 | 0.0163 | 0.034 | 0.0395 | 30.3704 | ROTBONDS↑ | 0.878 | 0.014 | 0.0159 | 0.096 | 0.1093 | 21.1825 |
| GeLLM$^3$O_L | | 0.906 | 0.022 | 0.0243 | 0.06 | 0.0662 | 16.6925 | | 0.902 | 0.064 | 0.071 | 0.166 | 0.184 | 11.7206 |
| MolEditRL | | 0.954 | **0.344** | **0.3891** | **0.634** | **0.7172** | **12.0673** | | 0.943 | **0.422** | **0.4742** | **0.83** | **0.9326** | **7.564** |
| BioT5 | | 1.0 | 0.0 | 0.0 | 0.27 | 0.27 | 17.3349 | | 1.0 | 0.0 | 0.0 | 0.196 | 0.196 | 20.749 |
| DrugAssist | MW↑ | 0.98 | 0.298 | 0.3041 | 0.354 | 0.3612 | 9.5465 | QED↓ | 0.978 | 0.2325 | 0.2377 | 0.2766 | 0.2828 | 11.0132 |
| GeLLM$^3$O_M | QED↓ | 0.926 | 0.072 | 0.0778 | 0.238 | 0.257 | 12.8711 | SA↑ | 0.906 | 0.086 | 0.0949 | 0.184 | 0.2031 | 10.3846 |
| GeLLM$^3$O_L | | 0.882 | 0.158 | 0.1791 | 0.316 | 0.3583 | 7.8458 | | 0.894 | 0.078 | 0.0872 | 0.172 | 0.1924 | 10.3566 |
| MolEditRL | | 0.944 | **0.35** | **0.4147** | **0.79** | **0.936** | **7.0482** | | 0.938 | **0.592** | **0.6311** | **0.878** | **0.936** | **7.3882** |
| BioT5 | | 1.0 | 0.0 | 0.0 | 0.016 | 0.016 | 49.8934 | | 1.0 | 0.0 | 0.0 | 0.07 | 0.07 | 36.9063 |
| DrugAssist | DRD2↓ | 0.95 | 0.062 | 0.0653 | 0.082 | 0.0863 | 14.6988 | DRD2↑ | 0.956 | 0.142 | 0.1485 | 0.192 | 0.2008 | 13.9149 |
| GeLLM$^3$O_M | HACCEPT↑ | 0.91 | 0.018 | 0.0198 | 0.034 | 0.0374 | 16.9543 | HACCEPT↑ | 0.904 | 0.004 | 0.0044 | 0.064 | 0.0708 | 28.5396 |
| GeLLM$^3$O_L | MW↓ | 0.916 | 0.01 | 0.0109 | 0.02 | 0.0218 | 24.1798 | SA↑ | 0.92 | 0.016 | 0.0174 | 0.05 | 0.0543 | 18.2579 |
| MolEditRL | | 0.962 | **0.1** | **0.104** | **0.264** | **0.2744** | **12.5672** | | 0.966 | **0.192** | **0.1988** | **0.288** | **0.2981** | **13.7681** |
| BioT5 | | 1.0 | 0.0 | 0.0 | 0.086 | 0.086 | 28.2977 | | 1.0 | 0.0 | 0.0 | 0.116 | 0.116 | 23.0032 |
| DrugAssist | DRD2↑ | 0.972 | 0.09 | 0.0926 | 0.14 | 0.144 | 15.8384 | DRD2↑ | 0.99 | 0.114 | 0.1152 | 0.158 | 0.1596 | 13.438 |
| GeLLM$^3$O_M | HACCEPT↑ | 0.904 | 0.03 | 0.0332 | 0.082 | 0.0907 | 23.1175 | JNK3↑ | 0.91 | 0.062 | 0.0681 | 0.13 | 0.1429 | 15.6618 |
| GeLLM$^3$O_L | JNK3↑ | 0.91 | 0.026 | 0.0286 | 0.054 | 0.0593 | 18.5348 | QED↓ | 0.912 | 0.034 | 0.0373 | 0.082 | 0.0899 | 18.1148 |
| MolEditRL | | 0.94 | **0.22** | **0.234** | **0.294** | **0.3128** | **13.1873** | | 0.938 | **0.258** | **0.2751** | **0.438** | **0.467** | **9.2013** |
| BioT5 | | 1.0 | 0.0 | 0.0 | 0.066 | 0.066 | 32.3437 | | 1.0 | 0.0 | 0.0 | 0.024 | 0.024 | 43.4139 |
| DrugAssist | GSK3B↑ | 0.948 | 0.056 | 0.0591 | 0.062 | 0.0654 | 21.3896 | DRD2↓ | 0.954 | 0.02 | 0.021 | 0.028 | 0.0294 | 19.8179 |
| GeLLM$^3$O_M | HDONORS↑ | 0.902 | 0.002 | 0.0022 | 0.004 | 0.0044 | 24.7065 | GSK3B↑ | 0.884 | 0.0 | 0.0 | 0.006 | 0.0068 | 60.738 |
| GeLLM$^3$O_L | QED↓ SA↑ | 0.914 | 0.012 | 0.0131 | 0.036 | 0.0394 | 18.4036 | HDONORS↑ LOGP↓ | 0.898 | 0.002 | 0.0022 | 0.008 | 0.0089 | 19.9507 |
| MolEditRL | | 0.954 | **0.206** | **0.2159** | **0.416** | **0.4361** | **14.599** | | 0.962 | **0.174** | **0.1809** | **0.232** | **0.2412** | **11.4978** |
| BioT5 | | 1.0 | 0.0 | 0.0 | 0.088 | 0.088 | 30.4482 | | 1.0 | 0.0 | 0.0 | 0.1 | 0.1 | 27.6262 |
| DrugAssist | DRD2↓ | 0.988 | 0.082 | 0.083 | 0.092 | 0.0931 | 21.3253 | GSK3B↓ | 0.992 | 0.09 | 0.0907 | 0.098 | 0.0988 | 24.7748 |
| GeLLM$^3$O_M | GSK3B↓ | 0.91 | 0.042 | 0.0462 | 0.094 | 0.1033 | 18.8187 | HDONORS↓ | 0.906 | 0.05 | 0.0552 | 0.108 | 0.1192 | 18.0398 |
| GeLLM$^3$O_L | HACCEPT↓ | 0.918 | 0.042 | 0.0458 | 0.12 | 0.1307 | 20.2584 | LOGP↑ | 0.91 | 0.034 | 0.0374 | 0.11 | 0.1209 | 20.1217 |
| MolEditRL | SA↓ | 0.986 | **0.122** | **0.1237** | **0.21** | **0.213** | **14.735** | MW↓ | 0.966 | **0.146** | **0.1511** | **0.212** | **0.2195** | **18.0029** |

QED↓, SA↑) require optimizing target activity while managing solubility, drug-likeness, and synthetic complexitymirroring real trade-offs in lead optimization pipelines. These results collectively showcase MolEditRLs effectiveness not only in individual property edits but also in realistic, multi-objective optimization scenarios critical for practical drug development.

# W  ETHICS STATEMENT

This work focuses on computational molecular editing for drug discovery applications. We acknowledge several ethical considerations: (1) Dataset Release: We release the MolEdit-Instruct dataset publicly to benefit the research community, following established practices for molecular datasets. All molecular data is derived from publicly available databases (ZINC, ChEMBL) and contains no proprietary or sensitive information. (2) Intended Applications: Our method is designed to support legitimate drug discovery research, and the dataset is intended for beneficial applications in medicine and chemistry. (3) Reproducibility: We provide comprehensive implementation details, hyperparameters, and dataset construction procedures to ensure reproducible research. (4) No human subjects were involved in this study, and all experiments were conducted on computational datasets.

# X  REPRODUCIBILITY STATEMENT

To ensure reproducibility of our results, we provide comprehensive details across multiple sections: (1) Model Architecture: Complete architectural specifications are provided in Appendix, including all hyperparameters, network dimensions, and training configurations. (2) Training Setup: Detailed training procedures, optimization settings, and hardware specifications are documented in Appendix. (3) Dataset Construction: The MolEdit-Instruct dataset construction process is thoroughly described in Appendix, including property definitions, filtering criteria, and prompt generation procedures. The dataset is publicly available on Hugging Face. (4) Experimental Details: All evaluation metrics,

baseline implementations, and experimental protocols are specified in Section 4. (5) Code Availability: Upon acceptance, we will release the complete implementation including model code, training scripts, and evaluation pipelines to facilitate reproduction of all reported results.

# Y  MORE VISUALIZATION OF MOLECULAR EDITING

To further illustrate the editing behavior of different models, we present additional qualitative results in Figure 8, Figure 9, and Figure 10. These figures show visualization of edits across 20 single-property tasks. For each task, subfigure (a) displays the source molecule, and subfigures (be) show successful edits produced by BioT5, DrugAssist, GeLLMO_L, and MolEditRL, respectively. Red-colored substructures indicate regions that have been modified relative to the source molecule. Across all tasks, MolEditRL consistently achieves the highest number of successful edits, as well as the best structural fidelitypreserving the core scaffold of the original molecule while precisely introducing the required modifications. Additionally, Figure 12, Figure 13, and Figure 14 highlight side-by-side visual comparisons of different models editing the same molecular structure for a single target property. These visualizations confirm that only MolEditRL can reliably perform property-aligned edits while preserving molecular similarity. Competing models often over-modify or disrupt key structural elements, leading to reduced similarity or invalid transformations.

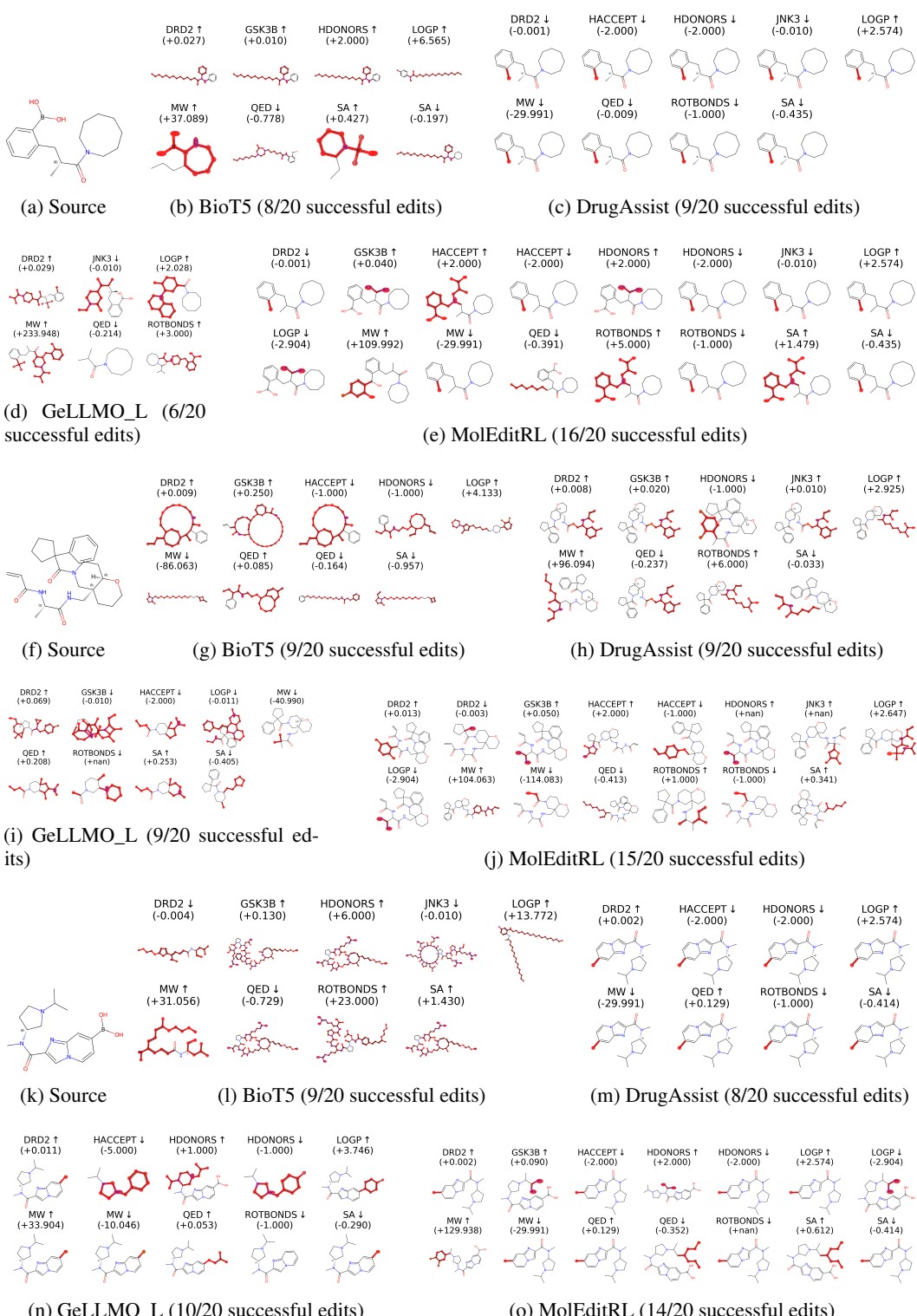

Figure 8: More visualization of edits on 20 tasks.

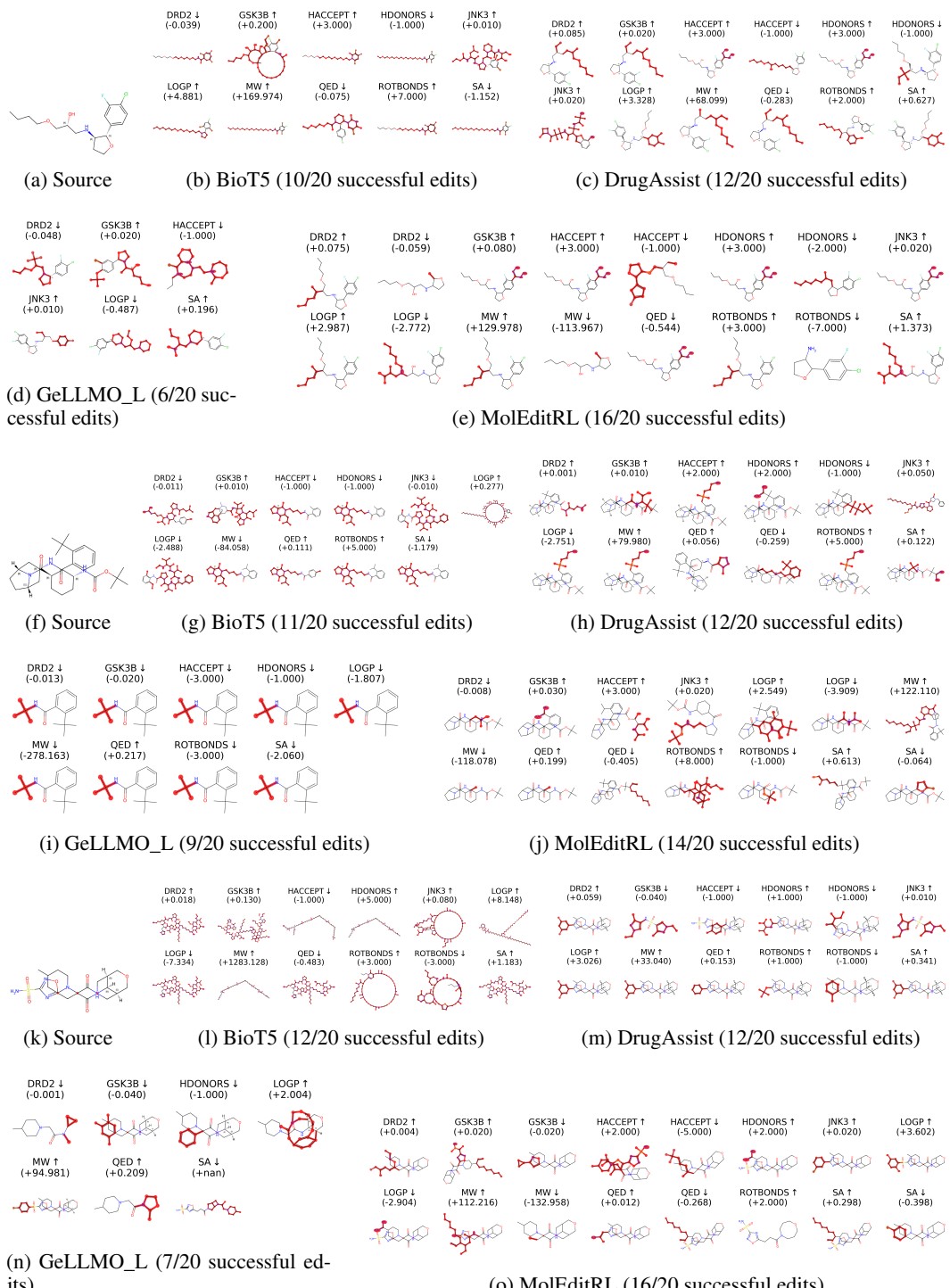

Figure 9: More visualization of edits on 20 tasks.

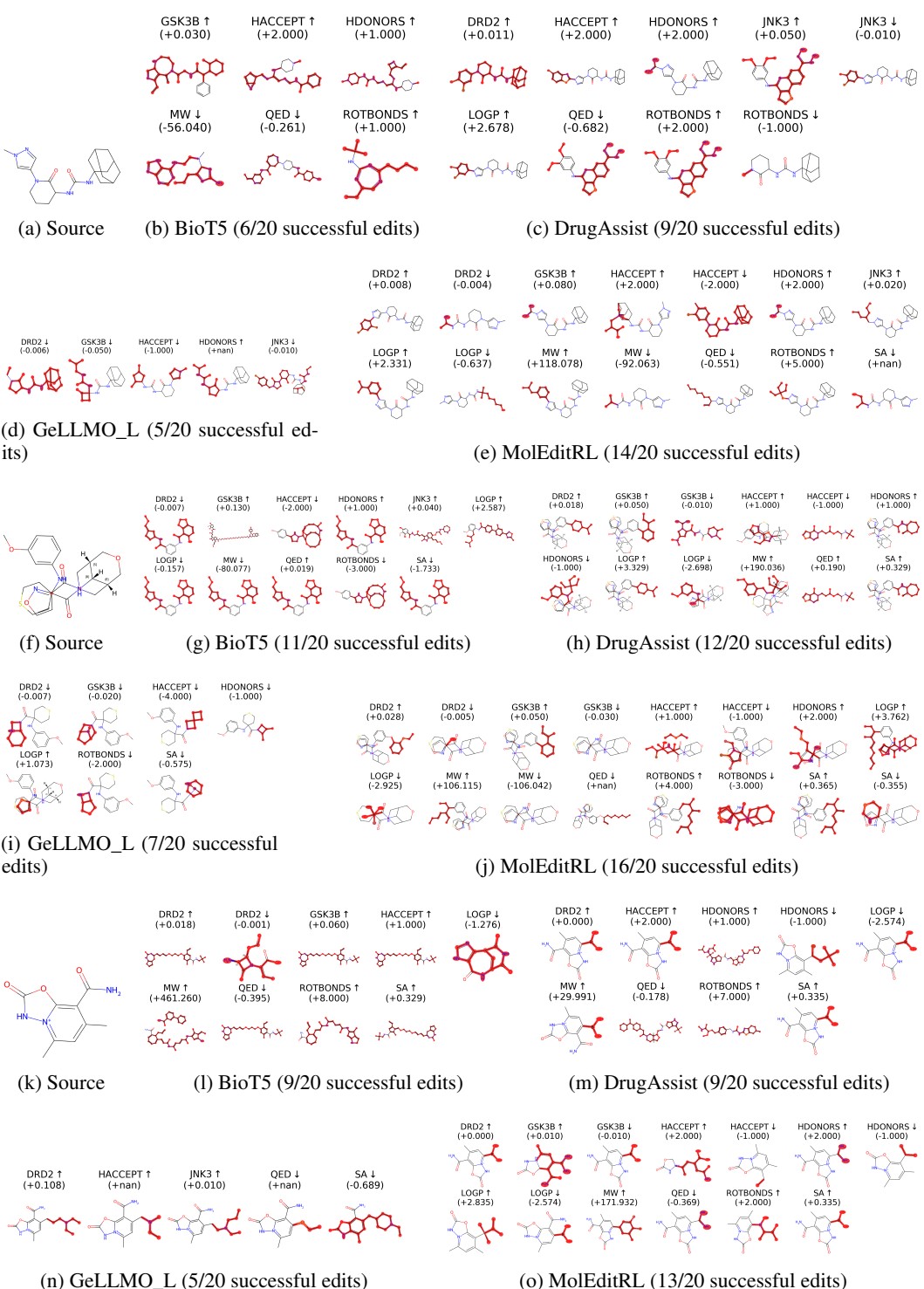

Figure 10: More visualization of edits on 20 tasks.

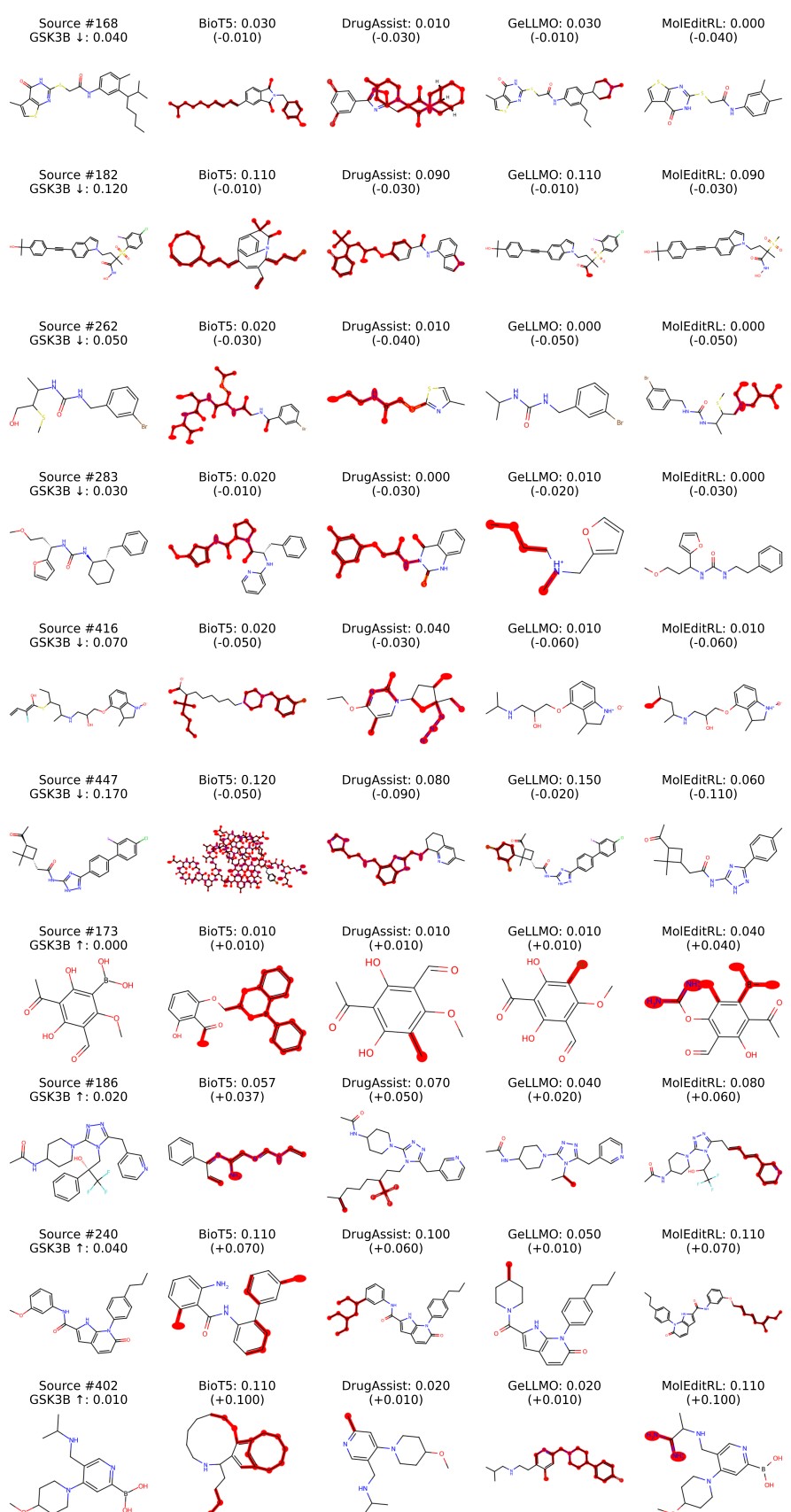

Figure 11: Qualitative comparison of molecular editing methods.

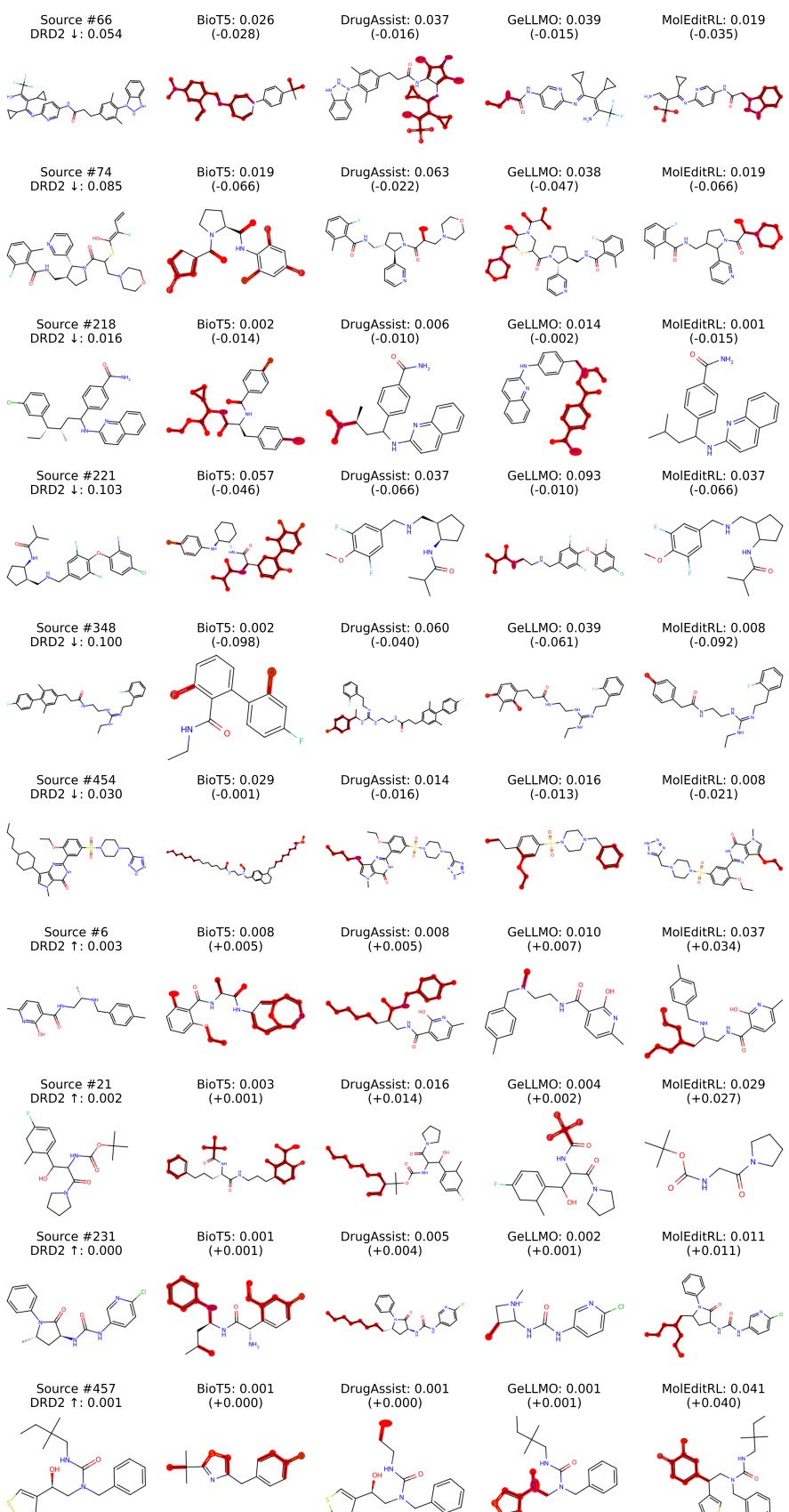

Figure 12: Qualitative comparison of molecular editing methods.

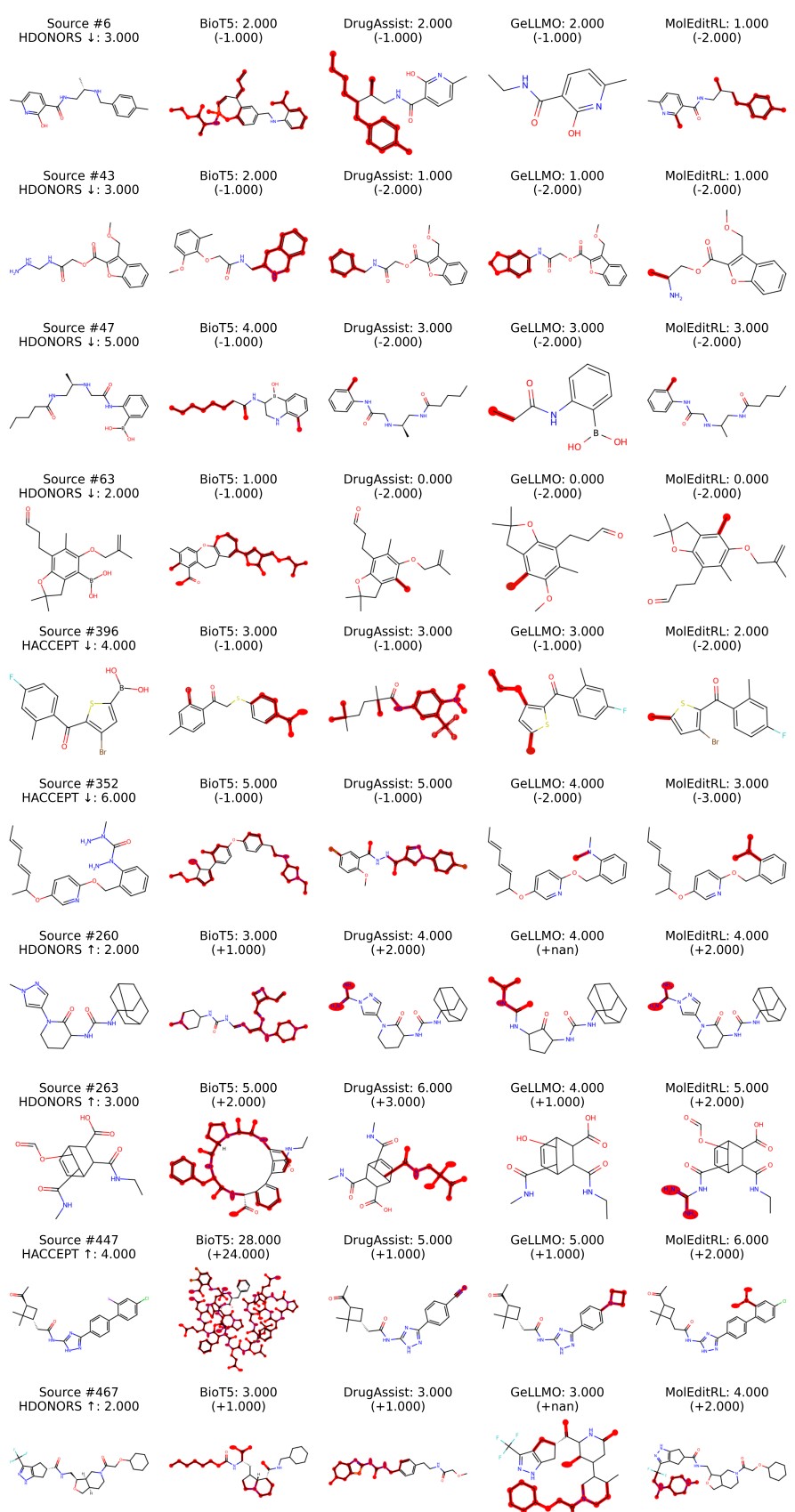

Figure 13: Qualitative comparison of molecular editing methods.

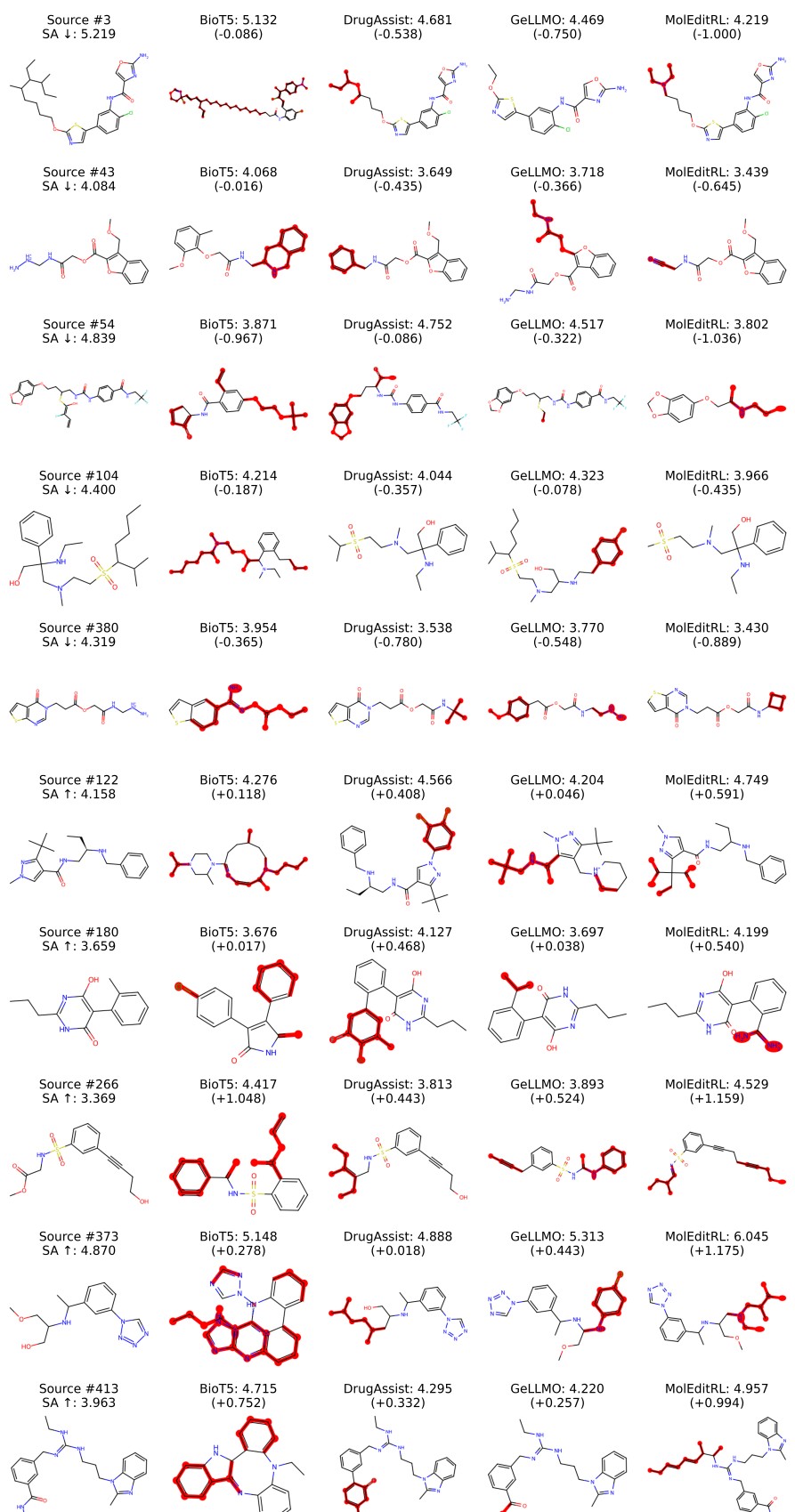

Figure 14: Qualitative comparison of molecular editing methods.

