# OpenReview forum: "MolEditRL: Structure-Preserving Molecular Editing via Discrete Diffusion and Reinforcement Learning"
_ICLR.cc/2026/Conference — ICLR 2026 Poster_

### Official Review · Reviewer_F1Hf · 2025-10-27

**Soundness:** 3
**Presentation:** 3
**Contribution:** 3
**Rating:** 6
**Confidence:** 2

**Summary:**

This paper introduces MolEditRL, a model that modifies input molecules to optimized some desired chemical property, which is provided by the user as a text prompt. The MolEditRL framework works by combining a graph diffusion model with a reinforcement learning training strategy, with rewards coming from an oracle (e.g. RDKit) for the chemical property of interest.
The paper also proposes a novel dataset, MolEdit, tailored for benchmarking molecular editing models with natural language instructions.
The method is benchmarked on MolEdit against baselines from the literature, and shows consistently state of the art performance across several tasks and evaluation metrics.

**Strengths:**

- The paper is quite well-written, as it clearly states the motivation and the high level idea of the method. Even though I am not familiar with the literature, I could easily understand most of the paper.
- The empirical results shown in the paper are very impressive, as consistently MolEditRL shows state-of-the-art performance across several tasks and evaluation metrics.
- The paper shows results not only on their new MolEdit dataset, but also on the C-MuMOInstruct dataset from the literature. Even here, MolEditRL is most of the time the leading model.

**Weaknesses:**

- Some details about the use of an oracle as a reward are unclear to me. It is unclear whether the model would work for chemical properties for which no oracle is available, or if the oracle can be noisy (e.g. a ML model for function/property prediction)
- Related to the point above, it is unclear to me how the oracle for the reward is chosen. It seems like the oracle should depend on the textual instruction S, but there are no details on how it is selected.
- A few chemical properties are reported. However, it is unclear how these properties were selected for benchmarking, and if the method shows the same performance on other kinds of properties.

Minor points:
- In Table 1 some negative vspace seem to have been used too aggressively.
- In Figure 4a, the choice of using a line plot is questionable, since the x axis is models. A bar plot like  FIg. 5b would have been more appropriate
- The authors should use the extra page to present some results on the generalization to unseen properties and on datasets other than their own, as I think it would strengthen the paper.

**Questions:**

- Please clarify the issues raised on the oracle.
- Why are you using textual instructions rather than a categorical variable (e.g. one-hot encoding for supported properties) for conditioning the diffusion?

---

> ### Author Response · Authors · 2025-11-23
> **Response to Reviewer F1Hf**
>
> We sincerely thank the reviewer for the positive evaluation. We are glad that the paper was found **well-written** and the empirical results **very impressive**, with MolEditRL achieving **state-of-the-art performance** across multiple datasets including C-MuMOInstruct. Below, we address each concern in detail.
>
>
> ### **(W1, W2, Q1) Oracle Choice and Robustness**
>
> Although RL fine-tuning requires an oracle to compute rewards, MolEditRL does not rely on a perfect or deterministic predictor; it remains effective even under noisy oracle conditions. To verify this, we injected controlled noise by randomly flipping oracle outputs at different noise levels (0–0.2). As shown in **Table R1**, **MolEditRL remains stable under all noise settings**, with only minor changes in validity and accuracy, demonstrating strong robustness to imperfect reward signals. These analyses have been provided in Appendix M in the revised paper.
>
> For oracle selection, each textual instruction $S$ is paired with a predefined reward function that corresponds to the described editing goal (e.g., “increase LogP” → LogP oracle). No additional modeling assumptions are required.
>
>
> **Table R1: Robustness to Noisy Oracle Across Different Noise Levels**
> |Task| Noise Level | Validity |Acc$_{\text{all}}$ (TS≥0.65)|Acc$_{\text{all}}$ (MCS≥0.6)|Acc$_{\text{all}}$ (GED≤4)|FCD↓|
> |-------------|------|----------|---------------------------|--------------------|--------------------|------|
> | LogP ↑ | 0.0 |  **0.976** | **0.462** | 0.498 | 0.214 | 7.812 |
> || 0.05 |  0.920 | 0.457 | 0.508 | **0.220** | **7.529** |
> || 0.1 |  0.944 | 0.460 | **0.514** | 0.219 | 7.582 |
> || 0.15 |  0.924 | 0.454 | 0.488 | 0.216 | 7.925 |
> || 0.2 |  0.914 | 0.448 | 0.472 | 0.212 | 8.215 |
> | SA ↓| 0.0  | **0.988** | 0.608 | **0.680** | **0.258** | **6.735** |
> || 0.05 |  0.980 | **0.618** | 0.658 | 0.236 | 7.174 |
> || 0.1 |  0.986 | 0.612 | 0.668 | 0.236 | 7.146 |
> || 0.15 |  0.984 | 0.618 | 0.672 | 0.238 | 7.055 |
> || 0.2 |  0.980 | 0.592 | 0.666 | 0.242 | 7.172 |
>
> ---
>
> ### **(W3, W4.3) Property Selection and Generalization**
>
> Our experiment already covers **18 diverse molecular properties** drawn from both our constructed dataset and the public C-MuMOInstruct dataset. These properties span **drug-likeness, physicochemical attributes, ADMET-related indicators, and biological activities**, and are all commonly used in molecular optimization benchmarks.
>
> We have evaluated MolEditRL on **unseen properties and external datasets**; these results were previously in the appendix. As suggested, we have now moved both sets of experiments to page 8 of the revised main paper.
>
> ---
>
> ### **(Q2) Advantages of Natural Language Instructions over Categorical Encodings**
>
> We use **textual instructions** rather than a categorical (e.g., one-hot) vector because they offer the flexibility needed for **generalization and fine-grained molecular editing**. A one-hot scheme fixes the dimensionality of the conditioning vector and cannot meaningfully represent **new properties or detailed structural operations**, whereas natural language can directly specify the desired modification.
>
> To illustrate this, we fine-tuned the pretrained model for 500 steps using **complex, localized editing prompts** that cannot be encoded with a simple categorical variable. As shown in Table R2, MolEditRL accurately executes these explicit structural edits, achieving high Functional Group Editing Success and strong overall performance across all tasks. More experiments can be found in Appendix Section L of the revised manuscript.
>
> * P1. “Remove a CO₂H group from {SMILE} and decrease its H-bond donor characteristics.”
> * P2. “Add an additional amide fragment to {SMILE} to increase its molecular weight.”
>
>
> **Table R2: Performance on complex localized editing instructions**
> |Prompt|Model|Validity|FG Editing Success|Acc$_{\text{all}}$(TS≥0.65)|FCD|
> |-|-|-|-|-|-|
> |P1|Reinvent4|0.656|0.252|0.078|15.77|
> ||MolEditRL|**0.970**|**0.810**|**0.572**|**7.01**|
> |P2|Reinvent4|0.512|0.328|0.102|14.58|
> ||MolEditRL|**0.968**|**0.798**|**0.360**|**8.98**|
>
> ---
>
>
> ### **(W4.1, W4.2) Presentation Issues: Table and Figure Formatting**
>
> We thank the reviewer for pointing out these presentation issues. The formatting issue in Table 1 has been fixed in the revised version.
> For Figure 4(a), we agree that a bar plot is more appropriate since the x-axis lists discrete models; the revised version has replaced the line plot with a bar chart.

---

> > ### Comment · Reviewer_F1Hf · 2025-11-26
> >
> > W3, W4, Q2: thanks for addressing my comments. The results look quite convincing.
> >
> > W1, W2, Q1: it's still unclear to me how the pairing from text to "predefined reward function" is chosen. Is this hard-coded? What happens if the natural language prompt is something for which you currently have no oracle?
> > Robustness to noise is quite convincing though.

---

> > > ### Author Response · Authors · 2025-11-27
> > > **Response to Reviewer F1Hf**
> > >
> > > We thank the reviewer for the follow-up questions and the opportunity to clarify how textual instructions are paired with reward oracles.
> > >
> > > 1. How text instructions are mapped to reward oracles
> > >
> > > Each training sample has a structured **task label** specifying which property to optimize (e.g., LogP↑, SA↓). During diffusion pretraining, the natural language instruction $S$ is **jointly encoded with the molecular graph** and directly guides how the model performs the edit.
> > >
> > > For the RL stage, however, **oracle selection is not based on parsing the text.** The corresponding **task label** deterministically selects the reward function (e.g., LogP↑ → LogP oracle), following the standard practice in molecular RL benchmarks (REINVENT, GCPN, MolDQN), where each task has a well-defined evaluator.
> > >
> > > ---
> > >
> > > 2. What if an instruction corresponds to a property with no oracle?
> > >
> > > If a new property oracle is available (e.g., RDKit/TDC metric or an ML predictor), we simply define **a new task label and reward function**. MolEditRL can then be fine-tuned on random molecules without requiring paired editing data (see Section 5 of the revised paper for BBBP/HIA/hERG results).
> > >
> > > If **no oracle exists at all**, then no reward can be computed, as is the case for any RL-based molecular optimizer. In this situation, MolEditRL can still produce **structure-preserving edits** through its diffusion prior, but it cannot guarantee improvement on the unsupported property.

---

### Official Review · Reviewer_zGL2 · 2025-10-30

**Soundness:** 3
**Presentation:** 3
**Contribution:** 3
**Rating:** 6
**Confidence:** 3

**Summary:**

This paper proposes **MolEditRL**, a framework combining discrete graph-diffusion and reinforcement learning (RL) for structure-aware molecular editing. It first uses a discrete diffusion model to learn to reconstruct molecules given editing instructions and source molecules, and then fine-tunes via an RL objective with property-based rewards and a KL-regularizer to enforce structural fidelity.
Experiments cover a new editing dataset (“MolEdit-Instruct”), single- and multi-property editing, and comparisons to state-of-the-art baselines, showing substantial improvements in editing success, scaffold preservation, and parameter efficiency.

**Strengths:**

The reviewer acknowledges the following contributions of the paper:

 **Novelty and Technical Contribution**

- The integration of discrete diffusion and reinforcement learning for molecular editing (not just generation) is an original and well-motivated application.

- The KL-regularized RL fine-tuning introduces a principled mechanism to balance property optimization with structural preservation, addressing the classic trade-off between exploration and validity.

- A particularly notable technical insight is how the authors address the high variance problem of rewards that appear only at the final step of the diffusion trajectory. By rewriting the reverse transition using x_{0}-parameterization and approximating gradients through a reward-weighted cross-entropy loss, the algorithm shows a more stable and computationally efficient estimator.

**Dataset Contribution**

The newly released MolEdit-Instruct dataset fills a key gap for large-scale, instruction-based molecular editing tasks and is a valuable contribution to the research community.

**Breadth of Experiments**

- The authors conduct comprehensive experiments covering single-property, multi-property, and cross-dataset generalization scenarios (both with the proposed dataset and C-MuMOInstruct (appendix)). The methods are compared against a variety of baselines, e.g., Drug-assist.

- The obtained results show gains in both editing success rate and structural similarity to the source molecule.

**Weaknesses:**

**Incremental Nature of Some Components**:

- While the integration of discrete diffusion and RL is novel, both components themselves (diffusion for graph generation and RL for property optimization) have been individually explored before in molecular design. The novelty thus lies more in **their combination and adaptation** rather than in a fundamentally new paradigm.

- In the meantime, the Reviewer found another paper [1] shares highly similar ideas with this work, i.e., using diffusion to generate a new updated molecule and then regularized by reinforcement by the PPO method. Though it should not count this work as identical [1] because this one was released after the ICLR deadline. Instead, the reviewer suggests that the author include this paper in the related works and discuss it.

**Limited Discussion on Computational Cost and Scalability**

- The paper reports efficiency in terms of parameter count but provides limited analysis of training/inference cost, scaling to larger biomolecules, or real-time editability.

- Reinforcement learning fine-tuning may still be computationally expensive due to oracle calls and multi-step diffusion.

**Ablation Gaps on Graph-Specific Components**

- Although the authors conduct extensive ablations, there is no explicit comparison showing the benefit of the structure-aware attention mechanism versus a standard transformer without graph bias. The structural-aware self-attention is emphasized as one of the selling points in the paper; thus, a deeper analysis of this property is important.

**Clarity and Accessibility**

- The methodology section is dense and mathematically detailed, which could challenge readers from a non-ML background. A higher-level intuition or illustrative example could improve accessibility.

- It would also be beneficial to discuss potential limitations, such as failure cases (e.g., when instructions are ambiguous or contradict structural constraints).

[1] Kaech, Benno, et al. "Refine Drugs, Don't Complete Them: Uniform-Source Discrete Flows for Fragment-Based Drug Discovery." Arxiv 2025

**Questions:**

Q1. In section 3.4, equation (8), how can the function $r(.)$ be implemented? Is this an oracle by the RDKit tool?

Q2. In section 4.5, the effect of different RL training schemes is discussed. What does this statement mean: "while GDPO leverages **x0-parameterization** to optimize ***only the final output*** " and "MolEditRL introduces KL-regularized optimization ***over the entire diffusion process*** "?

Does this mean that GDPO is also having the reward for the final step? Then how is GDPO different from MolEditRL in this case, given that both methods apply the same x0-parameterization techniques described in Equations (12-14)?

Other questions: please check my weakness section above regarding running time and the missing ablation study on the structure-aware module inside the transformer module.

---

> ### Author Response · Authors · 2025-11-23
> **Response to Reviewer zGL2 (1/2)**
>
> We appreciate the reviewer’s recognition of our work as an **original and well-motivated application**. We are encouraged by the positive assessment of our **balance between property optimization and structural preservation**, and the **stability and efficiency** of our estimator. Below, we address each concern in turn.
>
> ### **(W1) Contribution and Relation to Concurrent Work [1]**
>
>
> Although diffusion models and RL have been individually explored in molecular design, our work contributes a **distinct editing-focused framework**: (i) a **structure-conditioned discrete diffusion model** that edits molecules with explicit structural preservation; (ii) a **KL-regularized RL scheme** that optimizes properties while maintaining fidelity to the source scaffold; and (iii) a **unified instruction-driven interface** enabling fine-grained, localized edits—not de novo generation. These components jointly support a molecular editing paradigm that is not addressed by existing diffusion- or RL-only approaches.
>
>
> Regarding the concurrent work [1], we thank the reviewer for pointing it out. While both methods combine diffusion/flow models with RL, their **problem settings and technical designs differ substantially**. MolEditRL focuses on **instruction-guided, structure-preserving editing of complete molecules**, whereas InVirtuoGen [1] performs **fragment-based de novo generation** without explicit structural constraints, starting from molecular fragments. We have cited and discussed this work in the Related Works section of the revised manuscript.
>
> [1] Kaech, Benno, et al. “Refine Drugs, Don’t Complete Them: Uniform-Source Discrete Flows for Fragment-Based Drug Discovery.” arXiv, 2025.
>
> ---
>
> ### **(Q1) Implementation of Reward Function in Equation (8)**
>
>
> The function in Eq. (8) is implemented using property oracles provided by **RDKit [2] and TDC [3]**, as stated in the first paragraph of Section 3.4. RDKit provides reliable implementations of common molecular properties (e.g., LogP, QED, similarity scores), while TDC offers standardized predictors for additional therapeutic-relevant properties.
>
> [2] Bento, A. Patrícia, et al. "An open source chemical structure curation pipeline using RDKit." Journal of Cheminformatics 12.1 (2020): 51.
>
> [3] Huang, Kexin, et al. "Therapeutics data commons: Machine learning datasets and tasks for drug discovery and development." arXiv preprint arXiv:2102.09548 (2021).
>
> ---
>
>
>
> ### **(Q2) Comparison between GDPO and MolEditRL**
>
>
> When we say “GDPO leverages x₀-parameterization to optimize only the final output”, we mean that GDPO rewrites the policy gradient so that **all updates depend solely on the final generated molecule $G_0$**. Its learning signal comes only from the terminal reward, and **the intermediate denoising steps receive no explicit constraints**.
>
> By contrast, when we say “MolEditRL introduces KL-regularized optimization over the entire diffusion process”, we mean that MolEditRL adds **per-timestep KL regularization** to keep the updated policy aligned with the pretrained diffusion prior. Thus, although MolEditRL also uses a terminal reward and the same x₀-parameterization trick, **its optimization influences multiple diffusion steps rather than only the final one**. This KL-over-steps design stabilizes learning and preserves structural fidelity, which is essential for multi-task molecular editing.
>
> ---
>
> ### **(W3, Q3) Ablation Study on Structure-Aware Attention Mechanism**
>
>
> We conducted an ablation comparing models **with and without the structure-aware attention bias** during both pretraining and fine-tuning (**Table R3**).
>
> During pretraining alone, the improvements in accuracy and FCD are modest. However, after reinforcement learning fine-tuning, the advantage becomes much more substantial. This indicates that while structural bias provides useful inductive priors during pretraining, its main benefit emerges during fine-tuning when **explicit graph constraints** guide the model toward chemically valid and scaffold-preserving edits. These experiments have been added to Appendix H in the revised manuscript.
>
>
> **Table R3: Ablation Study on Structure-Aware Attention Mechanism Across Pretraining and Fine-tuning Stages.**
>
> |Task|Setting|Validity|Acc$_{\text{all}}$(TS≥0.65)|Acc$_{\text{all}}$(MCS≥0.6)|Acc$_{\text{all}}$(GED≤4)|FCD↓|
> |-|-|-|-|-|-|-|
> |LogP ↑|Pretrain w/o Structure Bias|0.744|0.176|0.208|0.182|13.714|
> ||Pretrain w/ Structure Bias|0.758|0.316|0.232|0.196|11.896|
> ||Finetune w/o Structure Bias|0.890|0.212|0.266|0.213|12.486|
> ||Finetune w/ Structure Bias|**0.976**|**0.462**|**0.498**|**0.218**|**7.812**|
> |SA ↓|Pretrain w/o Structure Bias|0.836|0.196|0.224|0.128|12.364|
> ||Pretrain w/ Structure Bias|0.842|0.213|0.256|0.195|10.522|
> ||Finetune w/o Structure Bias|0.904|0.412|0.468|0.176|8.452|
> ||Finetune w/ Structure Bias|**0.988**|**0.608**|**0.680**|**0.258**|**6.735**|

---

> ### Author Response · Authors · 2025-11-23
> **Response to Reviewer zGL2 (2/2)**
>
> ### **(W2, Q3) Computational Cost, Scalability, and Runtime Analysis**
>
> **1. Computational Cost:**
>
> As shown in **Table R2**, we evaluate MolEditRL on HIA optimization, a **property entirely excluded from pretraining**, using a **5,000 oracle query budget** on an A6000 server. Traditional RL baselines (GCPN, MolDQN, REINVENT4) fail to achieve meaningful performance under the same budgets, while MolEditRL **benefits from the pretrained structure-aware diffusion prior and adapts efficiently**. Although MolEditRL requires a one-time fine-tuning stage, this cost yields performance significantly better than both RL-from-scratch and LLM-based methods without fine-tuning.
>
>
> **2. Scalability and Real-time editing:**
>
> MolEditRL handles drug-like molecules (<128 atoms/tokens) and naturally scales to larger structures: the RoBERTa backbone supports sequences up to 512 tokens with no architectural changes. Extending to larger biomolecules (e.g., peptides, macrocycles) would primarily require vocabulary expansion.
> Besides, MolEditRL runs at **1.39 s/sample** with zero oracle calls at inference, making it suitable for interactive molecular editing workflows. More inference and computational efficiency results are provided in Appendices O and P.
>
>
> **Table R2: Computational Efficiency and Editing Performance Comparison Across Baselines.**
> |Method|Validity|Acc$_{\text{all}}$(TS≥0.65)|Acc$_{\text{all}}$(MCS≥0.6)|Acc$_{\text{all}}$(GED≤4)|FCD↓|Finetune Time (min)|Inference Time (s/sample)|
> |-|-|-|-|-|-|-|-|
> |GCPN|0.858|0.000|0.000|0.029|20.260|32.92|0.78|
> |MolDQN|1.000|0.003|0.142|0.045|13.152|35.76|4.73|
> |Reinvent4|0.835|0.124|0.196|0.139|13.786|40.51|0.85|
> |BioT5|**1.000**|0.000|0.004|0.000|15.782|N/A|0.76|
> |DrugAssist|0.976|0.273|0.291|0.127|8.231|N/A|1.52|
> |Gellm$^4$o-C\_L|0.909|0.222|0.244|0.114|8.648|N/A|4.56|
> |MolEditRL|0.958|**0.466**|**0.490**|**0.226**|**7.964**|58.14|1.39|
>
>
> ---
>
>
>
>
> ### **(W4) Clarity, and Discussion of Limitations**
>
> We appreciate the reviewer’s suggestion. In the revised version, we have expanded Appendix S to analyze additional failure cases, particularly those arising from conflicting natural-language instructions or scenarios where no valid oracle is available for reward computation.

---

> > ### Comment · Reviewer_zGL2 · 2025-11-25
> > **Response to Author**
> >
> > Thanks to the authors for providing additional experiments to address my questions. I will maintain my positive rating. If the paper be accepted, please include these discussions into the final version.

---

### Official Review · Reviewer_rzAV · 2025-10-31

**Soundness:** 4
**Presentation:** 3
**Contribution:** 3
**Rating:** 8
**Confidence:** 4

**Summary:**

This work proposes a molecular editing framework based on discrete diffusion and reinforcement learning. Specifically, authors design a structure-preserving attention mechanism with a bias term encoding adjacency information, to preserve the structural integrity of the template molecule during editing. The molecular generation process is implemented as a pre-trained discrete diffusion model, which is then fine-tuned with editing success and validity rewards on pair molecular edit data. The framework is then tested on several molecular design tasks involving multiple properties and outperforms the baselines on all metrics.

**Strengths:**

- The proposed structure-aware attention mechanism offers a potential solution to the complex problem of conditioning the molecular generation with structural constraints.

- The joint encoding of both instruction text and molecular structure achieves good alignment between text description and the molecular space.

- The performance of the proposed framework significantly outperforms all baselines by several tasks and metrics. Also, detailed visualizations and interpretations of the generative results as well as ablation studies are provided.

- The authors also provide a comprehensive dataset for molecular edit paired covering a wide range of properties.

**Weaknesses:**

- The RL setting relies on ready-to-use and fast property calculations, so generalizability to rare or unseen properties is limited (especially for protein binding).

- Some specific questions on methods and clarity. See Questions

**Questions:**

- Eq3: in the early denoising steps, as $E_{tgt}$ is inaccurate and noisy, how to prevent it from misinforming the denoising of the atoms?

- Since the language used for the editing instruction is well-structured and only the property type (and potential numbers) matter, how much does the language model backbone really contribute?

- Is the structure-preserving behavior more achieved by the structure-aware attention, or the explicit KL divergence objective during RL? I.e. what are the raw similarity values in the ablation study results?

---

> ### Author Response · Authors · 2025-11-23
> **Response to Reviewer rzAV**
>
> We thank the reviewer for the positive feedback, especially the recognition of our “**structure-aware attention mechanism**,” the “**good alignment between text and molecular space**,” and the “**significant outperformance over baselines**.” We address the remaining points below.
>
>
> ### **(Q1) Handling Noisy Edge Information in Early Denoising Steps (Equation 3)**
>
>
> We agree that $E_{tgt}$ is noisy in early denoising steps (and fully masked at $t=T$ during inference). **This is inherent to discrete diffusion models, and the network is trained to handle varying noise levels.** As shown in Equation 3, $E_{tgt}$ is injected only at $l = 0$. For all deeper layers ($l > 0$), the model relies on the learned attention weights $A^{l-1}$, allowing the multi-layer Transformer to progressively correct the initial noise.
>
> In addition, $E_{src}$ remains accurate at every layer and timestep, providing a stable structural prior; since source and target molecules are structurally similar in editing tasks, this strongly guides early denoising.
>
> ---
>
> ### **(Q2) Contribution of Language Model Backbone**
>
> Although many existing editing instructions correspond to a property type, using a **RoBERTa language backbone** provides substantially more flexibility. The language encoder enables the model to **generalize to unseen property descriptions** and handle instructions beyond a fixed attribute set. More importantly, as shown in our additional experiments (**Table R1**), the model can follow **complex, localized editing prompts** that cannot be represented using simple categorical inputs. More results are provided in Appendix L of the revised paper.
>
>
> * P1. “Remove a CO₂H group from {SMILE} and decrease its H-bond donor characteristics.”
> * P2. “Add an additional amide fragment to {SMILE} to increase its molecular weight.”
>
>
> **Table R1: Performance on complex localized editing instructions**
> |Prompt|Model|Validity|FG Editing Success|Acc$_{\text{all}}$(TS≥0.65)|FCD|
> |-|-|-|-|-|-|
> |P1|Reinvent4|0.656|0.252|0.078|15.77|
> ||MolEditRL|**0.970**|**0.810**|**0.572**|**7.01**|
> |P2|Reinvent4|0.512|0.328|0.102|14.58|
> ||MolEditRL|**0.968**|**0.798**|**0.360**|**8.98**|
>
> ---
>
> ### **(Q3) Structure-Aware Attention vs. KL Divergence for Structure Preservation**
>
> We conducted an ablation study comparing models **with and without the structure-aware attention bias** during both pretraining and fine-tuning (**Table R2**). The gains during pretraining are modest, indicating that the structural bias mainly acts as an inductive prior. However, after RL fine-tuning, the improvements become much larger.
>
> This suggests that the structural bias establishes the graph-level constraints needed for chemically valid, scaffold-preserving edits, while the **KL regularization further prevents the policy from over-optimizing reward and drifting away from the pretrained distribution**. Together, they jointly produce the strong structure-preserving behavior observed in our full model.
>
>
> **Table R2: Ablation Study on Structure-Aware Attention Mechanism.**
>
> |Task|Setting|Validity|Acc$_{\text{all}}$(TS≥0.65)|Acc$_{\text{all}}$(MCS≥0.6)|Acc$_{\text{all}}$(GED≤4)|FCD↓|
> |------|---------|----------|------------------------------|------------------------------|----------------------------|------|
> |LogP ↑|Pretrain w/o Structure Bias|0.744|0.176|0.208|0.182|13.714|
> ||Pretrain w/ Structure Bias|0.758|0.316|0.232|0.196|11.896|
> ||Finetune w/o Structure Bias|0.890|0.212|0.266|0.213|12.486|
> ||Finetune w/ Structure Bias|**0.976**|**0.462**|**0.498**|**0.218**|**7.812**|
> |SA ↓|Pretrain w/o Structure Bias|0.836|0.196|0.224|0.128|12.364|
> ||Pretrain w/ Structure Bias|0.842|0.213|0.256|0.195|10.522|
> ||Finetune w/o Structure Bias|0.904|0.412|0.468|0.176|8.452|
> ||Finetune w/ Structure Bias|**0.988**|**0.608**|**0.680**|**0.258**|**6.735**|
>
> ---
>
> ### **(W1) Generalizability to Rare/Unseen Properties**
>
> While RL requires property evaluators, our experiments show that MolEditRL is **robust to noisy or imperfect oracles** (Appendix M in the revised paper) and can **adapt to entirely new properties with only a small oracle budget (~5,000 evaluations)** (Appendix N), benefiting from the strong pretrained diffusion prior.
>
> The paper already evaluates **18 diverse properties** across the MolEdit dataset and the public C-MuMOInstruct benchmark, including several tied to **protein binding or inhibition** (DRD2, GSK3β, JNK3, hERG). Moreover, Section 5 of the revised paper includes experiments on **hERG as an unseen property**, further demonstrating MolEditRL’s ability to generalize to protein-binding–related tasks.

---

> > ### Comment · Reviewer_rzAV · 2025-11-28
> >
> > Thanks for the detailed responses. I would like to maintain my current scores.

---

### Official Review · Reviewer_JXjh · 2025-10-31

**Soundness:** 3
**Presentation:** 2
**Contribution:** 2
**Rating:** 2
**Confidence:** 3

**Summary:**

The paper introduces MolEditRL, a framework for structure-preserving molecular editing. The approach is designed to modify a source molecule based on a natural language instruction to optimize for desired chemical properties while maintaining high structural similarity. MolEditRL consists of a two-stage process: first, a discrete graph diffusion model is pretrained to reconstruct target molecules conditioned on both source structures and text prompts. Second, this model is fine-tuned using reinforcement learning (RL) with a KL-regularized objective to explicitly maximize property-related rewards. To facilitate this research, the authors constructed a new large-scale dataset, MolEdit-Instruct. The experimental results show that MolEditRL outperforms several baselines, including large language models, in both editing accuracy and structural fidelity.

**Strengths:**

1. The problem of controlled, structure-aware molecular editing is highly relevant to drug discovery. The paper’s core approach of combining a discrete graph diffusion model with a full-trajectory RL fine-tuning process is compelling. By operating on graph representations, it sidesteps the syntactic instability and representational ambiguity issues inherent in string-based methods, representing a step towards more reliable molecular design.
2. The construction of the MolEdit-Instruct dataset is a valuable contribution to the community. The experimental evaluation is extensive, comparing the proposed method against a strong and relevant set of recent baselines on a diverse range of single- and multi-property tasks.
3. The authors show clear motivation of the design choices, particularly the shift towards graph-based representations and the integration of RL for precise optimization.

**Weaknesses:**

1. The method for encoding the natural language instruction is not sufficiently detailed in the main text. Section 3.1 states that instruction tokens, source atoms, and target atoms are embedded and concatenated into a unified sequence. However, it is unclear how these instruction tokens are specifically embedded and integrated.
2. A well-known failure mode for generative models fine-tuned with RL for molecular optimization is "oracle hacking," where the model learns to generate molecules that achieve high scores from the property predictor but are chemically invalid, unstable, or non-synthesizable. While the KL-regularization term is likely intended to mitigate this by constraining the policy to stay close to the pretrained distribution, the paper does not explicitly discuss this critical issue or demonstrate how its method avoids it. Furthermore, while the paper includes REINVENT4 as a baseline, it misses a broader comparison with other established RL-based molecular optimization frameworks (e.g., MolDQN, GCPN) that could provide a more direct assessment of its RL component.
3. The experiments do not showcase scenarios that leverage the true power and flexibility of natural language, such as describing complex, localized edits or conveying chemical intuition. Without such examples, the language-conditioning aspect feels underexploited, and its contribution to the model's performance is unclear. The current property editing description is too simple.
4. The performance of language-conditioned models is often highly sensitive to the specific phrasing of the input prompt. The paper presents a single, fixed prompt template for each task (Table 6) but provides no analysis or ablation study on how variations in these prompts might affect editing outcomes. This is an omission, as it leaves the robustness of the instruction-following capability unevaluated.
5. The framework's success hinges on the quality of the distribution learned during the first pre-training stage. A high-quality pretrained model should provide a strong prior for the RL fine-tuning, ensuring that the search for high-reward molecules begins from a space of valid and realistic structures. However, the paper does not provide any direct, quantitative evaluation of the stage-one diffusion model (e.g., reconstruction accuracy, validity, and FCD on a held-out set before any RL fine-tuning). This makes it difficult to assess whether the strong final performance is due to a superior pretrained base model or solely the effectiveness of the RL stage.

**Questions:**

1. Oracle Hacking. Could you please explicitly discuss the problem of "oracle hacking" and elaborate on how the KL-regularization in your RL objective functions maintains chemical realism and prevents the model from generating non-synthesizable molecules?
2. Justifying Natural Language. Can you provide a more compelling justification for using natural language instructions over structured inputs for the current set of tasks? Better yet, could you demonstrate the model's capabilities on more complex editing tasks where the flexibility of natural language is genuinely advantageous?
3. Evaluating Pre-training. Could you provide standalone evaluation metrics for the discrete diffusion model after the pre-training stage is complete (but before RL)? Metrics like validity, reconstruction accuracy, and FCD against the ground-truth distribution would help disentangle the contributions of the two training stages.
4. Instruction Encoding Details. Could you please clarify the precise mechanism for embedding the instruction tokens and integrating them with the graph representations in the model's input?

---

> ### Author Response · Authors · 2025-11-22
> **Response to Reviewer JXjh (1/3)**
>
> We thank the reviewer for the constructive feedback. We appreciate the recognition that our approach is **highly relevant to drug discovery**, effectively **avoids syntactic instability in string-based methods**, and is supported by **a valuable MolEdit-Instruct dataset** and **extensive experiments**. Below, we address each concern with additional analyses and clarifications. These updates will be reflected in the revised manuscript.
>
> ### **(W1, Q4) Instruction Encoding and Integration Mechanism**
>
>
> **1. Unified Tokenization of Instructions and Molecular Graphs**
>
> Each training example consists of a natural language instruction $S = [s_1, \dots, s_n]$, a source molecule graph $G_{\text{src}}$, and a target molecule graph $G_{\text{tgt}}$. Both molecular graphs are serialized into linear sequences of atom tokens, where each atom is represented by its discrete type (e.g., $\langle C \rangle$, $\langle O^{+} \rangle$, $\langle Cl \rangle$). The instruction and both molecular sequences are then concatenated into a single unified input: $\mathbf{x} = [S;\ G_{\text{src}};\ G_{\text{tgt}}].$
>
> **2. Shared Token Embedding Space**
>
> All tokens—whether from natural language or molecular graphs—are embedded into the same vector space using a unified embedding matrix $E \in \mathbb{R}^{|\mathcal{V}| \times d_h}$, where the vocabulary size $|\mathcal{V}| = 51{,}933$ extends RoBERTa's original vocabulary with molecular-specific tokens. For each token $x_i$, the input embedding is computed as: $h_i = E(x_i) + E_{\text{pos}}(i)$, where $E_{\text{pos}}(i)$ is the positional embedding.
>
> **3. Unified Sequence Encoding**
>
> The concatenated token sequence is fed into the Structure-Aware Transformer encoder to form the initial hidden representation, corresponding to Equation (1) in the paper:
> $$h^0 = \big[h^{\text{inst}}_1, \dots, h^{\text{inst}}_n,\ h^{\text{src}}_1, \dots, h^{\text{src}}_{L_{\text{src}}},\ h^{\text{tgt}}_1, \dots, h^{\text{tgt}}_{L_{\text{tgt}}}\big].$$
>
> This formulation allows the model to learn cross-modal dependencies: instruction tokens can attend to molecular atoms to identify editing targets, while molecular atoms can attend to instructions to understand optimization objectives.
>
> **4. Injection of Graph Structure via Attention Bias**
>
> To preserve molecular topology, we inject graph connectivity information directly into the attention mechanism. At each Transformer layer $l$, the pre-softmax attention score between tokens $i$ and $j$ is computed as described in Equation (2) of the paper:
> $$\hat{A}^l_{i,j} = \frac{1}{\sqrt{d_k}}(h^l_i W_Q)(h^l_j W_K)^\top + b^l_{i,j}$$
> The bias term $b^l_{i,j}$ encodes structural relationships:
> - For pairs of source-graph atoms, $b^l_{i,j}$ incorporates bond connectivity from the source adjacency matrix across all layers
> - For target-graph atoms, adjacency information is injected at the first layer and propagates through subsequent layers via attention (Equations (3) and (4) of the paper)
> - Text tokens retain zero bias, ensuring graph topology only influences attention among molecular atoms
>
> Through this mechanism, the model maintains a unified embedding space while explicitly enforcing graph-structural constraints during attention computation.

---

> ### Author Response · Authors · 2025-11-22
> **Response to Reviewer JXjh (2/3)**
>
> ### **(W5, Q3) Evaluating Pre-training**
>
> Figure 5 in Section 4.6 of the paper already illustrates the performance gap between pretraining and fine-tuning, and **Table R1** now provides a more detailed quantitative analysis.
> The three similarity metrics **MACCS_FTS**, **RDK_FTS**, and **Morgan_FTS** measure **fingerprint-based structural similarity** and collectively reflect how well the model preserves molecular topology.
>
> The pretrained diffusion model alone achieves **strong structural fidelity**, with highly stable MACCS, RDK, and Morgan scores across all tasks, showing that it learns a **high-quality structural prior** without any reward optimization. These similarity metrics remain largely unchanged after RL, indicating that **fine-tuning does not distort the learned structural distribution**.
> In contrast, RL fine-tuning substantially **improves both validity and property-aligned accuracy**, and all tasks exhibit large **reductions in FCD**, demonstrating that RL moves the generated distribution **closer to real molecules** while enhancing property optimization. The validity improvements further show that KL-regularized RL effectively refines the pretrained model without compromising its structural prior.
>
> Therefore, the strong final performance of MolEditRL arises from the complementary **contributions of both stages**: diffusion pretraining provides a reliable and realistic structural foundation, and RL fine-tuning delivers targeted property improvements on top of this high-quality prior.
>
>
>
> **Table R1: Performance comparison between pretrained diffusion model and RL fine-tuned model**
>
> |Task||Validity|Acc$_{\text{all}}$(TS≥0.65)|Acc$_{\text{valid}}$(TS≥0.65)|MACCS_FTS|RDK_FTS|Morgan_FTS|FCD↓|
> |-|-|-|-|-|-|-|-|-|
> |QED↑|Pretrain|0.812|0.460|0.567|0.832|**0.736**|**0.684**|10.518|
> ||Finetune|**0.974**|**0.604**|**0.620**|**0.834**|0.735|0.667|**7.678**|
> |Haccept↑|Pretrain|0.750|0.266|0.355|0.789|0.688|0.609|9.489|
> ||Finetune|**0.968**|**0.484**|**0.500**|**0.798**|**0.691**|**0.623**|**7.316**|
> |LogP↑|Pretrain|0.758|0.316|0.417|0.776|0.672|**0.643**|11.896|
> ||Finetune|**0.964**|**0.578**|**0.599**|**0.795**|**0.682**|0.620|**7.012**|
> |DRD2↑|Pretrain|0.850|0.220|0.259|**0.796**|**0.698**|**0.637**|11.194|
> ||Finetune|**0.966**|**0.308**|**0.319**|0.791|0.677|0.629|**9.389**|
> |MW↑|Pretrain|0.774|0.142|0.184|0.783|0.651|0.562|10.890|
> ||Finetune|**0.960**|**0.404**|**0.421**|**0.805**|**0.673**|**0.576**|**6.588**|
>
> ---
>
> ### **(W2, Q1) Oracle Hacking and Baseline Comparison with RL-based Methods**
>
>
> Table R2 presents a comparison against three RL-based molecular optimization frameworks (GCPN, MolDQN, and REINVENT4) under a strict oracle budget of 5,000 queries on the HIA↑ task, a property completely excluded from pretraining. We evaluate **synthesizability** (SA ≤ 3.0), **drug-likeness** (QED ≥ 0.5), and **oral bioavailability potential** (satisfaction of all Lipinski rules). MolEditRL not only surpasses all baselines but even improves these metrics relative to the source molecules.
> These results highlight the importance of our **KL-regularized objective**: by constraining policy updates toward the pretrained diffusion prior, the model remains anchored to a distribution of valid, realistic, and structurally faithful molecules throughout fine-tuning. This prevents the policy from drifting into chemically implausible regions of chemical space that merely exploit predictor weaknesses.
>
>
> **Table R2: Comparison with RL-based methods on chemical realism metrics**
>
> |Method|Validity|Acc$_{\text{all}}$ (TS≥0.65)|FCD↓|Is_Synthesizable|Is_Druglike|Lipinski_RO5|
> |-|-|-|-|-|-|-|
> |Source Molecule|1.000|-|-|0.278|0.376|0.520|
> |GCPN|0.858|0.000|20.260|0.047|0.153|0.205|
> |MolDQN|**1.000**|0.003|13.152|0.092|0.178|0.246|
> |Reinvent4|0.835|0.124|13.786|0.278|0.370|0.428|
> |MolEditRL|0.958|**0.466**|**7.964**|**0.460**|**0.681**|**0.838**|

---

> ### Author Response · Authors · 2025-11-22
> **Response to Reviewer JXjh (3/3)**
>
> ### **W4 - Prompt Sensitivity Analysis**
>
>
> We would like to clarify that the **MolEdit-Instruct** pretraining dataset contains a *diverse set of paraphrased instructions* for every editing objective. The paper presents only one representative template for illustration, but the actual corpus includes **many alternative linguistic formulations** for the same property modification. This diversity exposes the model to variations in **syntax**, **vocabulary**, and **semantic emphasis** well before any fine-tuning.
>
> As shown in Table R3, we further evaluate the model using **five distinct SA↓ prompts**:
>
> * P1. “Reduce the synthetic accessibility of molecule {SMILE}.”
> * P2. “Make this molecule {SMILE} easier to synthesize.”
> * P3. “Adjust the structure of {SMILE} to lower its synthetic complexity.”
> * P4. “Modify {SMILE} so that its overall synthetic accessibility score decreases.”
> * P5. “Transform the molecule {SMILE} into a form that is simpler to assemble synthetically.”
>
> These results demonstrate that the model **does not rely on any specific phrasing strategy**. Instead, it leverages the **broad linguistic variability** introduced during pretraining to form a **prompt-robust semantic representation**, enabling consistent editing behavior even when users employ different natural-language descriptions for the same molecular objective.
>
>
> **Table R3: Prompt sensitivity analysis on SA↓ task with diverse instruction formulations**
>
> |Prompt|Validity|Acc$_{\text{all}}$(TS≥0.65)|Acc$_{\text{all}}$(MCS≥0.6)|Acc$_{\text{all}}$(GED≤4)|FCD↓|
> |-|-|-|-|-|-|
> |P1|**0.986**|0.636|**0.696**|0.244|7.107|
> |P2|0.974|0.604|0.618|0.253|7.307|
> |P3|0.979|**0.652**|0.636|0.224|7.157|
> |P4|0.984|0.629|0.614|0.244|**7.052**|
> |P5|0.980|0.646|0.642|**0.256**|7.339|
>
>
> ---
>
>
> ### **(W3, Q2) Flexibility of Natural Language Instructions**
>
>
> We fine-tuned the pretrained model for 500 steps using **complex, localized editing instructions** rather than simple property prompts:
>
> * P1. “Remove a CO₂H group from {SMILE} and decrease its H-bond donor characteristics.”
> * P2. “Add an additional amide fragment to {SMILE} to increase its molecular weight.”
> * P3. “Remove an aromatic ring from {SMILE} to lower its structural complexity and improve synthetic accessibility.”
> * P4. “Reduce the synthetic accessibility of molecule {SMILE}.”
> * P5. “Eliminate a CONH unit from {SMILE} to make the scaffold easier to assemble.”
>
> These instructions specify **explicit structural operations**, including functional group removal, fragment addition, and scaffold modification. As shown in Table R4, **FG Editing Success** measures the proportion of generated molecules that correctly perform the required functional group edit, while **Acc(_\text{all})(TS ≥ 0.65)** additionally requires structural similarity and successful property change.
>
> MolEditRL maintains **high performance across all prompts**, while preserving synthesizability and drug-likeness. These results show that the model accurately interprets **fine-grained chemical instructions** and performs **localized edits** that cannot be expressed through scalar property targets or structured numeric inputs. This demonstrates that natural language is not merely an interface for simple property tuning but enables **precise, interpretable, and expressive structural control**.
>
>
>
> **Table R4: Performance on complex localized editing instructions demonstrating natural language flexibility**
>
> |Prompt|Model|Validity|FG Editing Success|Acc$_{\text{all}}$(TS≥0.65)|FCD|MACCS_FTS|Is_Synthesizable|Is_Druglike|Lipinski_RO5|
> |-|-|-|-|-|-|-|-|-|-|
> |-|Source Molecule|1.000|-|-|-|-|0.278|0.376|0.520|
> |P1|Reinvent4|0.656|0.252|0.078|15.77|0.428|0.157|0.267|0.297|
> ||MolEditRL|**0.970**|**0.810**|**0.572**|**7.01**|**0.789**|**0.396**|**0.655**|**0.787**|
> |P2|Reinvent4|0.512|0.328|0.102|14.58|0.387|0.185|0.252|0.389|
> ||MolEditRL|**0.968**|**0.798**|**0.360**|**8.98**|**0.785**|**0.332**|**0.574**|**0.884**|
> |P3|Reinvent4|0.606|0.320|0.112|13.28|0.385|0.157|0.188|0.246|
> ||MolEditRL|**0.994**|**0.744**|**0.358**|**7.98**|**0.784**|**0.328**|**0.505**|**0.874**|
> |P4|Reinvent4|0.632|0.340|0.126|12.86|0.379|0.104|0.185|0.283|
> ||MolEditRL|**0.984**|**0.872**|**0.370**|**7.93**|**0.763**|**0.392**|**0.591**|**0.839**|
> |P5|Reinvent4|0.642|0.234|0.128|12.32|0.365|0.160|0.179|0.204|
> ||MolEditRL|**0.978**|**0.752**|**0.310**|**7.60**|**0.778**|**0.327**|**0.503**|**0.712**|

---

### Official Review · Reviewer_niJm · 2025-11-02

**Soundness:** 3
**Presentation:** 3
**Contribution:** 3
**Rating:** 6
**Confidence:** 3

**Summary:**

This paper presents MolEditRL, a framework for structure-preserving molecular editing that integrates discrete graph diffusion with reinforcement learning. The model first performs discrete diffusion pretraining conditioned on the source molecule and natural-language instruction, then applies RL fine-tuning with property-based rewards and KL regularization to balance optimization and structural fidelity. The authors also introduce MolEdit-Instruct, a large-scale dataset for instruction-based molecular editing. Experiments demonstrate substantial gains over existing approaches in both property optimization and structural similarity metrics with fewer parameters.

**Strengths:**

- Clear writing and strong presentation. The paper is well-structured with clearly outlined motivation, method, and evaluation.
- Technical soundness. The integration of discrete diffusion with RL fine-tuning is implemented carefully and supported by strong experimental performance.
- Comprehensive evaluation. Extensive experiments across single and multi-property editing tasks, with multiple structure-based similarity metrics and FCD, provide convincing evidence.
- Strong empirical results. The method consistently surpasses large language model baselines in both accuracy and structural fidelity while being significantly more parameter-efficient.

**Weaknesses:**

- Missing efficiency analysis. The paper highlights parameter efficiency but does not report oracle-query efficiency compared to baselines. Quantitative analysis of efficiency would strengthen practical claims.
- Limited novelty in the diffusion–RL combination. Similar hybrid ideas have appeared in related areas. The primary contribution lies in adapting within the molecular editing context rather than fundamentally extending these techniques, making the innovation more technical than conceptual.

**Questions:**

1. Hyperparameter sensitivity in RL fine-tuning. Hyperparameter sensitivity in RL fine-tuning. Did the authors conduct ablations on key hyperparameters such as $\beta$, policy-update stride, top-k during sampling, and the 0.2 partial-success reward? How did the authors determine the final values?
2. How are train/test splits constructed to avoid data leakage?
3. I am curious about the inference efficiency of the diffusion model. Could the authors compare the generation efficiency of the proposed method with that of other models?
4. The equation 5 appears to mis-type a $q$.

---

> ### Author Response · Authors · 2025-11-22
> **Response to Reviewer niJm (1/3)**
>
> We sincerely thank the reviewer for the positive evaluation and constructive feedback. We appreciate the recognition of our work's **clear presentation**, **technical soundness**, **comprehensive evaluation**, and **strong empirical results**, particularly that MolEditRL achieves superior **structural fidelity** with significantly **fewer parameters**. Below, we address each concern in detail. These revisions will be incorporated into the updated version of the manuscript.
>
> ### **W1: Efficiency Analysis (Oracle-Query Efficiency)**
>
> As shown in **Table R1**, we compare MolEditRL with three traditional RL-based molecular editing methods (GCPN, MolDQN, REINVENT4) on the task of improving HIA (Human Intestinal Absorption). **This property is entirely excluded from pretraining**, making it a suitable benchmark for evaluating practical oracle-query efficiency under unseen property conditions.
>
> The results demonstrate that MolEditRL consistently outperforms all baseline RL methods across all oracle-query budgets. This advantage primarily arises from the structure-aware discrete diffusion pretraining stage. As a result, MolEditRL starts RL fine-tuning with a strong structure-preserving prior. In contrast, existing RL approaches require oracle evaluations during training and rely on uninformed trial-and-error exploration to discover viable editing strategies, which results in a large number of oracle queries.
>
> **Table R1: Oracle-query efficiency comparison on HIA↑ task across different oracle budgets**
>
> |Oracle Queries|Method|Validity|Acc$_{\text{all}}$ (TS≥0.65)|Acc$_{\text{all}}$ (MCS≥0.6)|Acc$_{\text{all}}$ (GED≤4)|FCD↓|
> |-|-|-|-|-|-|-|
> |1000|GCPN|0.842|0.000|0.000|0.026|22.140|
> ||MolDQN|**1.000**|0.002|0.138|0.086|15.598|
> ||Reinvent4|0.598|0.092|0.148|0.044|14.682|
> ||MolEditRL|0.864|**0.388**|**0.408**|**0.202**|**9.686**|
> |2000|GCPN|0.882|0.000|0.000|0.022|21.175|
> ||MolDQN|**1.000**|0.000|0.122|0.026|14.857|
> ||Reinvent4|0.608|0.104|0.167|0.052|14.236|
> ||MolEditRL|0.898|**0.396**|**0.448**|**0.218**|**9.489**|
> |3000|GCPN|0.878|0.000|0.000|0.016|21.324|
> ||MolDQN|**1.000**|0.002|0.156|0.036|14.828|
> ||Reinvent4|0.711|0.114|0.187|0.080|13.997|
> ||MolEditRL|0.902|**0.430**|**0.472**|**0.216**|**8.609**|
> |4000|GCPN|0.850|0.000|0.000|0.022|20.672|
> ||MolDQN|**1.000**|0.002|0.156|0.032|13.362|
> ||Reinvent4|0.769|0.118|0.192|0.105|13.868|
> ||MolEditRL|0.928|**0.448**|**0.488**|**0.214**|**8.387**|
> |5000|GCPN|0.858|0.000|0.000|0.029|20.260|
> ||MolDQN|**1.000**|0.003|0.142|0.045|13.152|
> ||Reinvent4|0.835|0.124|0.196|0.139|13.786|
> ||MolEditRL|0.958|**0.466**|**0.490**|**0.226**|**7.964**|
>
>
> ---
>
> ### **W2: Novelty**
>
> We appreciate the reviewer’s perspective and would like to clarify that MolEditRL is not a simple adaptation of previously explored diffusion–RL combinations. Instead, it is specifically designed to address the **unique challenges of discrete, structure-constrained molecular editing**. Prior hybrid approaches typically operate in continuous or string-based spaces and lack mechanisms to preserve molecular scaffolds or perform localized graph edits.
>
> In contrast, MolEditRL introduces a **structure-aware discrete diffusion** process tailored to atom–bond space, a **unified text–graph editing architecture** that enforces topological consistency throughout denoising, and a **KL-regularized reinforcement learning objective** that stabilizes optimization while maintaining structural fidelity. These components work together to enable **precise, controllable, and structure-preserving edits** that existing methods cannot support, and they further allow MolEditRL to **generalize to unseen properties with minimal oracle usage**.
>
>
> ---
>
> ### **Q1: Hyperparameter Sensitivity in RL Fine-tuning**
>
> **1. KL Regularization Weight $\beta$**
>
> As shown in Table R2, we conducted ablations on the KL-regularization weight $\beta$ by training models with $\beta \in {0, 0.1, 0.2, 0.3, 0.4, 0.5}$ under a fixed fine-tuning budget of 500 steps. Setting $\beta = 0$ removes the structural prior and leads to unstable or overly aggressive edits, whereas larger $\beta$ values over-constrain the policy and suppress property improvements. Based on these results, we adopt $\beta = 0.1$ as the default setting in all experiments.
>
> **Table R2: Ablation study on KL regularization weight $\beta$**
>
> |Task|$\beta$|Validity|Acc$_{\text{all}}$ (TS≥0.65)|Acc$_{\text{all}}$ (MCS≥0.6)|Acc$_{\text{all}}$ (GED≤4)|FCD↓|
> |-|-|-|-|-|-|-|
> |LogP ↑|0.0|**0.986**|0.278|0.376|0.186|9.142|
> ||0.1|0.976|**0.462**|**0.498**|**0.214**|**7.812**|
> ||0.2|0.920|0.416|0.458|0.196|9.762|
> ||0.3|0.884|0.386|0.414|0.206|9.649|
> ||0.4|0.840|0.350|0.402|0.210|10.812|
> ||0.5|0.842|0.356|0.380|0.194|10.700|
> |SA ↓|0.0|0.982|0.602|**0.700**|0.208|7.295|
> ||0.1|**0.988**|**0.608**|0.680|**0.258**|**6.735**|
> ||0.2|0.942|0.576|0.582|0.186|8.503|
> ||0.3|0.956|0.574|0.576|0.192|8.679|
> ||0.4|0.928|0.562|0.558|0.186|8.900|
> ||0.5|0.918|0.544|0.548|0.194|9.100|

---

> ### Author Response · Authors · 2025-11-22
> **Response to Reviewer niJm (2/3)**
>
> ### **Q1: Hyperparameter Sensitivity in RL Fine-tuning**
>
> **2. Policy Update Stride**
>
>
> As shown in Table R3, we fine-tune models with policy update stride ∈ {1, 2, 3, 4, 5} under an identical budget of 500 steps. Each cell in the table reports Validity / Acc_\text{all} (TS ≥ 0.65) / FCD. The results indicate that stride = 1 provides the fastest and most effective improvements in both property optimization and structural fidelity, consistently achieving the highest accuracy and lowest FCD across training steps. Larger strides decrease update frequency, leading to slower convergence and noticeably weaker performance. Based on these observations, we adopt **stride = 1** as the default setting.
>
>
> **Table R3: Sensitivity analysis of policy-update stride**
>
> |Task|Step|Stride=1|Stride=2|Stride=3|Stride=4|Stride=5|
> |-|-|-|-|-|-|-|
> |LogP ↑|99|0.910/0.436/8.425|0.816/0.326/11.164|0.802/0.334/11.037|0.776/0.332/10.998|0.780/0.290/11.597|
> ||199|0.933/0.442/8.076|0.842/0.318/11.159|0.834/0.358/10.430|0.796/0.346/11.031|0.824/0.344/10.494|
> ||299|0.958/0.438/8.294|0.860/0.370/10.569|0.866/0.368/9.431|0.838/0.372/9.901|0.828/0.348/10.499|
> ||399|0.956/0.456/7.965|0.914/0.364/9.844|0.862/0.356/10.451|0.866/0.370/9.950|0.860/0.358/10.736|
> ||499|**0.976/0.462/7.812**|0.938/0.408/9.095|0.898/0.382/9.884|0.866/0.372/9.803|0.888/0.366/10.348|
> |SA ↓|99|0.984/0.587/7.466|0.848/0.526/9.482|0.844/0.516/9.466|0.780/0.430/11.379|0.788/0.436/10.665|
> ||199|0.982/0.580/7.510|0.872/0.536/9.378|0.876/0.522/9.474|0.818/0.456/10.692|0.834/0.486/10.459|
> ||299|0.978/0.591/7.322|0.916/0.556/8.818|0.904/0.550/9.191|0.862/0.480/10.197|0.896/0.548/8.881|
> ||399|0.971/0.599/7.251|0.946/0.598/8.333|0.910/0.558/8.757|0.904/0.504/9.925|0.904/0.558/8.636|
> ||499|**0.988/0.608/6.735**|0.956/0.592/8.202|0.932/0.512/9.737|0.924/0.530/9.523|0.920/0.574/8.904|
>
>
>
>
> **3. Top-k Sampling Strategy**
>
> As shown in Table R4, we evaluated the top-k parameter during sampling by testing top-k ∈ {5, 10, 15, 20, 25}. The results indicate that performance remains generally stable across this range, with only mild variation in validity and structural accuracy. Smaller top-k values may reduce output diversity, whereas larger values can introduce unnecessary stochasticity. Based on these observations, we adopt **top-k = 15** as the default setting.
>
> **Table R4: Sensitivity analysis of top-k sampling parameter**
>
> |Task|Top-k|Validity|Acc$_{\text{all}}$ (TS≥0.65)|Acc$_{\text{all}}$ (MCS≥0.6)|Acc$_{\text{all}}$ (GED≤4)|FCD↓|
> |-|-|-|-|-|-|-|
> |LogP ↑|5|0.956|0.460|0.462|0.208|7.924|
> ||10|0.950|0.458|0.438|0.202|7.977|
> ||15|**0.976**|**0.462**|**0.498**|0.214|**7.812**|
> ||20|0.946|0.448|0.490|0.216|8.110|
> ||25|0.940|0.456|0.492|**0.224**|8.082|
> |SA ↓|5|0.982|0.600|0.642|0.232|7.043|
> ||10|**0.990**|0.578|0.564|0.202|7.192|
> ||15|0.988|0.608|**0.690**|**0.258**|6.735|
> ||20|0.980|**0.638**|0.684|0.252|**6.657**|
> ||25|0.990|0.616|0.671|0.242|6.921|
>
>
>
> **4. Partial-Success Reward**
>
> As shown in Table R5, we tested partial-success reward values in {0, 0.2, 0.4, 0.6, 0.8, 1.0}, where this reward provides a non-zero score for chemically valid but unsuccessful edits. The results show that removing the partial reward (0) prevents the model from distinguishing valid-but-incorrect molecules, leading to unstable optimization and reduced validity. In contrast, very large partial rewards (≥ 0.6) blur the distinction between successful and unsuccessful edits, which lowers accuracy and increases FCD. Moderate values in the range 0.2–0.4 yield the most balanced performance. Based on these observations, we adopt **0.2 as the default partial-success reward**.
>
> **Table R5: Sensitivity analysis of partial-success reward values**
>
> |Task|Partial Reward|Validity|Acc$_{\text{all}}$ (TS≥0.65)|Acc$_{\text{all}}$ (MCS≥0.6)|Acc$_{\text{all}}$ (GED≤4)|FCD↓|
> |-|-|-|-|-|-|-|
> |LogP ↑|0.0|0.952|**0.484**|0.508|0.202|7.906|
> ||0.2|**0.976**|0.462|0.498|0.214|**7.812**|
> ||0.4|0.962|0.464|**0.528**|**0.218**|7.859|
> ||0.6|0.958|0.480|0.504|0.202|8.234|
> ||0.8|0.964|0.434|0.468|0.208|8.890|
> ||1.0|0.964|0.402|0.438|0.216|9.458|
> |SA ↓|0.0|0.976|0.600|0.634|0.234|7.110|
> ||0.2|**0.988**|0.608|**0.680**|**0.258**|**6.735**|
> ||0.4|0.984|**0.626**|0.656|0.220|7.142|
> ||0.6|0.980|0.542|0.582|0.208|7.773|
> ||0.8|0.984|0.590|0.594|0.212|7.998|
> ||1.0|0.984|0.530|0.556|0.196|8.530|
>
> ---
>
> ### **Q2: Train/Test Split Construction**
>
>
> All molecules are sampled from the ZINC database and filtered by Lipinski’s Rule of Five. To prevent data leakage, we construct the splits at the molecule level using canonical SMILES: duplicate or equivalent structures are removed, and **no molecule in the test set appears in the training set or in any RL fine-tuning inputs**. This ensures that all test-time molecules are **entirely unseen** by the model.

---

> ### Author Response · Authors · 2025-11-22
> **Response to Reviewer niJm (3/3)**
>
> ### **Q3: Inference Efficiency Comparison**
>
> As shown in Table R6, we compare the baselines and MolEditRL under different skip-step settings during the reverse diffusion process. Here, step denotes the stride of the denoising schedule: with 2000 total diffusion steps, step = 500 corresponds to 4 denoising predictions, whereas step = 50 corresponds to 40 predictions.
>
> The results indicate that even with very coarse schedules such as step = 500, MolEditRL maintains competitive accuracy, achieves lower FCD than large LLM-based baselines, and preserves low per-sample inference time. All timings were measured on an A6000 server. These findings demonstrate that MolEditRL is robust to step skipping and provides a strong balance between efficiency and generation quality. Based on these observations, we adopt **step = 50** in the main experiments, as it offers the best trade-off between accuracy and inference efficiency.
>
>
> **Table R6: Inference efficiency comparison across baselines and different denoising step settings**
>
> | Method | Validity | Acc$_{\text{all}}$(TS≥0.65) | Acc$_{\text{all}}$(MCS≥0.6) | Acc$_{\text{all}}$(GED≤4) | FCD↓ | Time/Sample (s) |
> |--------|----------|-----------------------------|------------------------------|---------------------------|-------|------------------|
> | BioT5 | 1.000 | 0.000 | 0.004 | 0.000 | 13.4000 | 0.7550 |
> | DrugAssist | 0.988 | 0.537 | 0.551 | 0.202 | 9.0500 | 1.5221 |
> | Gellm$^4$o-C\_L  | 0.849 | 0.218 | 0.224 | 0.104 | 10.6700 | 4.5600 |
> | MolEditRL (step=500) | 0.900 | 0.536 | 0.590 | 0.220 | 7.9273 | 1.0608 |
> |  (step=400) | 0.924 | 0.568 | 0.628 | 0.240 | 7.5318 | 1.0684 |
> |  (step=250) | 0.952 | 0.588 | 0.644 | 0.234 | 7.3480 | 1.0954 |
> |  (step=200) | 0.956 | 0.576 | 0.638 | 0.234 | 7.4078 | 1.1135 |
> |  (step=100) | 0.972 | 0.590 | 0.668 | 0.232 | 7.1624 | 1.2039 |
> |  **(step=50)** | **0.988** | **0.608** | **0.680** | **0.258** | **6.7350** | **1.3857** |
> |  (step=40) | 0.979 | 0.624 | 0.682 | 0.234 | 6.5894 | 1.4778 |
> |  (step=20) | 0.988 | 0.628 | 0.686 | 0.246 | 6.4828 | 1.9286 |
> |  (step=10) | 0.984 | 0.636 | 0.690 | 0.238 | 6.2055 | 2.8393 |
>
>
> ---
>
> ### **Q4: Typo in Equation 5**
> Equation (5) in the paper is written as:
> $
> q(G^{1:T}_{\text{tgt}}\!\mid G^{0}_{\text{tgt}}) \;=\; \prod\nolimits_{\substack{t=1}}^T q(G^{t}_{\text{tgt}}\!\mid G^{t-1}_{\text{tgt}}),
> \quad
> q(G^{t}_{\text{tgt}}\!\mid G^{t-1}_{\text{tgt}}) = \bigl(V^{t-1}_{\text{tgt}}\,{Q}^V_t,\; E^{t-1}_{\text{tgt}}\,{Q}^E_t\bigr),
> $
> In our notation, the lowercase $(q(\cdot))$ denotes the **forward diffusion probability distribution**, while the uppercase $(Q_t^V)$ and $(Q_t^E)$ represent the **discrete transition matrices** for atoms and bonds.
> The distinction between lowercase ($q$) and uppercase ($Q$) is therefore intentional rather than a typographical mistake.
> To avoid any possible confusion, in the revised version we will explicitly clarify the meaning. If you has further concerns about this equation, we welcome additional questions for clarification.

---

> ### Comment · Reviewer_niJm · 2025-11-28
>
> Thanks to the authors for the detailed reply. The revisions and additional experiments adequately resolve my earlier questions. I will keep my positive score.

---

### Author Response · Authors · 2025-12-01
**Summary of Reviews and Rebuttal Updates for Paper 6541**

**Dear Area Chair,**

We appreciate your efforts in handling our submission. To assist your efficient assessment, we provide a summary of the reviewer consensus and the key updates made during the rebuttal.

**1. Overview & Consensus: 4/5 Positive Reviews**

Our paper received **4 positive ratings (8, 6, 6, 6)** and 1 negative rating (2).
- Reviewer rzAV (Score 8): Expressed strong support, identifying the **"structure-aware attention mechanism"** as a key innovation and praising the **"significant outperformance over baselines"**.
- Reviewers niJm, zGL2, & F1Hf (Score 6s): All three responded positively. niJm and zGL2 explicitly confirmed that the new experiments **fully resolved their earlier concerns**. F1Hf noted that the rebuttal results **"look quite convincing"**, and we have addressed their final question regarding oracle selection.

**2. Addressing the Main Dissent (Reviewer JXjh, Score 2)**

The single negative rating was driven by concerns regarding pre-training evaluation, oracle hacking, and the necessity of natural language. We have addressed these extensively with new experiments that were well-received by other reviewers:
- **Pre-training Validation:** We added standalone quantitative metrics (**Appendix I**) isolating the pre-training stage performance, demonstrating that the diffusion model learns a strong structural prior before any RL fine-tuning.
- **No "Oracle Hacking":** We compared MolEditRL against RL baselines (GCPN, MolDQN, REINVENT4). Results (**Appendix J**) show MolEditRL achieves higher validity (95.8%) and better synthesizability than baselines, confirming that our KL-regularized objective effectively preserves chemical realism.
- **Necessity of Natural Language:** We introduced complex, localized editing tasks (e.g., "Remove a $CO_2H$ group...") in **Appendix L**. Results show MolEditRL successfully performs fine-grained structural edits that cannot be achieved via simple property tags.

**3. Key Manuscript Updates**

We have uploaded a revised paper with new analyses requested by reviewers:
- **Efficiency**: Validated high inference speed (**1.39s/sample**) and superior oracle-query/computational efficiency against RL baselines (**Appendix N, O, P**)
- **Robustness & Stability**: Verified system stability under **noisy oracles** (**Appendix M**) and varying **hyperparameters** (KL weight, update stride, top-k sampling, and partial rewards) (**Appendix D, E, F, G**).
- **Mechanism Ablations:** Quantified the critical contributions of the structure-aware attention mechanism and the pre-training stage (**Appendix H, I**).
- **Advanced Evaluation:** Demonstrated robustness to diverse prompts (**Appendix K**), success on complex localized editing tasks (**Appendix L**), and superior chemical realism compared to RL baselines (**Appendix J**).

We believe the robust experimental results and the strong consensus from 4 out of 5 reviewers confirm the value of this work. Thank you for your time and effort in handling this submission.

Sincerely,

The Authors

---

### Meta-Review · Area_Chair_7gtj · 2026-01-06

**Summary:**

This submission received a strong overall response, with 4 out of 5 reviewers providing positive evaluations (scores: 8, 6, 6, 6). Reviewers consistently highlighted the novelty and effectiveness of the proposed approach, particularly the structure-aware attention mechanism, and noted the substantial performance gains over relevant baselines.

The authors’ rebuttal was thorough and responsive; critically, all substantive concerns from the initial review, including those raised by reviewer JXjh, have now been convincingly resolved through additional analyses and experiments.

**Reviewer Concerns:**

All the reviewers' concerns have been addressed.

**Reviewer Scores:**

JXjh might consider raising his/her score if they have the chance to participate.

---

### Decision · Program_Chairs · 2026-01-26

Accept (Poster)